Resource

# Multimodal profiling reveals site-specific adaptation and tissue residency hallmarks of γδ T cells across organs in mice

Anastasia du Halgouet [1,8], Kerstin Bruder[2], Nina Peltokangas[3,4,5], Aurélie Darbois [1], David Obwegs[2], Marion Salou [1], Robert Thimme [2], Maike Hofmann [2], Olivier Lantz [1,6,7] & Sagar [2] ✉

γδ T cells perform heterogeneous functions in homeostasis and disease across tissues. However, it is unclear whether these roles correspond to distinct γδ subsets or to a homogeneous population of cells exerting context-dependent functions. Here, by cross-organ multimodal single-cell profiling, we reveal that various mouse tissues harbor unique site-adapted γδ subsets. Epidermal and intestinal intraepithelial γδ T cells are transcriptionally homogeneous and exhibit epigenetic hallmarks of functional diversity. Through parabiosis experiments, we uncovered cellular states associated with cytotoxicity, innate-like rapid interferon-γ production and tissue repair functions displaying tissue residency hallmarks. Notably, our observations add nuance to the link between interleukin-17-producing γδ T cells and tissue residency. Moreover, transcriptional programs associated with tissue-resident γδ T cells are analogous to those of CD8⁺ tissue-resident memory T cells. Altogether, this study provides a multimodal landscape of tissue-adapted γδ T cells, revealing heterogeneity, lineage relationships and their tissue residency program.

Receptor-based classification of T cells segregates them into αβ and γδ T cell lineages. Deciphering the nonredundant functions of γδ T cells compared to their αβ counterparts is a matter of intense research[1–3]. While αβ T cells constitute a major fraction of T cells in murine organs, γδ T cells dominate in numbers in some tissues, for example, epidermis and the small intestine[4]. γδ T cells are important in various physiological processes and maintain normal tissue functions at steady state[5–8]. During immune threats, γδ T cells carry out diverse functions in different tissues ranging from direct killing of the infected cells to neutrophil recruitment and enhanced antigen presentation[9–11]. They

can also take on protective roles by downmodulating the inflammatory response and promoting tissue repair[5,12–18]. However, it is unclear whether these functions correspond to distinct γδ subsets or to a few subsets exhibiting context-dependent activities depending upon their microenvironment. Deciphering this conundrum requires systematic profiling of γδ T cells across multiple organs at single-cell resolution.

Additionally, understanding the tissue adaptation features of γδ T cells can provide essential insights to promote protective immunity. It has been shown that CD8⁺ tissue-resident memory T (T_RM) cells, mucosal-associated invariant T (MAIT) cells and natural killer T (NKT)

¹Institut National de la Santé et de la Recherche Médicale U932, PSL University, Institut Curie, Paris, France. ²Department of Medicine II (Gastroenterology, Hepatology, Endocrinology, and Infectious Diseases), Freiburg University Medical Center, Faculty of Medicine, University of Freiburg, Freiburg, Germany. ³Max Planck Institute of Immunobiology and Epigenetics, Freiburg, Germany. ⁴Faculty of Biology, University of Freiburg, Freiburg, Germany. ⁵Würzburg Institute of Systems Immunology, Max Planck Research Group at the Julius-Maximilians-Universität Würzburg, Würzburg, Germany. ⁶Laboratoire d'Immunologie Clinique, Institut Curie, Paris, France. ⁷Centre d'Investigation Clinique en Biothérapie Gustave-Roussy Institut Curie (CIC-BT1428) Institut Curie, Paris, France. ⁸Present address: National Institute of Dental and Craniofacial Research, National Institutes of Health, Bethesda, MD, USA. ✉e-mail: sagar@uniklinik-freiburg.de

cells share a common transcriptional program of tissue residency[19,20]. By contrast, the existence of a tissue residency program for γδ T cells remains unknown. Using parabiotic mice, it is well established that the skin, intestine, liver and adipose tissue host mainly tissue-resident γδ T cell populations[8,21–23]. However, parabiosis experiments in these studies were performed using flow cytometry, providing limited insights into their tissue residency features. Hence, deciphering the tissue adaptation features of γδ T cells in multiple organs combining parabiosis with single-cell transcriptomics holds the key to identifying specific tissue-resident γδ subsets and their underlying regulatory programs.

Here, utilizing multimodal single-cell sequencing, we profiled γδ T cells across seven organs and grouped them into eight subsets delineated by the expression of *Sell*, *Ly6c2*, *Cd160*, *Gzmb*, *Rorc*, *Areg* and *Klrg1* and cell cycling genes. Single-cell profiling of cell surface proteins further revealed the functional capacities of these identified subsets. Further, we revealed that epidermal and intestinal γδ T cells exhibit a homogenous transcriptional profile and showcase open chromatin at gene loci associated with diverse functions. Single-cell T cell antigen receptor (TCR) profiling revealed the lineage relationships of the identified subsets. Furthermore, using parabiotic mice, we uncovered the cellular and molecular hallmarks of γδ T cell tissue residency across organs. Moreover, comparing the tissue residency features of γδ T cells to those of $T_{RM}$ and NKT cells revealed analogous transcriptional programs of tissue residency. Altogether, our data represent a cross-organ single-cell multimodal landscape of γδ T cells and provide a highly resolved map of their tissue residency features.

## Results

### Heterogeneity and tissue-adapted features of γδ T cells
To investigate the tissue-specific heterogeneity and site-adapted features of γδ T cells in mice, we used a single-cell multimodal approach and sorted γδ T cells from various organs using flow cytometry (Fig. 1a,b, Extended Data Fig. 1a and Supplementary Note). Sorted cells were subjected to single-cell RNA sequencing (scRNA-seq), single-cell assay for transposase accessible chromatin during sequencing (scATAC-seq), single-cell TCR sequencing and cell surface profiling using TotalSeq. Integration, clustering and visualization of gene expression data were performed using Harmony and Seurat[24,25] (Extended Data Fig. 1b–e and Supplementary Note). We identified 22 clusters in the dataset, revealing substantial heterogeneity in the γδ T cell compartment across tissues (Fig. 1c). Cells from different tissues contributed differentially to these clusters, indicating site-specific adaptation (Fig. 1d–f). Principal component analysis (PCA) based on average gene expression profiles revealed that γδ T cells from the skin, small intestine and large intestine are remarkably distinct compared to cells from other organs (Fig. 1g).

Next, we characterized the identified γδ T cell clusters using differentially expressed genes (Supplementary Table 1). We grouped 22 clusters into 8 major subsets with unique gene expression profiles (Fig. 2a–c). *Sell*+*Ly6c2*− (clusters 0, 7 and 11) and *Sell*+*Ly6c2*+ (clusters 2 and 8) subtypes exhibited the highest expression of genes associated with lymphocyte migration (*Sell* and *S1pr1*) and maturation (*Tcf7* and *Lef1*)[26–30] (Fig. 2d). Clusters 4, 6, 10, 12 and 14 were classified as *Cd160*+ γδ T cells (Fig. 2a,b,d). CD160 has been shown to control interferon-γ (IFN-γ) secretion by natural killer (NK) cells[31]. We reasoned that *Cd160*+ γδ T cells may represent a distinct site-adapted subset of IFN-γ-producing cells across different organs (Fig. 2e). Clusters 1 and 9 mainly consisted of small and large intestinal γδ intraepithelial lymphocytes (IELs) expressing *Cd8a* and *Itgae* (encoding CD103)[32], categorized as *Gzmb*+ IELs due to their cytotoxic phenotype (*Gzma* and *Gzmb*; Fig. 2d). Clusters 3, 5 and 19 consist of interleukin-17-producing γδ T (γδT17) cells, characterized by the expression of *Rorc* (encoding RORγt), *Il17a* and *Zbtb16*, which controls the development of Vγ6+ γδT17 cells[33,34] (Heilig and Tonegawa nomenclature[35]; Fig. 2d). We termed these cells *Rorc*+ γδ T cells. Interestingly, skin *Rorc*+ γδ T cells (mainly cluster 5) clustered separately from their counterparts in other organs (cluster

3) and exhibited a distinct transcriptional signature (Fig. 1c,d, Extended Data Fig. 1f and Supplementary Table 2). Moreover, skin harbored a transcriptionally distinct cluster 13 characterized by the expression of *Areg*, *Ctla2a* and *Gem* (Fig. 2c–e). We denoted this subtype as the *Areg*+ γδ T cell subset. This cluster represents dendritic epidermal T cells (DETCs) previously described as expressing *Areg* and playing a tissue repair function following injury[36,37]. We further identified two minor groups of γδ T cells—an effector-like subset (cluster 17, *Klrg1*+) and proliferating cells (cluster 15)[38,39] (Fig. 2b,d and Extended Data Fig. 1g). Taken together, these results demonstrate that γδ T cells constitute a heterogeneous group of site-specific T cells that are highly adapted to their local microenvironment.

### Maturation states and functional capacities of γδ subsets
We next refined the phenotypes of identified γδ subsets using a panel of TotalSeq antibodies (Fig. 3a,b and Extended Data Fig. 2). *Sell*+*Ly6c2*− and *Sell*+*Ly6c2*+ cells also expressed CD62L (encoded by *Sell*), indicating concordance between gene expression and cell surface phenotype (Fig. 3c). *Sell*+*Ly6c2*− γδ T cells expressed the highest levels of CD62L and CD24, an immature T cell marker (Fig. 3c). Meanwhile, CD122, a marker often associated with IFN-γ-producing cells[40], was highly expressed by *Sell*+*Ly6c2*+ cells (Fig. 3c)[41]. Both subsets expressed CD27, which is also associated with IFN-γ-producing cells[42] (Fig. 3c). Notably, *Cd160*+ γδ T cells expressed NK-1.1, validating our gene expression-based classification, as this marker has been associated with IFN-γ production[43,44] (Fig. 3c). *Gzmb*+ IELs exhibited the highest levels of CD8a (Fig. 3c). The cell surface expression of KLRG1 was specific to *Klrg1*+ cells (Fig. 3c). *Rorc*+ cells expressed cell surface markers associated with the γδT17 cell lineage, such as CD44, ICOS and CCR6 (Fig. 3c). Interestingly, a fraction of *Rorc*+ cells also expressed Nkp46, classically expressed by NK cells[45] (Fig. 3c). Furthermore, we used cell surface proteins for dimensionality reduction, identifying many γδ subsets solely through these markers, highlighting the functional relevance of our gene expression-based classification (Fig. 3d). We sought to further validate the markers used to classify the γδ subtypes utilizing antibodies against LY6C and CD160 as well as the RORγt-GFP reporter line[46]. We first assessed the suitability of identified genes to define the eight γδ subsets. We observed that the expression of these genes was exclusive to the defined subtypes, indicating their appropriateness to classify γδ T cell subsets (Fig. 3e and Extended Data Fig. 3). We further quantified the fraction of these subsets in each organ using scRNA-seq as well as flow cytometry revealing a remarkable similarity in the calculated fraction of γδ T cells between both methods (Fig. 3f–i and Extended Data Fig. 4). Overall, flow cytometry analysis validated the gene expression-based quantification of *Sell*+*Ly6c2*−, *Sell*+*Ly6c2*+, *Cd160*+ and *Rorc*+ γδ subsets across organs.

### Epigenetic features and functional diversity of γδ subsets
To understand the underlying regulatory networks of γδ cell states, we simultaneously profiled gene expression and chromatin accessibility in γδ T cells across different organs. Clustering based on DNA accessibility using Signac[47] classified them into 17 clusters (Fig. 4a and Extended Data Fig. 5a–d). Importantly, the gene expression data from the same cells were already integrated into the transcriptome-based clustering shown in Fig. 1c, allowing us to map the transcriptome-based cell-type annotation on the chromatin accessibility-based clustering depicted in Fig. 4a. Assigning each scATAC-seq cluster to one of the eight γδ subsets indicated high consistency between the gene expression and chromatin accessibility (Fig. 4b and Extended Data Fig. 5e). We further looked at the differentially accessible peaks in each cluster (Fig. 4c and Supplementary Tables 3 and 4). Clusters 0, 5 and 10, which were classified as *Sell*+*Ly6c2*− cells, also exhibited open chromatin regions associated with *Tcf7*, *Lef1*, *Sell* and *Cd24a* (Fig. 4a–c and Extended Data Fig. 5f). Cluster 2, consisting of *Sell*+*Ly6c2*+ cells, exhibited higher accessibility at the *Ifng* and *Tbx21* loci necessary for IFN-γ production[48]

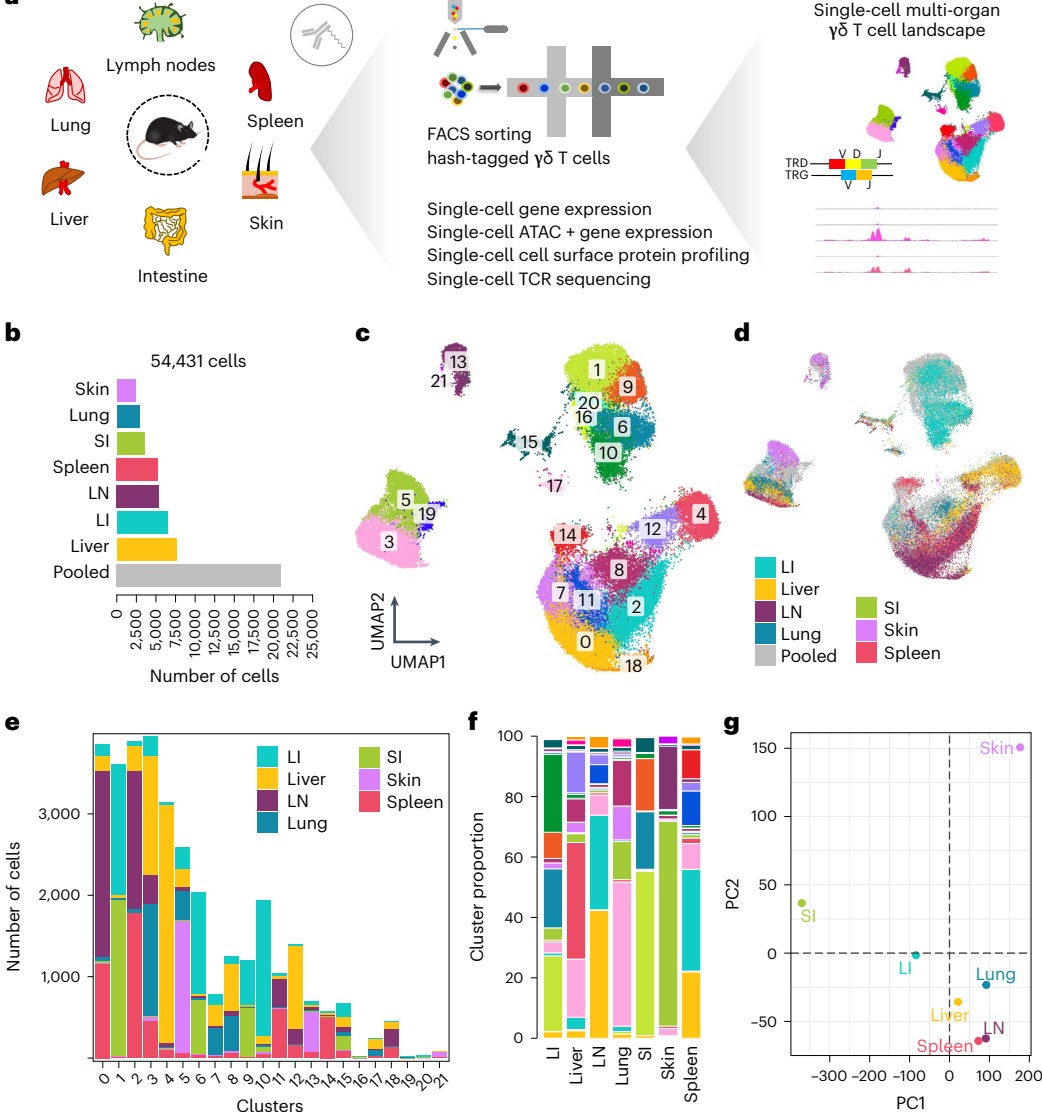

**Fig. 1 | Multi-organ γδ T cell profiling reveals heterogeneity and site-specific adaptation. a**, Schematic diagram showing the experimental workflow. **b**, Bar plot showing the number of cells sequenced from each organ. Note that cells profiled using simultaneous gene expression and chromatin accessibility were pooled without hashing and cannot be assigned to a particular organ. These cells are labeled 'pooled'. **c,d**, UMAP representation based on gene expression profiling depicting 22 clusters in the data (clusters 0 to 21; *n* = 54,431 cells from 11 mice examined over five independent experiments) (**c**) and the tissue of origin of each cell labeled in different colors (**d**). **e,f**, Bar plots showing the number of cells from different tissues in each cluster (**e**) and the cluster proportion in each tissue (**f**). **g**, PCA based on average gene expression profiles of γδ T cells from different tissues revealed that γδ T cells from the skin, small intestine and large intestine exhibit distinct transcriptional signatures compared to other organs. FACS, fluorescence-activated cell sorting; SI, small intestine; LI, large intestine; LN, lymph node.

(Fig. 4c and Extended Data Fig. 5g). Similarly to the gene expression-based classification, *Cd160*⁺ γδ T cells exhibited heterogeneity in their chromatin accessibility landscape and were subdivided into three different subsets—clusters 3, 7 and 4—based on their tissue of origin, indicating site-specific adaptation of this subset (Fig. 4a,b). Clusters 1, 9 and 12, classified as *Gzmb*⁺ IELs, exhibited open regions across *Cd8a*, *Kit*, *Gzma*, *Gzmb* and *Gzmk* loci (Fig. 4c). Uniquely, γδ IELs displayed differentially open chromatin regions across genes encoding various interleukins (Fig. 4c and Extended Data Fig. 6a,b). Notably, we did not detect their transcripts (Extended Data Fig. 6c,d). This suggests that although IELs do not express these cytokines at steady state, they might have the propensity to express them during infection or inflammation which could explain their previously reported protective role[49,50]. Based on the chromatin accessibility features, *Rorc*⁺ γδ T cells were substantially

heterogeneous and grouped into five different clusters—clusters 8, 11, 15, 16 and 14. Cluster 14 specifically comprised skin γδ T17 cells and exhibited several similar chromatin accessibility features as the *Areg*⁺ γδ subset, indicating skin-specific adaptation (Fig. 4c). Cluster 15 showed open chromatin at gene loci belonging to both IFN-γ-associated and IL-17-associated programs (*Rorc*, *Il17a* and *Tbx21*; Extended Data Fig. 6e–g). Finally, the skin-derived *Areg*⁺ γδ T cell subset revealed a unique chromatin accessibility landscape with the highest number of differentially regulated open chromatin regions (Fig. 4c and Extended Data Fig. 6h). These included several genes encoding interleukins and growth factors without evident transcription (Fig. 4c and Extended Data Fig. 6i–n). This suggests that the *Areg*⁺ γδ subset has the propensity to release factors that may have an important role in promoting tissue repair and wound healing[37].

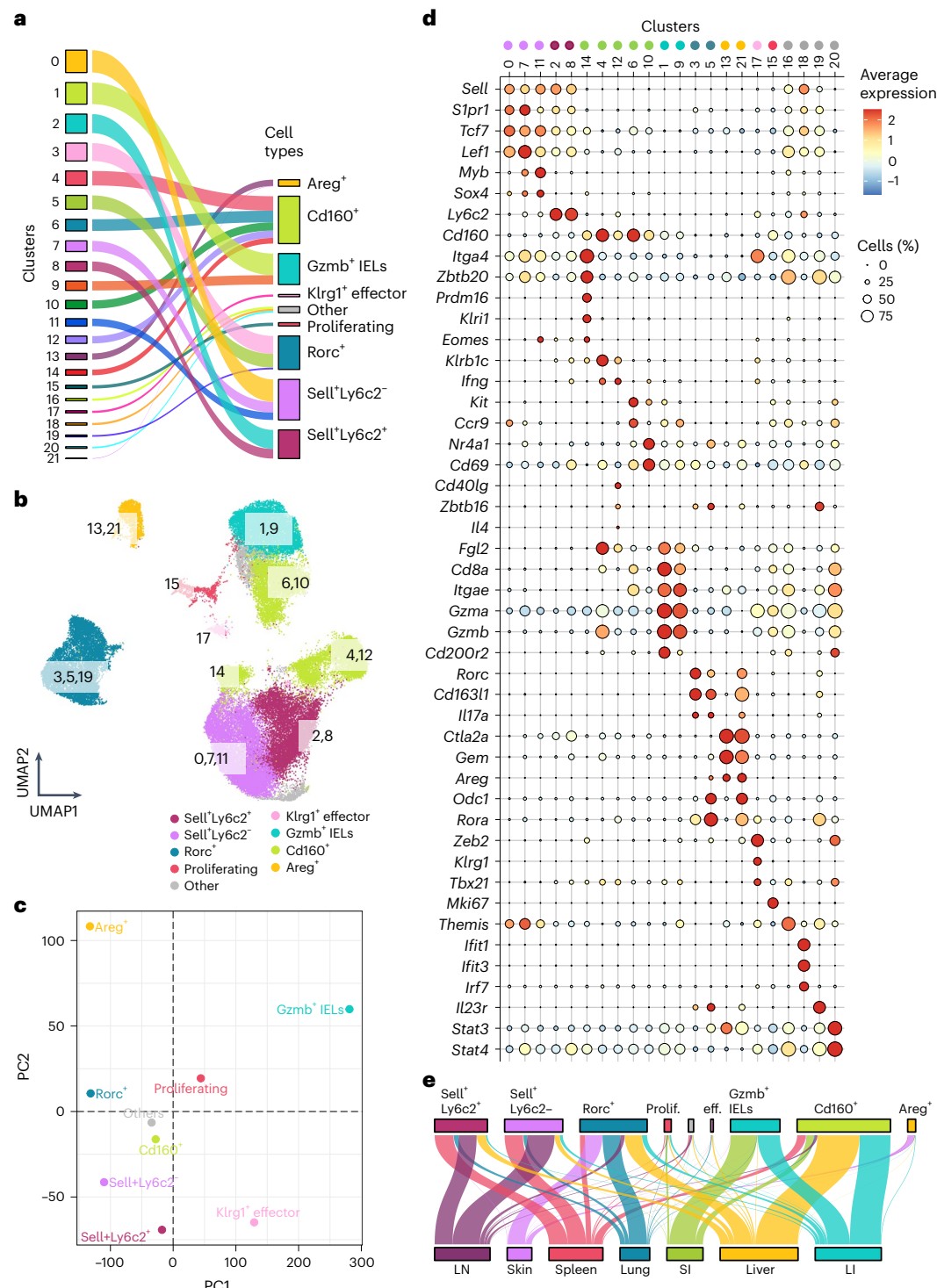

**Fig. 2 | Eight major site-adapted γδ subsets across tissues. a,** Alluvial plots connecting the cell clusters shown in Fig. 1c to cell types. Based on the gene expression profiles, we categorized 22 γδ T cell clusters into eight subsets. Cells not assigned to a particular subset are shown as 'other'. **b,** UMAP representation showing the γδ cell subsets in different colors. Numbers represent scRNA-seq clusters shown in Fig. 1c. **c,** PCA revealed that intestinal IELs and *Areg*⁺ γδ T cells

from the skin exhibit a distinct transcriptional signature compared to the other γδ subsets. **d,** Dot plot showing the key differentially expressed genes in each cluster shown in Fig. 1c. Color represents the mean expression of the gene in the respective cluster, and dot size represents the fraction of cells in the cluster expressing the gene. **e,** Alluvial plots connecting eight γδ subsets shown in Fig. 2b to the tissue of their origin.

To identify transcription factor (TF) motifs that characterize γδ subsets, we used chromVAR[51]. TF motifs belonging to the TCF/LEF family were enriched in clusters 0, 5 and 10 composed of *Sell⁺ Ly6c2⁻* cells (Fig. 4d and Extended Data Fig. 7a). Chromatin regions associated with the T-box family of TFs were enriched in clusters 2 and 3 containing

*Sell⁺Ly6c2⁺* and liver *Cd160⁺* cells, respectively (Fig. 4d and Extended Data Fig. 7b). Cluster 7, representing splenic *Cd160⁺* cells, displayed specific motif enrichment of the ETS family of TFs (Fig. 4d). The IRF/STAT family of TFs was enriched in two clusters of intestinal γδ IELs (clusters 1 and 9; Fig. 4d and Extended Data Fig. 7c). Another intestinal cluster,

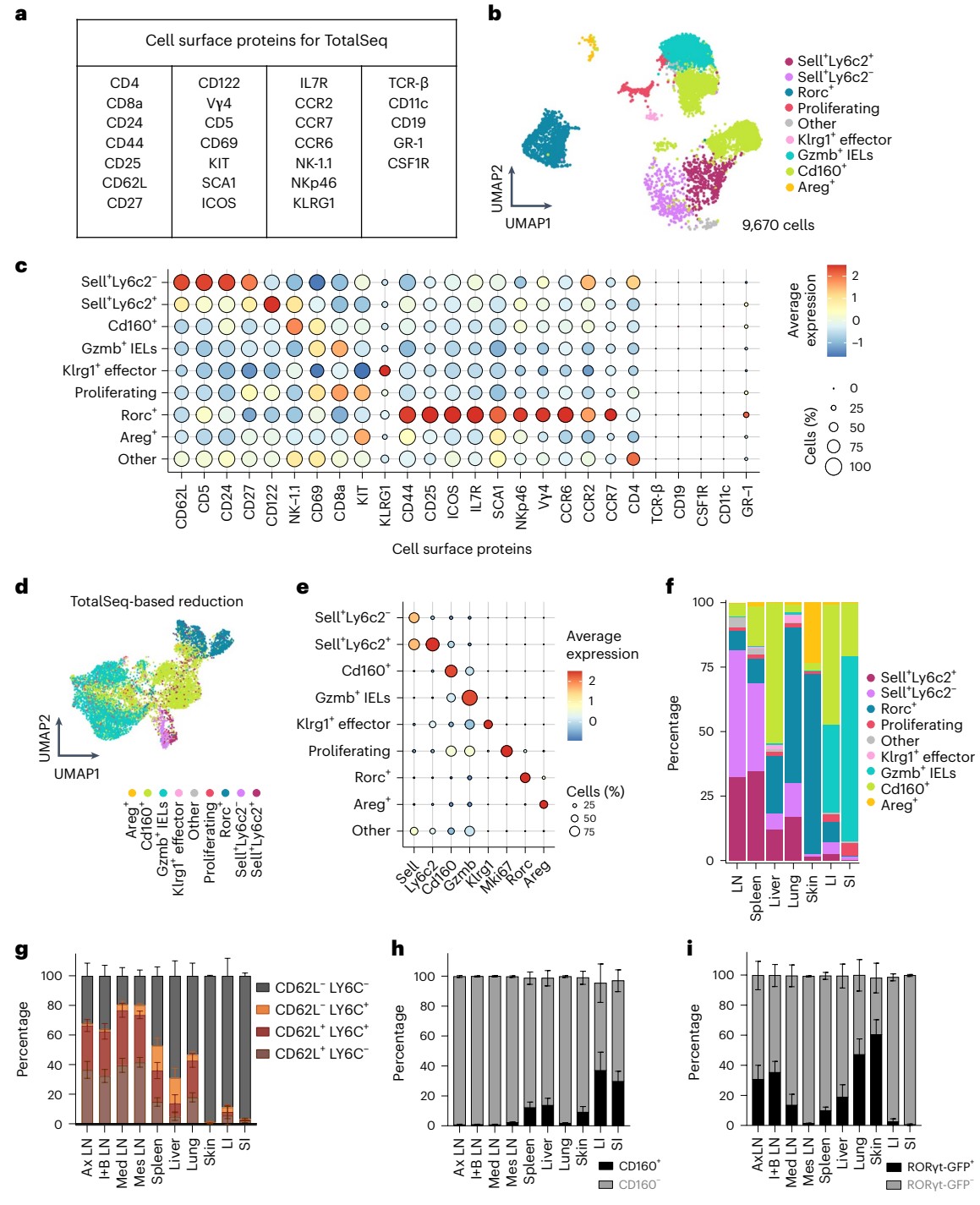

**Fig. 3 | Single-cell epitope profiling of γδ T cell subsets. a**, Table listing 26 antibodies used to profile γδ T cells using TotalSeq. **b**, UMAP representation depicting 9,670 cells that were simultaneously profiled for gene expression and 26 cell surface proteins (*n* = 3 mice). Colors represent different γδ subsets. **c**, Dot plot showing the normalized mean expression of analyzed cell surface proteins in the respective γδ subset. The dot size represents the fraction of cells in the subset expressing a cell surface protein. **d**, UMAP representation of dimensionality reduction performed based on cell surface protein expression. Colors represent different γδ subsets identified based on gene expression.

**e**, Dot plot showing the expression of genes used to classify eight γδ subsets. Color represents the mean expression of the gene in the respective γδ subset, and dot size represents the fraction of cells in that subset expressing the gene. **f**, Bar plot showing the fraction of identified γδ subsets in each organ based on scRNA-seq data. **g**–**i**, Quantification of CD62L+Ly6C− and CD62L+Ly6C+ (**g**), CD160+ (**h**) and RORγt-GFP+ (**i**) γδ T cells using flow cytometry (*n* = 8 mice). The bar graphs depict the mean ± s.d. (error bars). Ax LNs, axillary lymph nodes; I + B LNs, inguinal and brachial lymph nodes; Med LNs, mediastinal lymph nodes; Mes LNs, mesenteric lymph nodes.

cluster 12, had specific enriched motifs of NFAT family TFs (Fig. 4d). Interestingly, *Rorc*+ γδ T cells displayed substantial heterogeneity in enriched TF motifs of RAR-related orphan receptors, GATA3, GATA4, Krüppel-like and nuclear factor-κB family members (Fig. 4d and

Extended Data Fig. 7d). Finally, *Areg*+ cells exhibited enriched TF motifs from the RUNX and AP1 families (Fig. 4d and Extended Data Fig. 7e). Altogether, single-cell chromatin accessibility profiling reveals distinct epigenetic features of tissue-adapted γδ T cell subsets (Extended Data

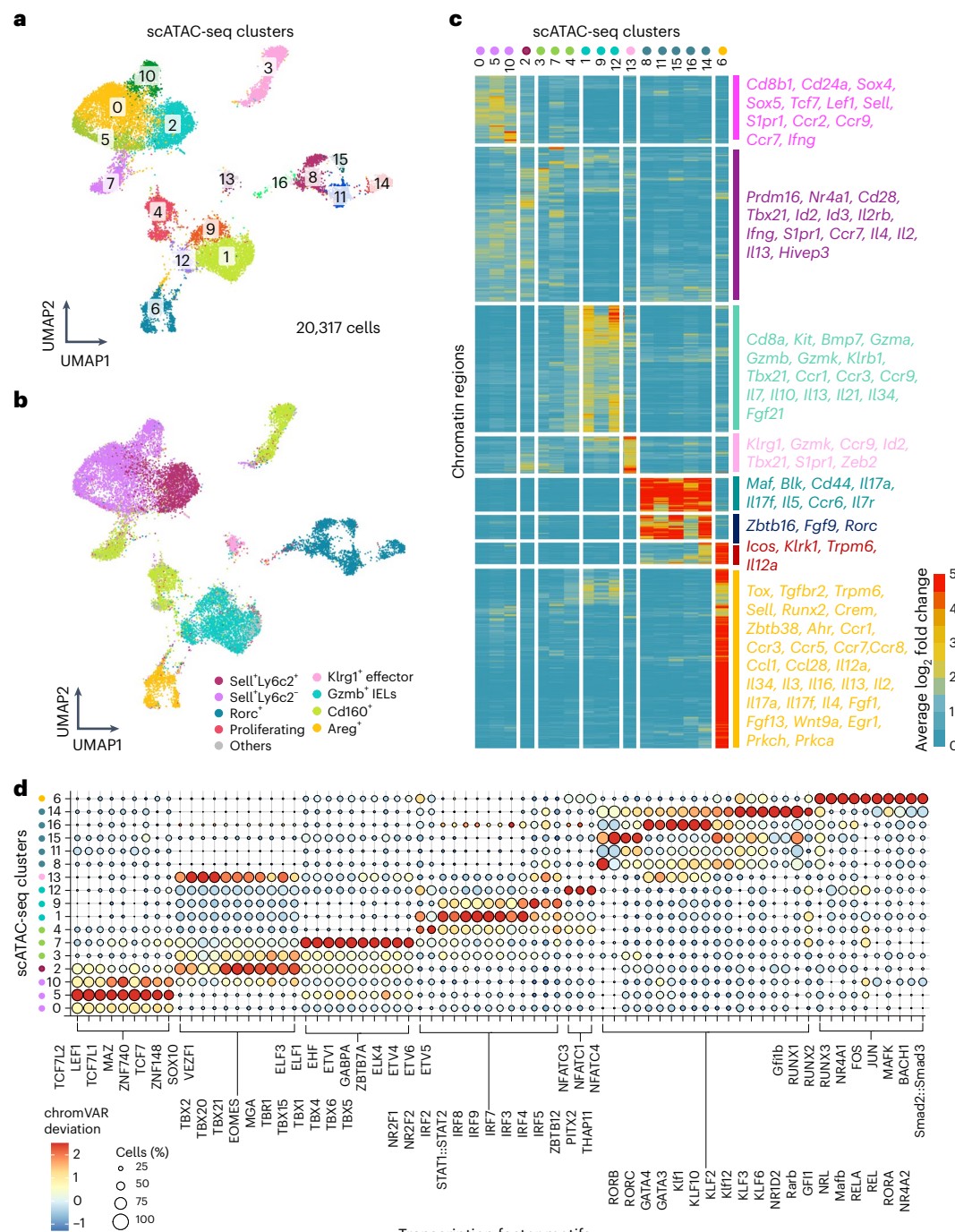

**Fig. 4 | Cross-tissue single-cell chromatin landscape of γδ T cell subsets.**
**a**, UMAP representation showing the 17 clusters (clusters 0–16) identified using the chromatin accessibility profiles of γδ T cells across seven organs (*n* = 20,317 cells from 6 mice examined over two independent experiments). **b**, UMAP representation based on chromatin accessibility showing the eight γδ subsets identified based on gene expression profiles in different colors. **c**, Heat map showing differentially regulated chromatin regions in each cluster. Color represents the average log$_2$-fold change of each chromatin region in each cluster compared to the rest. Key genes closest to the corresponding chromatin regions are listed. **d**, Dot plot showing chromVAR deviations for the top 10 enriched TF motifs in each scATAC-seq cluster. Differential testing for each cluster versus the rest was performed on the chromVAR *z*-score.

Fig. 7f). Moreover, we uncovered that epidermal and intestinal γδ T cells are rather transcriptionally homogeneous, showing simultaneous epigenetic features associated with cytotoxicity, cytokine production, tissue repair and wound healing, indicating functional diversity.

### TCR diversity and lineage relationships of γδ subsets
Next, we performed gene expression and TCR repertoire profiling of γδ T cells across organs (Fig. 5a–c, Extended Data Fig. 8a,b and Supplementary Table 5). Using the Shannon diversity score, we observed that the spleen and lymph nodes possessed a more diverse T cell receptor Gamma (TRG) repertoire compared to other organs (Fig. 5d and Extended Data Fig. 8c). *Sell⁺Ly6c2⁻* cells (clusters 0, 7 and 13) had small clones with the highest diversity score, highlighting its naïve features (Fig. 5e and Extended Data Fig. 8d). *Sell⁺Ly6c2⁺* (clusters 4 and 6) and *Cd160⁺* (clusters 3, 8, 11 and 12) subsets comprised medium-sized clones (Fig. 5e). *Gzmb⁺* γδ IELs (clusters 2 and 5) were mainly Vγ7⁺, with a

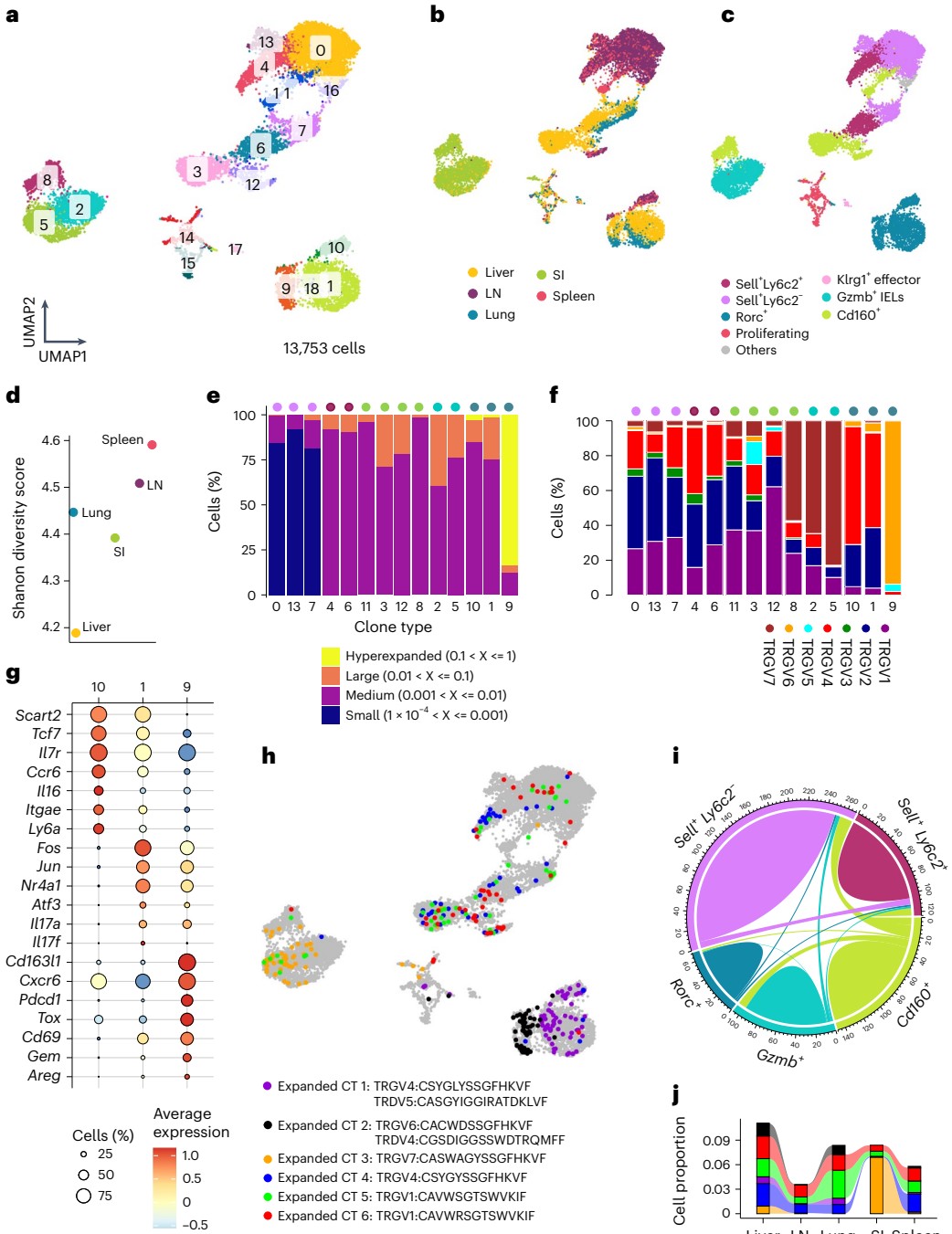

**Fig. 5 | TCR features and lineage relationships of γδ subsets across tissues.**
**a**–**c**, UMAP representation based on gene expression profiling depicting 19 clusters in the data where cells were simultaneously profiled for gene expression and TCR rearrangement configurations (clusters 0 to 18) (**a**) and the tissue of origin (**b**) as well as the annotated cell types of each cell labeled in different colors (*n* = 13,753 cells from three mice) (**c**). **d**, Dot plot showing the Shannon diversity score calculated based on TCR repertoire in different organs. Dots in different colors represent different organs. **e**, Bar plot showing the distribution of clonal sizes in the different clusters. Dots in different colors on the top of bars represent different cell types. **f**, Bar plot quantifying the variable γ-chain usage of γδ T cells within each tissue. Seven variable γ chains are depicted in different colors.

Dots in different colors on the top of bars represent different cell types. **g**, Dot plot showing the expression of selected genes differentially expressed between three *Rorc*⁺ clusters (clusters 10, 1 and 9). Color represents the mean expression of the gene in the respective cluster, and dot size represents the fraction of cells in that cluster expressing the gene. **h**, UMAP representation highlighting the cells with top six expanded clonotypes. Each expanded clonotype is represented in a different color. Amino acid sequence of the complementarity-determining region 3 (CDR3) region is further listed. **i**, Chord diagram depicting the clonal overlap among γδ subsets. **j**, Alluvial plots connecting the profiled organs to the top six expanded clonotypes depicted in Fig. 5h. CT, clonotype.

few cells exhibiting Vγ1 chain rearrangement, consistent with previous reports (Fig. 5f)[52,53]. Of particular interest was the *Rorc*⁺ subset, which exhibited tissue-specific heterogeneity and formed four clusters (clusters 1, 9, 10 and 18; Fig. 5a,c). Clusters 1 and 10 contained *Rorc*⁺ cells with

Vγ2 and Vγ4 usage (Fig. 5f). Cluster 9 consisted of Vγ6⁺ cells primarily from the liver and lung, exhibiting highly expanded TCR clonotypes and the lowest diversity score (Fig. 5c,e,f and Extended Data Fig. 8d). These Vγ6⁺ cells presented a distinct gene expression signature with

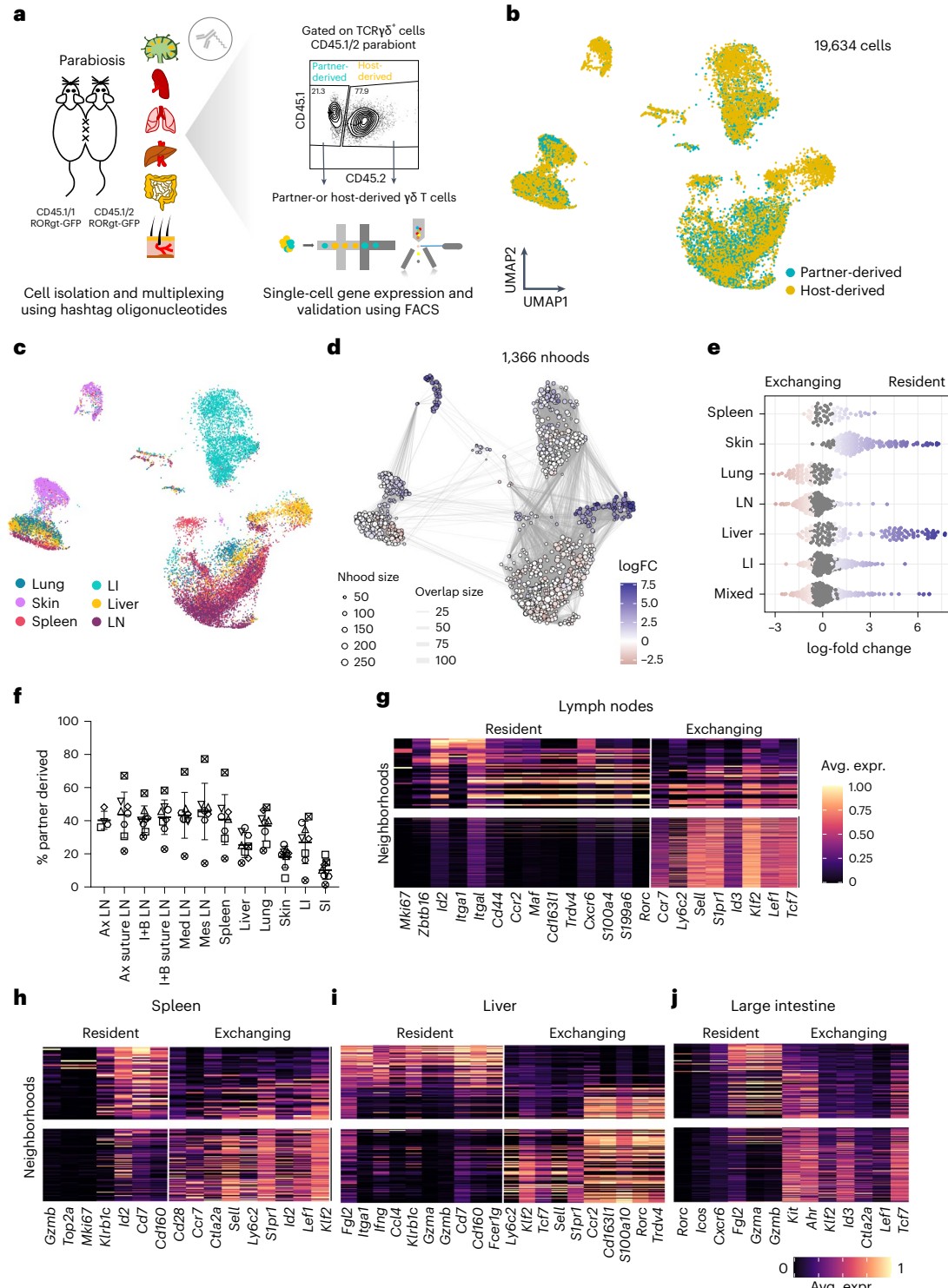

**Fig. 6 | Parabiosis reveals molecular features of γδ tissue residency across organs. a**, Schematic representation of the experimental design for the parabiosis experiments. **b**,**c**, UMAP representation showing the partner-derived and host-derived γδ T cells in different colors (**b**) and the tissue of origin of each cell (**c**). Colors represent different tissues in **c** (n = 19,634 cells from 6 mice; three parabiotic pairs). **d**, Neighborhood graph representation of the results obtained from Milo differential abundance testing. Nodes are neighborhoods, colored by their log fold change. Neighborhoods abundant in host-derived (resident) γδ T cells are depicted in blue. Non-differential abundance neighborhoods (false discovery rate (FDR) of 10%) are colored in white, and sizes correspond to the number of cells in each neighborhood. Graph edges depict the number of cells shared between neighborhoods. The layout of nodes is determined by the

position of the neighborhood index cell in the UMAP in **b**. **e**, Beeswarm plot of the distribution of log fold change in abundance between resident and exchanging compartments in neighborhoods containing cells from different tissues. Differentially abundant neighborhoods at an FDR of 10% are colored. **f**, Plot showing the quantification of partner-derived γδ T cells across different tissues (n = 8 mice; four parabiotic pairs). The graph depicts the mean ± s.d. (error bars). **g**–**j**, Heat map of key differentially expressed genes between neighborhoods abundant in resident and exchanging compartments in the lymph nodes (**g**), spleen (**h**), liver (**i**) and large intestine (**j**). Rows indicate neighborhoods, and columns denote shortlisted differentially expressed genes (FDR of 5%). Expression values for each gene are scaled between 0 and 1.

upregulation of *Cxcr6*, genes associated with chronic TCR stimulation (*Pdcd1* and *Tox*) and similar genes to those expressed by DETCs (*Areg* and *Gem*; Fig. 5g).

Next, we analyzed the six top expanded clones by plotting them on the uniform manifold approximation and projection (UMAP) representation (Fig. 5h). The first two belonged to *Rorc*⁺ cells exclusively, showing early lineage segregation (Fig. 5h). The third mainly came from *Gzmb*⁺ IELs and overlapped exclusively with intestinal *Cd160*⁺ cells suggesting a shared developmental origin (Fig. 5h). Further analysis of all detected clonotypes in *Gzmb*⁺ IELs revealed their maximum overlap exclusively with intestinal *Cd160*⁺ cells, emphasizing the uniqueness of this lineage (Fig. 5i). The next three highly expanded clones were predominantly scattered among *Sell*⁺*Ly6c2*⁻, *Sell*⁺*Ly6c2*⁺ and *Cd160*⁺ cells (Fig. 5h). This pattern hints toward a common developmental origin of these three subsets. Interestingly, the fifth highly expanded clonotype was shared among all cell subsets except *Rorc*⁺ cells, including *Gzmb*⁺ IELs (Fig. 5h). This finding suggests that *Gzmb*⁺ IELs with Vγ1 rearrangement and a transcriptional signature similar to the Vγ7 chain might share ontogeny with other γδ subsets. Moreover, the presence of these six highly expanded clones across all organs hints toward the existence of expanded precursors from the thymus seeding different tissues (Fig. 5j). Lastly, quantifying the clonal overlap among the profiled organs revealed that liver and lymph node clones exhibit the highest and least overlap with other tissues, respectively (Extended Data Fig. 8e,f). In summary, TCR clonotype analysis provided valuable insights into the distinct origins of *Gzmb*⁺ and *Rorc*⁺ subsets, while also hinting toward a common origin for *Sell*⁺*Ly6c2*⁻, *Sell*⁺*Ly6c2*⁺ and *Cd160*⁺ cells.

## Tissue residency programs of γδ T cells across organs

To understand the tissue residency features of γδ T cells across different organs, we conducted parabiosis experiments, profiling partner-derived and host-derived γδ T cells across six tissues using scRNA-seq (Fig. 6a). Importantly, the profiled cells were already integrated into the UMAP representation depicted in Fig. 1c. Here, our focus was on single cells from parabiotic mouse pairs (Fig. 6b,c). To distinguish tissue-resident cells from exchanging γδ T cell populations at higher resolution, we utilized the Milo framework, which models cellular states as overlapping neighborhoods on a *k*-nearest neighbor (KNN) graph[54]. Using this approach, we identified 1,366 neighborhoods that were differentially abundant in partner- and host-derived γδ T cells (Fig. 6d). The skin exhibited mostly tissue-resident neighborhoods, followed by the liver (Fig. 6e). The large intestine, lymph nodes and spleen also contained several tissue-resident neighborhoods (Fig. 6e). The lung, on the other hand, had very few tissue-resident neighborhoods (Fig. 6e). Independent flow cytometry data confirmed these observations (Fig. 6f and Extended Data Fig. 9a,b). To explore the transcriptional programs associated with tissue residency in each organ, we conducted differential gene expression analysis between partner-derived and host-derived neighborhoods. In lymph nodes, genes regulating proliferation (*Mki67*), the expression of integrin subunits (*Itga1* and *Itgal*) and the γδT17 program (*Maf* and *Rorc*) were associated with tissue residency (Fig. 6g). In the spleen and liver, molecular programs associated with cytotoxicity (*Gzmb* and *Gzma*) and innate-like IFN-γ production (*Ifng*, *Klrb1c* and *Cd160*) were mainly tissue resident (Fig. 6h,i). In contrast to lymph nodes, many neighborhoods associated with the γδT17 signature (*Rorc* and *Cd163l1*) comprised exchanging cellular states in the liver (Fig. 6i). In the large intestine, both cytotoxicity (*Gzmb*) and γδT17 (*Rorc*) signatures were associated with tissue residency (Fig. 6j). Importantly, akin to circulating memory CD8⁺ T cells, *Tcf7*, *Lef1*, *Klf2*, *Sell* and *S1pr1* were enriched in exchanging γδ states across organs (Fig. 6g–j). In summary, we reveal that cytotoxicity and rapid IFN-γ production-related molecular programs are general hallmarks of γδ tissue residency, while *Tcf7*, *Lef1* and *Klf2* define the core transcriptional program of circulating γδ T cells.

## Subset-specific residency features of γδ T cells

To explore the subset-specific tissue residency characteristics of γδ T cells, we performed automatic grouping of γδ neighborhoods and identified 15 groups (Fig. 7a). Groups 5, 6, 7, 9 and 10 were enriched in host-derived neighborhoods, while the rest comprised exchanging cells (Fig. 7b). Groups 5 and 6 consisted of *Areg*⁺ cells from the skin and *Cd160*⁺ cells from the liver, respectively, and were highly resident (Fig. 7b–d). Liver and spleen CD160⁺ cells were confirmed as more tissue-resident than CD160⁻ cells via flow cytometry (Fig. 7e,f). Group 7, containing skin *Rorc*⁺ cells, was also more abundant in resident neighborhoods, unlike *Rorc*⁺ cells from other organs (group 3; Fig. 7b–d), confirmed by flow cytometry (Fig. 7g). RORγt-GFP⁺ cells from epithelial tissues like lung and liver were not completely exchanging (Fig. 7h), suggesting distinct exchanging and resident γδ subsets in these tissues. We separately analyzed *Rorc*⁺ cells from the lung and liver, identifying common genes characterizing γδT17 tissue residency (Fig. 7i–k). Resident γδT17 cells exclusively expressed *Sdc4* and *Pdcd1* (encoding PD-1; Fig. 7l). PD-1 has been shown previously to be expressed by a subset of γδT17 cells that are Vγ6⁺ and display a T$_{RM}$ phenotype[55]. Next, we investigated γδ subsets in the skin. In the skin, two major γδ subsets were identified: *Rorc*⁺ (group 7) and *Areg*⁺ (group 5) cells (Fig. 7c,d). *Rorc* expression was absent in *Areg*⁺ cells, indicating the RORγt-GFP⁻ compartment should represent the highly resident *Areg*⁺ skin cells (Fig. 7m). Indeed, we detected very few partner-derived RORγt-GFP⁻ cells in the skin of parabiotic pairs of mice (Fig. 7m). Group 9, comprising *Gzmb*⁺ IELs, was also enriched in tissue-resident neighborhoods (Fig. 7b–d). On the other hand, groups 2, 11 and 12, consisting of *Sell*⁺*Ly6c2*⁻ and *Sell*⁺ *Ly6c2*⁺ cells, were mainly exchanging, supported by flow cytometry analysis (Fig. 7b–d and Extended Data Fig. 9c,d). Overall, we identified γδ subset tissue residency features and their associated molecular programs across tissues.

---

**Fig. 7 | Cell-type-specific residency features of γδ subsets across tissues.**
**a**, Neighborhood grouping, overlaid on the neighborhood graph as in Fig. 6d. Colors denote the assignment of neighborhoods to discrete groups using Louvain clustering. **b**, Beeswarm plot showing log fold change in abundance between resident and exchanging compartments in neighborhood groups. Differentially abundant neighborhoods at an FDR of 10% are colored. **c,d**, Bar plot showing the fraction of neighborhoods assigned to identified γδ subsets (**c**) and organ in each neighborhood group (**d**). **e,f**, Quantification of partner-derived total, CD160⁺ and CD160⁻ cells in the liver (**e**) and spleen (**f**) (*n* = 8 mice; four parabiotic pairs). Statistical analysis using Tukey's multiple-comparisons test (one-way analysis of variance (ANOVA)). Significance levels (with 95% confidence interval): \*\**P* < 0.01; \*\*\**P* < 0.001. Liver: \*\*\*Total versus CD160⁺ = 0.0002; \*\*Total versus CD160⁻ = 0.0018; \*\*\*CD160⁺ versus CD160⁻ = 0.0002. Spleen: \**Total versus CD160⁺ = 0.0023; \*\*Total versus CD160⁻ = 0.0067; \*\*CD160⁺ versus CD160⁻ = 0.002. **g**, Plot showing the quantification of partner-derived RORγt-GFP⁺ cells across different tissues (*n* = 8 mice; four parabiotic pairs).

**h**, Representative flow cytometry plots showing the fraction of resident and exchanging RORγt-GFP⁺ cells in the lung and liver. **i,j**, UMAP representation of *Rorc*⁺ cells identifying five clusters in lung and liver (**i**), showing resident and exchanging cells in different colors (**j**). **k**, Bar plot quantifying exchanging and resident γδ T cells in the lung and liver. **l**, Dot plot of differentially expressed genes in clusters enriched in resident versus exchanging *Rorc*⁺ cells. Color represents the mean expression of the gene in the respective cluster, and dot size represents the fraction of cells in the cluster expressing the gene. **m**, Quantification of partner-derived total, RORγt -GFP⁺, RORγt -GFP⁻ cells in the skin (*n* = 8 mice; four parabiotic pairs). Statistical analysis with Tukey's multiple-comparisons test (ANOVA), displaying the mean ± s.d. (error bars). Significance levels (with 95% confidence interval): \*\**P* < 0.01; \*\*\**P* < 0.001. \*\*Total versus RORγt⁺ = 0.0084; \*\*\*Total versus RORγt⁻ = 0.0004; \*\*RORγt⁺ versus RORγt⁻ = 0.0013. Ax S, axillary lymph nodes suture side; I + B S, inguinal and brachial lymph nodes suture side.

---

## Analogous programs of γδ and T$_{RM}$ tissue residency

Next, we sought to systematically compare the tissue residency features of γδ T cells to those of CD8$^+$ T$_{RM}$ cells. Unlike T$_{RM}$ cells, we did not identify a universal transcriptional program for γδ T cell tissue residency. To bridge this gap, we conducted a supervised analysis using key factors responsible for T$_{RM}$ formation and maintenance to identify the correspondence between the tissue residency features of γδ T cells and T$_{RM}$ cells. T$_{RM}$ cells across many tissues are CD69$^+$CD103$^{+56}$. Furthermore,

human T$_{RM}$ cells express CD49a encoded by *Itga1*[57]. We found various γδ subsets expressing *Cd69*, *Itgae* and *Itga1* (Fig. 8a–d). Gut-resident *Gzmb*$^+$ IELs expressed all three markers, while the liver-resident *Cd160*$^+$ γδ subset expressed *Cd69* and *Itga1* (Fig. 8a–d). The skin-resident *Areg*$^+$ subset expressed mainly *Itgae* (Fig. 8a–d). Circulating *Rorc*$^+$, *Sell*$^+$*Ly6c2*$^-$ and *Sell*$^+$*Ly6c2*$^+$ subsets did not express *Itgae* and *Itga1* and exhibited open chromatin and transcriptional programs associated with circulating memory T cells (Fig. 8d and Extended Data Fig. 10). These

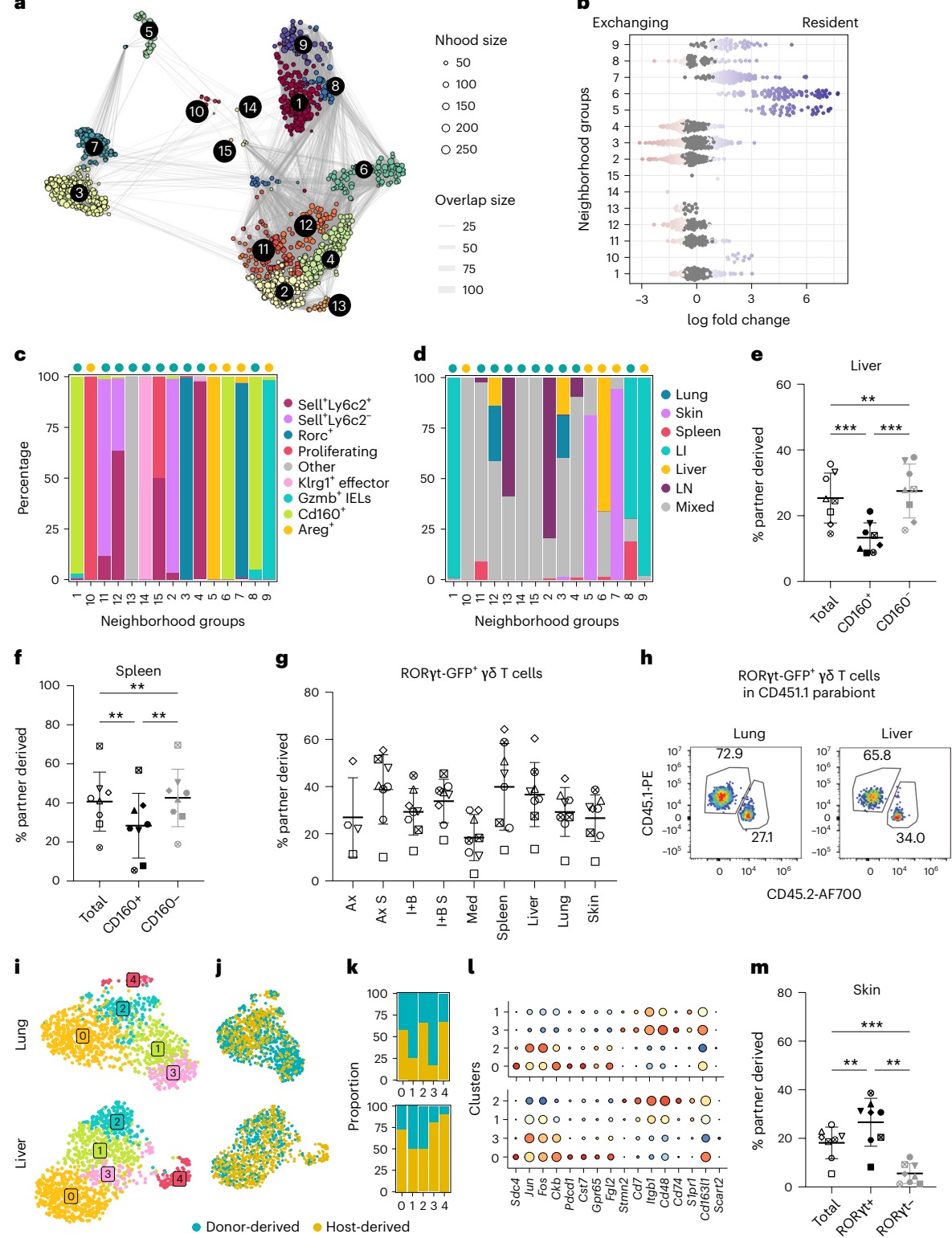

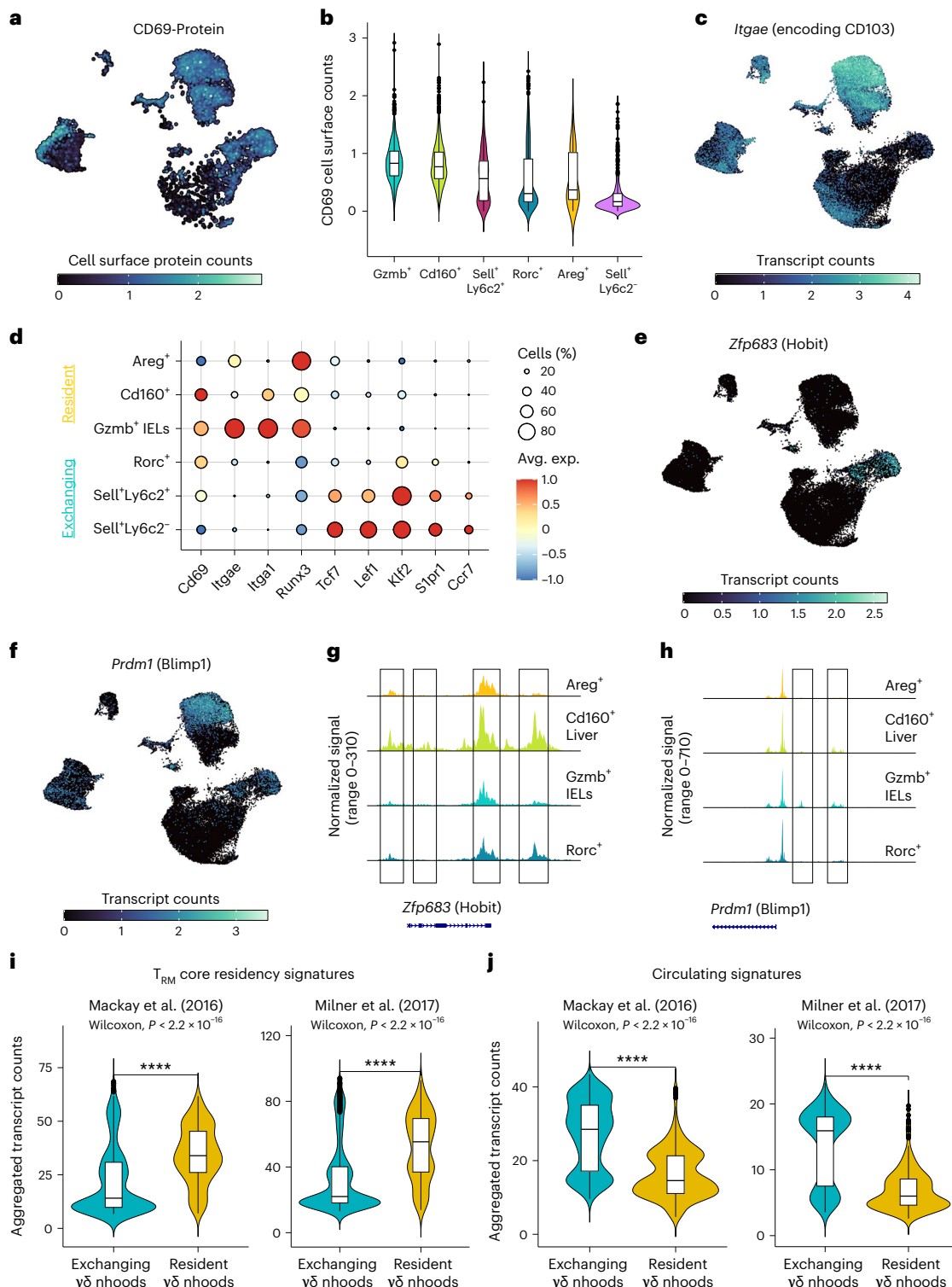

**Fig. 8 | Analogous tissue residency programs of γδ and T_RM cells. a**, UMAP representation showing the normalized cell surface expression of CD69 profiled using TotalSeq. **b**, Violin plots with box plots showing the quantification of CD69 cell surface levels in different γδ subsets (Gzmb⁺: $n = 3,470$; Cd160⁺: $n = 3,777$; Sell⁺Ly6c2⁺: $n = 407$; Rorc⁺: $n = 1,226$; Areg⁺: $n = 38$; Sell⁺Ly6c2⁻: $n = 317$ cells). **c**, UMAP representation showing the normalized transcript counts of *Itgae*. **d**, Dot plot showing the expression of key genes associated with T_RM and circulating memory T cells in various tissue-resident and exchanging γδ subsets. Color represents the mean expression of the gene in the respective cluster, and dot size represents the fraction of cells in the cluster expressing the gene. **e,f**, UMAP representation showing the normalized transcript counts of *Hobit*

(**e**) and *Blimp1* (**f**). **g,h**, Chromatin accessibility tracks showing the frequency of Tn5 integration across regions of the genome encoding *Hobit* (**g**) and *Blimp1* (**h**) for four γδ subsets. **i,j**, Violin plots including box plots showing the aggregated transcript counts of genes presenting T_RM core residency signatures (**i**) and circulating signatures (**j**) from two previous studies[20,58] in neighborhoods enriched in exchanging and resident γδ T cells (exchanging γδ neighborhoods: $n = 984$; resident γδ neighborhoods: $n = 382$). Significance was assessed using a two-tailed Wilcoxon rank-sum test. In **b**, **i** and **j**, the box plots are structured as follows: the central line represents the median, while the upper and lower limits of the boxes correspond to the upper and lower quartiles, respectively. The whiskers extend to 1.5 times the interquartile range (IQR).

results indicate that features of $T_{RM}$ tissue residency are also present in tissue-resident γδ T cells, albeit with some tissue-specific variability. Runx3 has been shown to be a key regulator of $T_{RM}$ differentiation[58]. Importantly, all tissue-resident γδ subsets expressed Runx3, indicating that Runx3 may play a similar role in establishing γδ tissue residency (Fig. 8d). Moreover, Hobit (*Zfp638*) and Blimp1 (*Prdm1*) are also central regulators of various tissue-resident lymphocyte lineages[20]. In γδ subsets, Hobit expression was restricted to the liver-resident *Cd160*⁺ γδ T cells, while Blimp1 expression was mainly identified in intestinal *Gzmb*⁺ γδ IELs (Fig. 8e,f). Few cells in the liver-resident *Cd160*⁺ compartment also expressed Blimp1 (Fig. 8f). Chromatin accessibility assessment of the regions encoding Hobit and Blimp1 further supported these findings (Fig. 8g,h). These results indicate that, unlike in $T_{RM}$ and NKT cells, Hobit and Blimp1 may play tissue-specific roles in establishing γδ tissue residency.

Finally, apart from investigating the role of a few marker genes and TFs in establishing γδ tissue residency, we examined the core gene expression signatures defining $T_{RM}$ residency[20,58] in resident and exchanging γδ neighborhoods (Supplementary Table 6). Our analysis revealed that neighborhoods abundant in tissue-resident γδ T cells across organs exhibited significantly higher expression of $T_{RM}$ core residency genes, while circulatory memory signatures were more abundant in circulating γδ neighborhoods (Fig. 8i,j). These findings indicate that although there is heterogeneity in the factors required for the establishment of tissue residency among γδ subsets, the overall hallmarks are analogous between γδ and $T_{RM}$ cells. Overall, these analyses provide a highly resolved view of transcriptional programs governing γδ T cell tissue residency in comparison to their αβ counterparts.

## Discussion

While several studies have explored the heterogeneity of γδ T cells at single-cell resolution[23,36,55,59–61], a comprehensive cross-organ multimodal study detailing their site-specific adaptation and tissue residency features was still lacking. In this study, we provide a multimodal landscape of γδ T cells across several epithelial tissues and lymphoid organs. We demonstrate that γδ T cells in epithelial tissues are epigenetically and transcriptionally unique subsets that are highly adapted in barrier organs. γδ subsets in secondary lymphoid organs substantially differ from their counterparts residing in barrier tissues.

Previous studies have suggested that DETCs and intestinal IELs share several common features in maintaining epithelial barrier integrity and promoting wound healing and regeneration upon damage and inflammation[4]. We clearly demonstrate that both γδ subsets have open chromatin loci associated with a distinct set of interleukins and growth factors, although most of them are not transcribed at steady state. Therefore, we argue that DETCs and intestinal IELs may exert their regenerative roles through the production of interleukins following immune challenge. Although DETCs express *Areg* and *Il13* as described previously[36], we did not detect the expression of KGF (encoded by *Fgf7*) or other growth factors in these cells. However, chromatin regions encoding several growth factors including *Fgf7* were significantly open in DETCs, indicating that they may be poised to synthesize these growth factors upon activation during injury. Paradoxically, although implicated in tissue repair functions, intestinal IELs exhibit cytotoxic features and display an open *Ifng* locus. Therefore, the mechanisms through which DETCs and intestinal IELs strike a balance between limiting pathogens through cytotoxicity while still fostering pro-repair properties remain to be elucidated[32]. Our data negate the existence of multiple γδ subsets performing these distinct functions and suggest a context-dependent and interleukin-based mechanism, as shown in skin carcinogenesis[62].

Using the parabiosis mouse model, we lay out a highly resolved single-cell map of tissue residency features of γδ T cells across organs. We did not find a common universal transcriptional program associated with γδ tissue residency. While *Itgae* was restricted to skin-resident and gut-resident γδ T cells, *Itga1* and Hobit were predominantly expressed in liver-resident γδ T cells. Liver $T_{RM}$ subsets have also been shown to be CD103⁻ and CD49a⁺[20,63]. Notably, Runx3 was uniformly expressed in all tissue-resident γδ T cells. Although Runx3 is required for the development of DETCs and regulates CD103 expression[64], its role in establishing γδ tissue residency in other organs has not been explored. While single genes and TFs linked to $T_{RM}$ formation displayed tissue-specific regulation in distinct γδ subsets, the core signatures associated with $T_{RM}$, NKT and MAIT cell tissue residency closely resemble those of all tissue-resident γδ T cells, suggesting the existence of a core genome-wide transcriptional program associated with tissue residency across all lymphocyte lineages. Furthermore, the circulatory programs associated with effector and central memory αβ T cells (for example, *Klf2* and *S1pr1*) were strikingly similar to circulating γδ T cells.

Although our study primarily focuses on γδ T cells in mice, it highlights several parallels between γδ T cells in mice and humans. For instance, in both mice and humans, tissue-resident γδ T cells in the liver are characterized by the expression of CD49a and CD69[65]. Furthermore, in both species they exhibit the expression of *Gzmb* and *Blimp1*, and demonstrate restricted TCR diversity[65,66]. In the intestine, γδ IELs in both mice and humans primarily express *Itgae*, *Gzma* and *Gzmb*[67]. Moreover, in humans, the peripheral blood contains a subset of naïve γδ T cells that express *TCF7* and *LEF1*, and these cells exhibit a diverse TCR repertoire similar to Sell⁺Ly6c⁻ naïve cells observed in mice[68]. A thorough single-cell multimodal profiling of human γδ T cells across diverse organs, combined with the findings in this study, can yield valuable insights into γδ T cell biology across species and clues in utilizing them to enhance protective immunity in diseases.

## Online content

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

## Methods

### Mice

Experiments were performed using mice from two different animal facilities—Max Planck Institute of Immunobiology and Epigenetics in Freiburg (Germany) and Curie Institute in Paris (France). Parabiosis experiments were performed in Paris. C57BL/6J mice in Freiburg were obtained from in-house breeding and were kept in the animal facility of the Max Planck Institute of Immunobiology and Epigenetics in specific-pathogen-free conditions with a 12-h light/12-h dark cycle, a temperature range of 20–23 °C and 60% humidity. All animal experiments were performed in accordance with the relevant guidelines and regulations and approved by the review committee of the Max Planck Institute of Immunobiology and Epigenetics and the Regierungspräsidium Freiburg, Germany. For the experiments performed in Paris, CD45.1/1 and CD45.1/2 animals were generated in-house by crossing CD45.1/1 B6 animals with CD45.2/2 RORγt-GFP B6-MAIT$^{CAST}$ mice. All experiments were conducted in an accredited animal facility by the French Veterinarian Department following ethical guidelines approved by the ethics committee of the Institut Curie CEEA-IC (Authorization APAFIS no. 24245–2020021921558370-v1 given by National Authority) in compliance with the international guidelines. Mice were housed in a specific-pathogen-free facility at the Curie Institute with a 12-h light/12-h dark cycle, a temperature range of 22–24 °C and 70% humidity.

### Cross-tissue single-cell preparation

All animals were euthanized using carbon dioxide or cervical dislocation. All organs were collected fresh (that is, right after euthanasia) in $CO_2$-independent medium (Gibco) and maintained on ice until processing.

**Spleen and lymph nodes.** To isolate cells from the spleen and lymph nodes, tissues were dissected and placed on a 40-µm cell strainer (Falcon, Corning) kept on a 50-ml tube (Falcon, Corning) and were mashed on the cell strainer using the back of the 1-ml syringe plunger. Ten milliliters of PBS containing 0.5% BSA and 2 mM EDTA was continuously added while mashing to collect the single-cell suspension in a 50-ml tube. Collected cells were centrifuged at 400g for 5 min at 4 °C. The pellet was resuspended in 10 ml PBS and passed through a 30-µm nylon filter (CellTrics, Sysmex) kept in a 15-ml tube (Falcon, Corning). Cells were again centrifuged at 400g for 5 min at 4 °C. Afterwards, the pellet was resuspended in 100 µl of PBS containing 0.5% BSA and 2 mM EDTA for subsequent FACS staining. Red blood cell lysis was performed for splenic samples using red blood cell lysis buffer (10×, BioLegend) according to the manufacturer's protocol.

**Skin.** Skin single-cell suspensions were obtained as previously described[69]. Briefly, dorsal skin tissue was dissected (flattened, epidermis side up) and incubated at 37 °C for 45 min in 1 ml of 500 CU Dispase (Corning). The tissue was then chopped in RPMI 1640 GlutaMAX media supplemented with 1 mM sodium pyruvate, 1 mM nonessential amino acids, 50 µM β-mercaptoethanol, 20 mM HEPES, 100 U ml$^{-1}$ penicillin, 100 mg ml$^{-1}$ streptomycin, 0.5 mg ml$^{-1}$ DNase I (all products from Sigma-Aldrich) and 0.25 mg ml$^{-1}$ Liberase TL (Roche) and incubated for 1 h 45 min at 37 °C in a 5% $CO_2$ incubator. After filtering on a 40-µm filter kept in a 50-ml tube, the cells were washed twice in PBS containing 0.5% BSA and 2 mM EDTA, and the cell suspension was removed of skin debris using cell debris removal solution (Miltenyi) following the manufacturer's instructions.

**Liver and lung.** To ensure complete lung and liver perfusion (evidenced by organ color change caused by the loss of red blood cells), a 20-ml syringe with a 22-gauge needle was used to inject 1× PBS starting with the right ventricle of the heart (10 ml) followed by the hepatic portal vein (10 ml). After perfusion and dissection of the liver and lung, the tissues were finely minced and digested using collagenase D (0.7 mg ml in PBS) for 30 min at 37 °C on a shaker in Freiburg, while in Paris, the Gentlemacs operating system (Miltenyi) with the m_impTumor_01 program was used. After washing the cell pellet twice at 400g for 5 min, the pellet was resuspended in 8 ml of 44% Percoll density gradient solution and underlaid with 5 ml of 67% Percoll density gradient solution. Centrifugation (without breaks) was performed at 1,600g for 20 min at room temperature. The cell layer containing mononuclear cells at the interface of the 44% and 67% density gradient centrifugation media was removed, transferred and washed. The resulting pellet was resuspended in 100 µl staining buffer (PBS containing 0.5% BSA and 2 mM EDTA).

**Intestinal IELs.** To isolate IELs from the large and small intestines, tissues were dissected, cleaned to remove feces, cut open and chopped into 2-cm pieces. The pieces were treated with 1 mM 1,4-Dithioerythritol to release IELs (2×, 20 min each at 37 °C, constant shaking). The supernatant was filtered through 70-µm cell strainers (Falcon, Corning) kept in a 50-ml tube (Falcon, Corning) on ice. Cells were washed with PBS containing 0.5% BSA and 2 mM EDTA, and 44% and 67% density gradient centrifugation was performed as described above. After washing, the resulting pellet was resuspended in 100 µl staining buffer (PBS containing 0.5% BSA and 2 mM EDTA).

### Antibody staining, flow cytometry and single-cell sorting

One hundred microliters of antibody staining solution was prepared in PBS containing 0.5% BSA and 2 mM EDTA and added to the isolated cells resuspended in 100 µl staining buffer as described above. Cells were incubated for 20 min on ice, washed three times with 2 ml of 0.5% BSA in PBS and resuspended in 3 ml after the last wash for cell sorting. The following antibodies were used: TCRγδ-APC (BioLegend, 1:100 dilution), TCRβ-BV421 (BioLegend, 1:100 dilution), CD45.1-PE (BD Biosciences, 1:100 dilution), CD45.2-AF700 (BioLegend, 1:100 dilution), CD45.2-PerCP5.5 (BD Biosciences, 1:100 dilution), CD160-PECy7 (BioLegend, 1:100 dilution), Ly6C-BV510 (BioLegend, 1:100 dilution) and CD62L-BV421 (BioLegend, 1:100 dilution). Zombie Aqua and Zombie Green fixable viability kits (BioLegend) were used to distinguish dead and living cells. Living TCRγδ$^+$ single γδ T cells were sorted in BSA-coated tubes containing 50 µl of PBS using a FACSAria cell sorter (BD Biosciences) equipped with BD FACSDiva software (v8.0.2). Using pulse geometry gates (FSC-W × FSC-H and SSC-W × SSC-H), doublets/multiplets were excluded. After the completion of sorting, the cells were processed through the different 10x Genomics workflows.

### Single-cell RNA sequencing

Single-cell RNA sequencing was performed using 10x Genomics with feature barcoding technology to multiplex cell samples from different organs so that they could be loaded on one well to reduce costs and minimize technical variability. Hashtag oligonucleotides were obtained as purified and already oligo-conjugated in TotalSeq-B (3′ chemistry) and TotalSeq-C (5′ chemistry) formats from BioLegend. Cells were stained with barcoded antibodies together with the staining solution before FACS sorting as described above. The antibody concentrations used were 1 µg per million cells, as recommended by the manufacturer (BioLegend) for flow cytometry applications. After staining, cells were washed three times in PBS containing 2% BSA and 0.01% Tween 20, followed by centrifugation (300g for 5 min at 4 °C) and supernatant exchange. After the final wash, the cells were resuspended in PBS, filtered through 40-µm cell strainers and processed for sorting. Sorted γδ T cells were processed through the 10x Genomics single-cell 3′ or V(D)J workflow according to the manufacturer's instructions. Libraries were pooled to desired quantities to obtain appropriate sequencing depths as recommended by 10x Genomics and sequenced on a NovaSeq 6000 flow cell.

## Single-cell surface protein profiling

To profile γδ T cells using antibodies for the quantification of cell surface proteins on a single-cell level, the following antibodies were obtained as purified, oligo-conjugated TotalSeq-B reagents from BioLegend: CCR6, NKp46, CD117, KLRG1, CCR7, CD8a, CD5, CD122 (IL-2Rβ), CD127 (IL-7Rα), CD278 (ICOS), Ly-6A/E (Sca-1), CD69, CD44, CD27, CD24, CD62L, CD25, NK-1.1, CD4, CCR2, TCR Vγ2, TCRβ, CD11c, CD19, GR-1 and CSF1R. Cells were stained with barcoded antibodies together with the staining solution before FACS sorting as described above. The antibody concentrations used were 1 μg per million cells, as recommended by the manufacturer (BioLegend) for flow cytometry applications. Sorted γδ T cells were processed through the 10x Genomics 3′ workflow according to the manufacturer's instructions. Libraries were pooled in the desired ratio together with the gene expression libraries to obtain appropriate sequencing depths as recommended by 10x Genomics and sequenced on a NovaSeq 6000 flow cell.

## Single-cell simultaneous chromatin accessibility and gene expression profiling

To perform simultaneous measurement of chromatin accessibility and gene expression, we could not barcode different organs with hashtag oligonucleotides; hence, after sorting γδ T cells from different tissues, we pooled them to obtain enough cells (80,000–100,000) to perform nuclear extraction according to the 10x Genomics protocol. Thereafter, single nuclei were processed through the Chromium Single Cell Multiome ATAC + Gene Expression workflow according to the manufacturer's instructions. Gene expression and chromatin accessibility libraries were sequenced to obtain appropriate sequencing depths as recommended by 10x Genomics using the Illumina NovaSeq 6000 system.

## Single-cell simultaneous gene expression and TCR profiling

Simultaneous single-cell RNA and TCR sequencing was performed using 10x Genomics V(D)J workflow with feature barcoding technology to multiplex cell samples from different organs. Hashtag oligonucleotides were obtained as purified and already oligo-conjugated in TotalSeq-C (5′ chemistry) format from BioLegend. Single-cell suspensions from three female mice (aged 8 weeks) for each organ (spleen, liver, lung, lymph node and small intestine) were pooled together before sorting. Cells were stained with barcoded antibodies together with the staining solution before FACS sorting as described above. The antibody concentrations used were 1 μg per million cells, as recommended by the manufacturer (BioLegend) for flow cytometry applications. After staining, cells were washed three times in PBS containing 2% BSA and 0.01% Tween 20, followed by centrifugation (300g for 5 min at 4 °C) and supernatant exchange. After the final wash, the cells were resuspended in PBS, filtered through 40-μm cell strainers and processed for sorting. Single-cell TCR libraries were generated using the following primers[70]. First PCR: 2 μM of forward primer (5′- GATCTACACTCTTTCCCTACACGACGC-3′) and 0.5 μM of each reverse primer (5′-TCGAATCTCCATACTGACCAAGCTTGAC-3′, 5′-GTCTTCAGCGTATCCCCTTCCTGG-3′, 5′-CTTTCAGGCACAGTAAGCC AGC-3′ and 5′-TCTTCAGTCACCGTCAGCCAACTAA-3′). Second PCR: 1 μM of forward primer (5′-GATCTACACTCTTTCCCTACACGAC GC-3′) and 1 μM of each reverse primer (5′-CCACAATCTTCTTGGATGAT CTGAGACT-3′ and 5′-GTCCCAGTCTTATGGAGATTTGTTTCAGC-3′). Libraries were pooled to desired quantities to obtain appropriate sequencing depths as recommended by 10x Genomics and sequenced on a NovaSeq 6000 flow cell.

## Parabiosis experiments

To evaluate the recirculatory and residential properties of γδ T cells, in accordance with published methods[71], congenically distinct (CD45.1/1 or CD45.1/2 and CD45.2/2) aged-matched mice were surgically joined at their olecranon's and knee joints using a non-absorbable 3-0 suture followed by suturing the skin of both animals together using 5-0 absorbable Vicryl sutures. Five weeks after surgery, animals were euthanized, and organs were collected for subsequent FACS or flow cytometry analysis. Staining was performed with the relevant antibodies in staining buffer containing PBS supplemented with 0.5% BSA, 2 mM EDTA and anti-FcR 2.4G2 (Institut Curie, produced in-house, 0.25 μg per million cells) for 20 min at 4 °C. Flow cytometry acquisition was performed using a Cytoflex (Beckman) cytometer with CytExpert software v2.4. Data were analyzed using FlowJo software (v10.8.0) and GraphPad Prism v8. For cell sorting, organs from different mice were processed separately as described above. For each organ, two pools were obtained by regrouping single-cell suspensions of parabionts with identical congenic markers. These organ pools were then stained with different hashtag oligonucleotides following the manufacturer's instructions (BioLegend) and regrouped, which resulted in two tubes containing all organs for each congenic marker. FACS was then performed using a FACSAria cell sorter (BD) equipped with BD FACSDiva software v6. Resident or circulating γδ T cells were sorted in two distinct BSA-coated tubes. Sorted cells were processed through the 10x Genomics 3′ workflow according to the manufacturer's instructions. Libraries were pooled in the desired ratio together to obtain appropriate sequencing depths as recommended by 10x Genomics and sequenced on a NovaSeq 6000 flow cell.

## Quantification of gene expression, protein abundance, chromatin accessibility and TCR counts

Quantification of gene expression and/or cell surface protein abundance counts was performed using either cellranger-4.0.0 or cellranger-6.0.0 using the count command, which performs alignment, filtering, barcode counting and UMI counting as well as process feature barcoding data. Simultaneous counting of transcripts and open chromatin regions was performed through cellranger-arc-2.0.1 using the count command, which performs alignment, filtering, barcode counting, peak calling and counting of both ATAC and RNA molecules. Alignments were performed using prebuilt Cell Ranger and Cell Ranger ARC mouse mm10 references. Simultaneous quantification of gene expression, hashtag abundance and TCR counts/repertoire were performed using the multi command of cellranger-7.1.0.

## Computational analysis of single-cell gene expression data

We analyzed scRNA-seq data using the R package (v4.1.3 and v4.2.2) Seurat (v4.3.0). We combined five batches of single-cell/single-nucleus gene expression datasets (including cells obtained from parabiotic pairs) and made two observations: there was technical variability that needed to be removed, and we identified various small contaminating clusters of B cells and myeloid cells. To remove batch effects, we integrated the data using Harmony[25], an algorithm that uses joint embedding to group cells by cell type rather than dataset-specific conditions. Harmony was executed using the RunHarmony function in Seurat with group.by.vars set to each batch. Furthermore, the clusters containing B cells and myeloid cells were removed. Low-quality cells were removed using the parameters described in Extended Data Fig. 1c. Importantly, ribosomal genes (small and large subunits) as well as predicted genes with *Gm*-identifier were excluded from the analysis. The normalization method was set to 'LogNormalize'. Dimensionality reduction was performed using the RunUMAP function, where reduction was set to 'harmony' and dims to 1:30. Default resolution was used for clustering. To characterize the clusters, differential gene expression analysis was performed using the FindMarkers function in Seurat.

## TotalSeq analysis

Of the five batches, the TotalSeq experiment was performed on one batch. We used the same UMAP coordinates obtained using dimensionality reduction based on gene expression data to visualize the expression of TotalSeq antibodies. Normalization was performed using Seurat with the normalization method set to 'CLR', which performs a centered log ratio transformation for normalization.

## Single-cell chromatin accessibility analysis

scATAC-seq data analysis was performed using Signac (v1.8.0). Experiments to simultaneously measure chromatin accessibility and gene experiments were performed in two batches from mice belonging to two different animal facilities in Germany and France, which led to technical variability in the datasets. Two-step filtering was applied to scATAC-seq data. Only those nuclei that were also included in scRNA-seq analysis were included in the chromatin accessibility analysis. In the second step, scATAC-seq-based filtering was applied using the parameters described in Extended Data Fig. 5b. Furthermore, common peaks identified in all batches were included in the analysis. Integration was again performed using Harmony with the RunHarmony function. The 'group.by.vars' option was set to each batch, and reduction was set to 'lsi'. Dimensionality reduction was performed using the RunUMAP function, in which reduction was set to 'harmony'. For clustering of cells, the resolution in the FindClusters function was set to 0.6. To characterize the clusters, differentially accessible peaks were obtained using the FindMarkers function with test.use = 'LR'. TF motif analysis was performed using the RunChromVAR function in Signac.

## Computational analysis of simultaneous single-cell gene expression and TCR data

We analyzed scRNA-seq data using Seurat and single-cell TCR data using scRepertoire (v1.7.0). Clusters containing B cells and myeloid cells were removed. Low-quality cells were removed using the parameters described in Extended Data Fig. 1c. As previously described, ribosomal genes (small and large subunits) as well as predicted genes with *Gm*-identifier were excluded from the analysis. The normalization method was set to 'LogNormalize'. Dimensionality reduction was performed using the RunUMAP function, where reduction was set to 'pca' and dims to 1:30. Default resolution was used for clustering. To characterize the clusters, differential gene expression analysis was performed using the FindMarkers function in Seurat. In total, 13,753 cells were included in the scRNA-seq analysis. Of 13,753 cells, clonotypes were detected in 4,220 cells. While performing the TCR repertoire analysis using scRepertoire, cloneCall parameter was set to 'strict', which uses the V(D)JC genes comprising the TCR plus the nucleotide sequence of the CDR3 region to call the clonotypes. Both $\gamma$ and $\delta$ chains were used for the clonotype analysis wherever detected. Shannon diversity score was calculated based on nucleotide sequences of the TRG repertoire. Apart from the new clonotypes, we identified all the clonotypes of $\gamma\delta$ T cells previously summarized[72], except for those exhibiting the TRAV15-1-DV6-1 usage found in NKT-like IFN-$\gamma$/IL-4 double producers (Supplementary Data 1d,e and Supplementary Note).

## Neighborhood analysis

To assess the differential contribution of exchanging and resident $\gamma\delta$ T cells across organs and cell states, we applied the Milo algorithm[54] (using the R package miloR 1.2.0), which models cell states as overlapping neighborhoods based on a KNN graph as the basis for abundance testing. The KNN graph was built using the buildGraph function with $k = 30$ and $d = 30$, and neighborhoods were defined using the makeNhoods function with prop = 0.1, $k = 30$, $d = 30$ and refined = TRUE. Neighborhoods were grouped using groupNhoods with max.lfc. delta = 10. Neighborhoods with a log$_2$ fold change > 1 were considered resident, while the rest were denoted as circulating.

## Reporting summary

Further information on research design is available in the Nature Portfolio Reporting Summary linked to this article.

## Data availability

The primary read files and the raw counts for all single-cell sequencing datasets reported in this paper are available to download from the Gene Expression Omnibus under accession number https://www.ncbi.nlm.nih.gov/geo/query/acc.cgi?acc=GSE222454. Processed data can be downloaded from https://github.com/sagar161286/multimodal_gdTcells/.

## Code availability

Codes to reproduce the data analysis and figures are available at https://github.com/sagar161286/multimodal_gdTcells/.

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

## Acknowledgements

We thank D. Grün, T. Boehm and A. Nusser for critical reading of the paper. We thank J. Bodinek-Wersing from the Lighthouse Core Facility at the Medical Faculty, University of Freiburg for the support with cell sorting. Sagar is supported by the Department of Medicine II, Freiburg University Medical Center, Faculty of Medicine, University of Freiburg. This study was supported by the Research Commission from the Faculty of Medicine of the University of Freiburg (Project ID 3095050051 to Sagar), Deutsche Forschungsgemeinschaft (DFG, German Research Foundation) through SFB 1160 (Project ID 256073931 to Sagar), TRR 359 (Project ID 491676693 to Sagar) and 322977937/GRK2344 (to Sagar), Agence Nationale de la Recherche Grant MAITrepair ANR-20-CE15-0028-01 (to O.L.), European Research Council (ERC–2019-AdG-885435; to O.L.) and Fondation pour la Recherche Médicale, grant number FDT202106013036 (to A.H.)

## Author contributions

A.H. performed parabiosis experiments, tissue preparation for cell sorting and flow cytometry analysis and contributed to single-cell experiments as well as data interpretation and paper writing; K.B. prepared gene expression, chromatin accessibility, cell surface protein and TCR libraries; N.P. contributed to mouse handling and single-cell preparation from tissues; A.D. contributed to parabiosis experiments and single-cell preparation from tissues; D.O. and M.S. contributed to data analysis, R.T. and M.H. provided key scientific inputs and contributed to data interpretation and paper writing; O.L. supervised A.H. and A.D., provided key scientific inputs and contributed to data interpretation and paper writing; S. conceptualized and coordinated the study, acquired funding, conducted experiments, analyzed the data and wrote the paper with the help of A.H. and O.L. All authors read and edited the paper.

## Competing interests

The authors declare no competing interests.

## Additional information

**Extended data** is available for this paper at https://doi.org/10.1038/s41590-023-01710-y.

**Correspondence and requests for materials** should be addressed to Sagar.

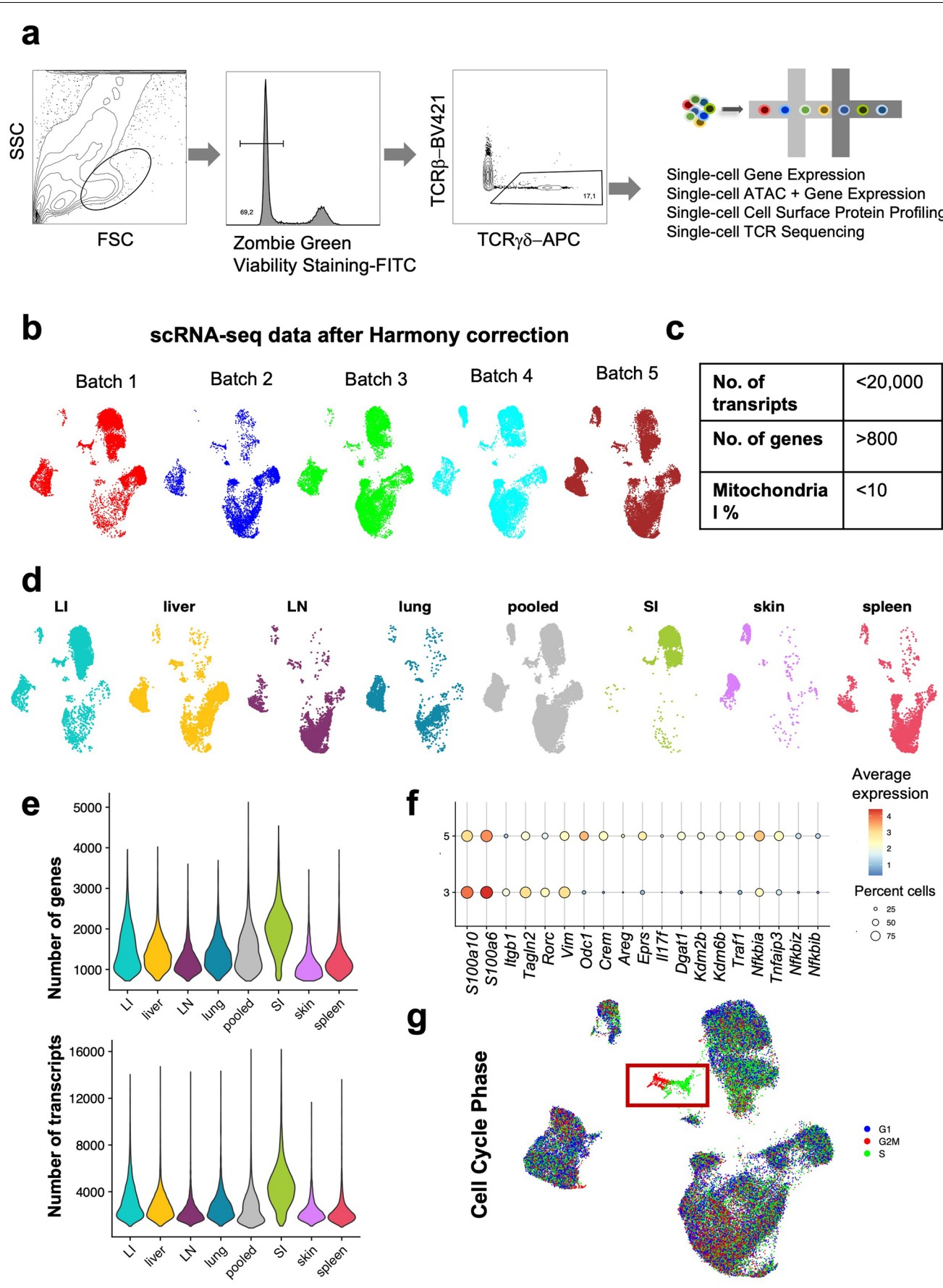

**Extended Data Fig. 1 | See next page for caption.**

**Extended Data Fig. 1 | Quality control analysis of the scRNA-seq data.**
**a**, Representative flow cytometry plots showing our strategy to sort γδ T cells for single-cell multiomics experiments using 10x Genomics. **b**, UMAP representation showing the origin of cells across different batches in different colors after batch correction was performed using Harmony. Note that the experiments were performed in five batches using mice from two different animal facilities, leading to batch-associated variability. **c**, Table listing the cut-offs used to remove low-quality cells for scRNA-seq analysis. **d**, UMAP representation showing the origin of cells across different organs in different colors. **e**, Violin plots showing the number of genes and transcripts quantified per cell from each tissue in the scRNA-seq data. **f**, Dot plot showing the expression of selected genes differentially expressed between $Rorc^+$ clusters 3 and 5. Color represents the mean expression of the gene in the respective cluster, and dot size represents the fraction of cells in that cluster expressing the gene. **g**, UMAP representation depicting the cell cycle phase of each cell. Cells in S, $G_2$ and M phases are highlighted using a dark red rectangle.

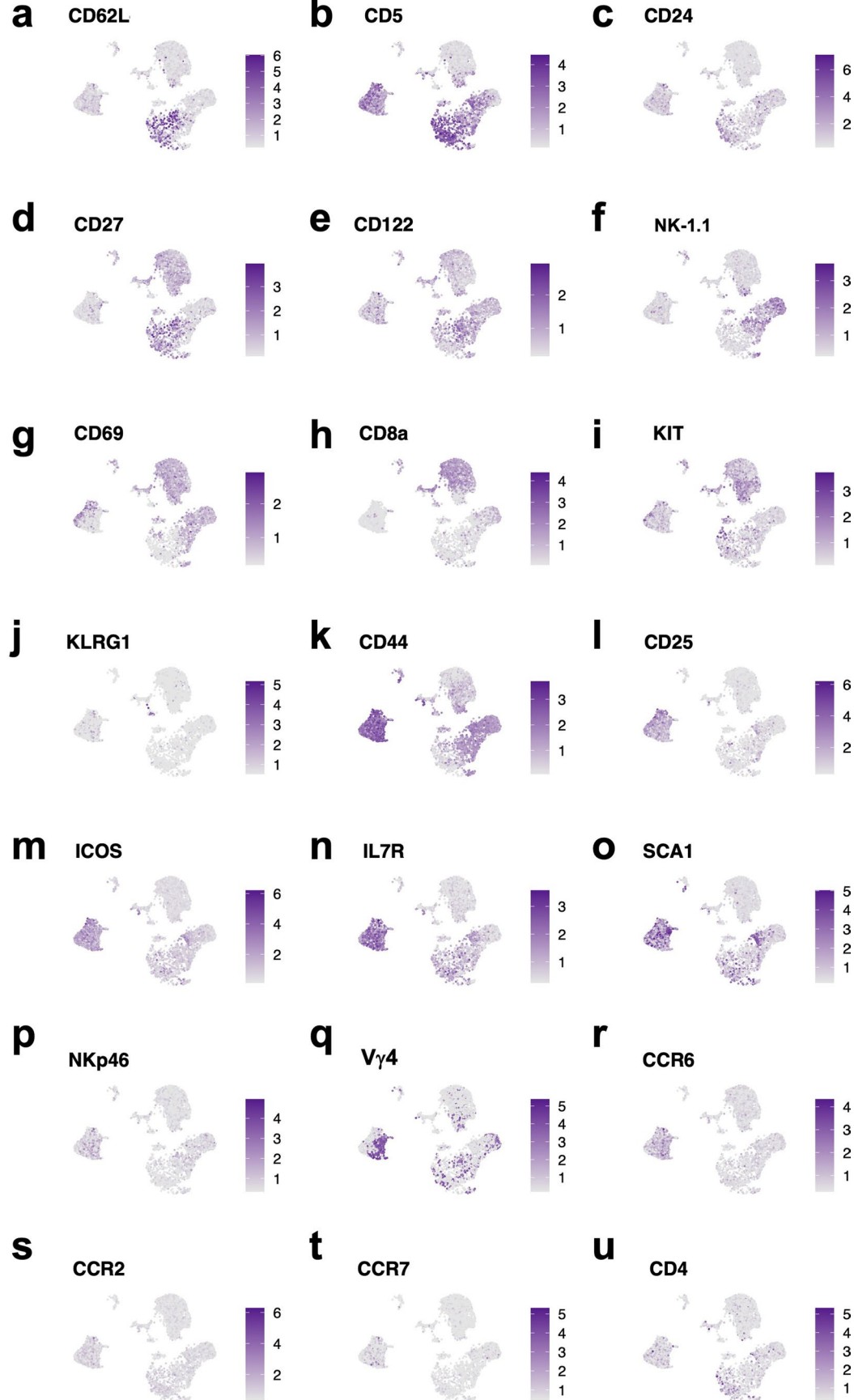

**Extended Data Fig. 2 | Quantification of cell surface protein markers. a-u**, UMAP representation showing the normalized expression of 21 cell surface markers profiled using TotalSeq™.

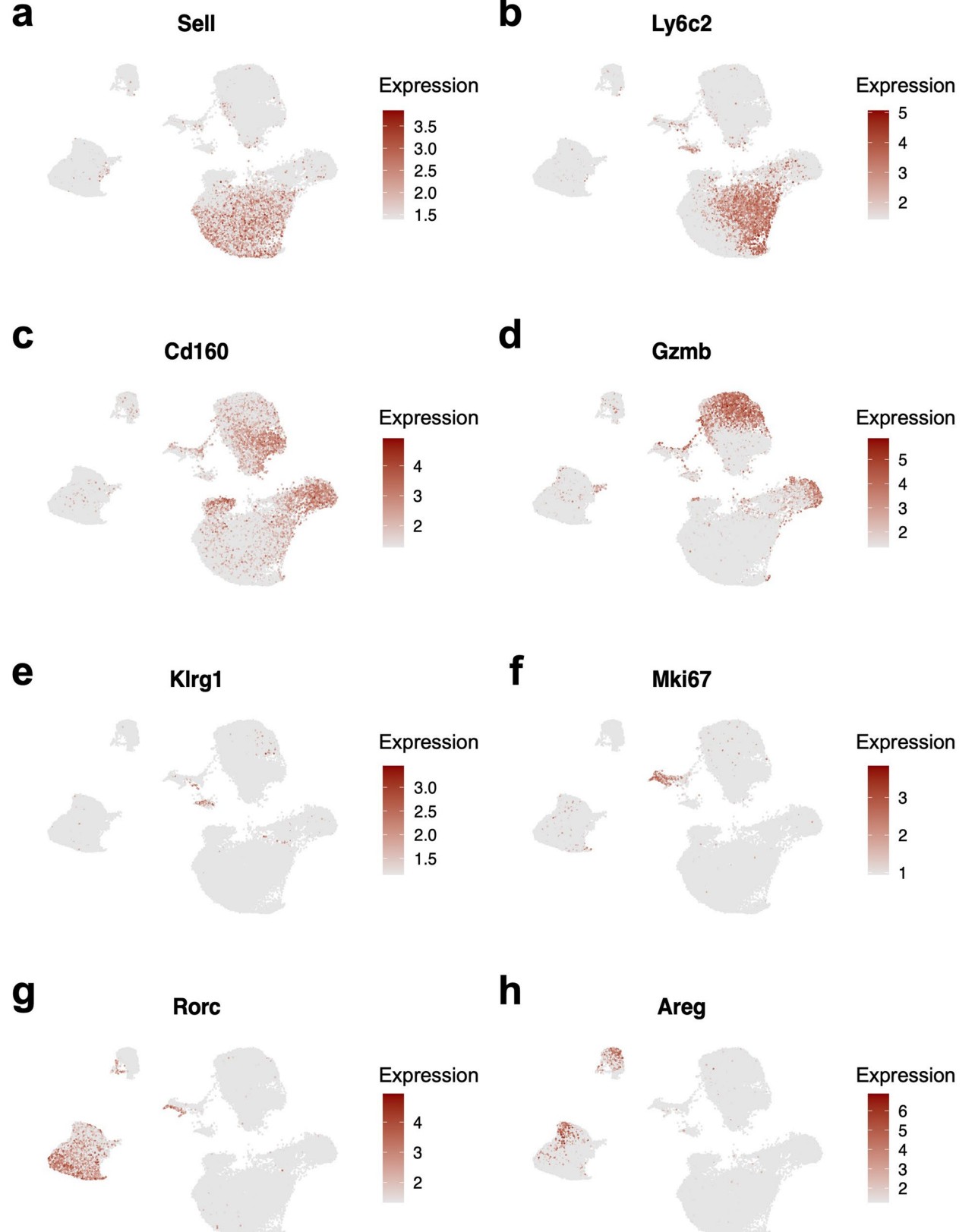

**Extended Data Fig. 3 | Genes identifying various γδ subsets. a-h**, UMAP representation showing the normalized transcript counts of *Sell*, *Ly6c2*, *Cd160*, *Gzmb*, *Klrg1*, *Mki67*, *Rorc* and *Areg* used to classify γδ T cell subsets based on scRNA-Seq.

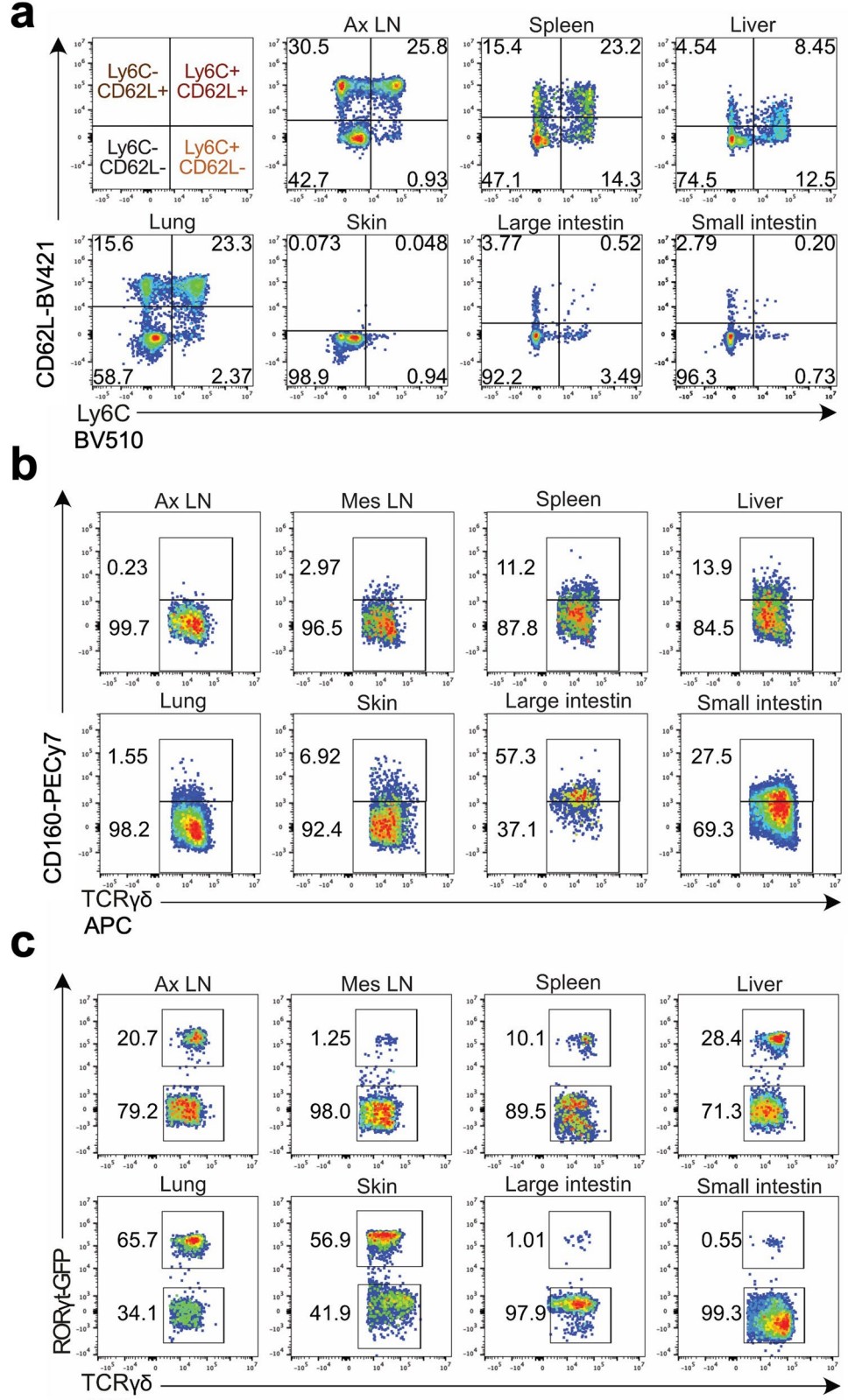

**Extended Data Fig. 4 | Characterizing γδ T cell subsets using flow cytometry. a-c**, Representative flow cytometry plots showing the fraction of γδ T cells stratified using CD62L and LY6C (**a**), CD160 (**b**) and RORγt-GFP (**c**) across different organs. Quantification of the data is shown in Fig. 3g–i. Ax LN, axillary lymph nodes; Mes LN, mesenteric lymph nodes.

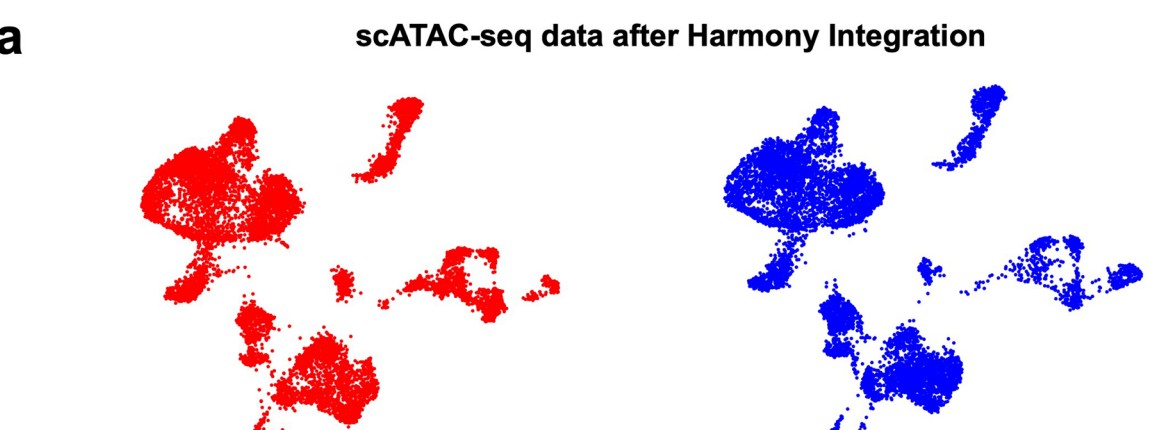

**a** scATAC-seq data after Harmony Integration

Batch 1          Batch 2

**b**

| No. of ATAC counts | Nucleosome Signal | TSS enrichment |
|---|---|---|
| 1000-100000 | <2 | >1 |

**c**

**d** Number of counts    TSS enrichment    Nucleosome signal

**e** scRNA-seq clusters on scATAC-seq UMAP

**f** Cd24a

**g** Ifng

**Extended Data Fig. 5 | See next page for caption.**

**Extended Data Fig. 5 | Quality control analysis of the scATAC⁻seq data.**
**a**, UMAP representation showing the origin of cells across two batches in different colors after batch correction was performed using Harmony. **b**, Table listing the cut-offs applied to remove low-quality cells for scATAC-seq analysis. **c**, Plot showing the distribution of the TSS enrichment score and Tn5 insertion frequency at TSS sites in the scATAC-Seq data. **d**, Plots showing the total number of chromatin accessibility counts, TSS distribution and nucleosome signal of the cells included in the analysis after filtering. **e**, UMAP representation based on chromatin accessibility showing the 22 clusters identified based on gene expression profiles shown in Fig. 1c in different colors. **f, g**, Chromatin accessibility tracks showing the frequency of Tn5 integration across regions of the genome encoding *Cd24a* (**f**) and *Ifng* (**g**) for cells grouped by scATAC-seq clusters.

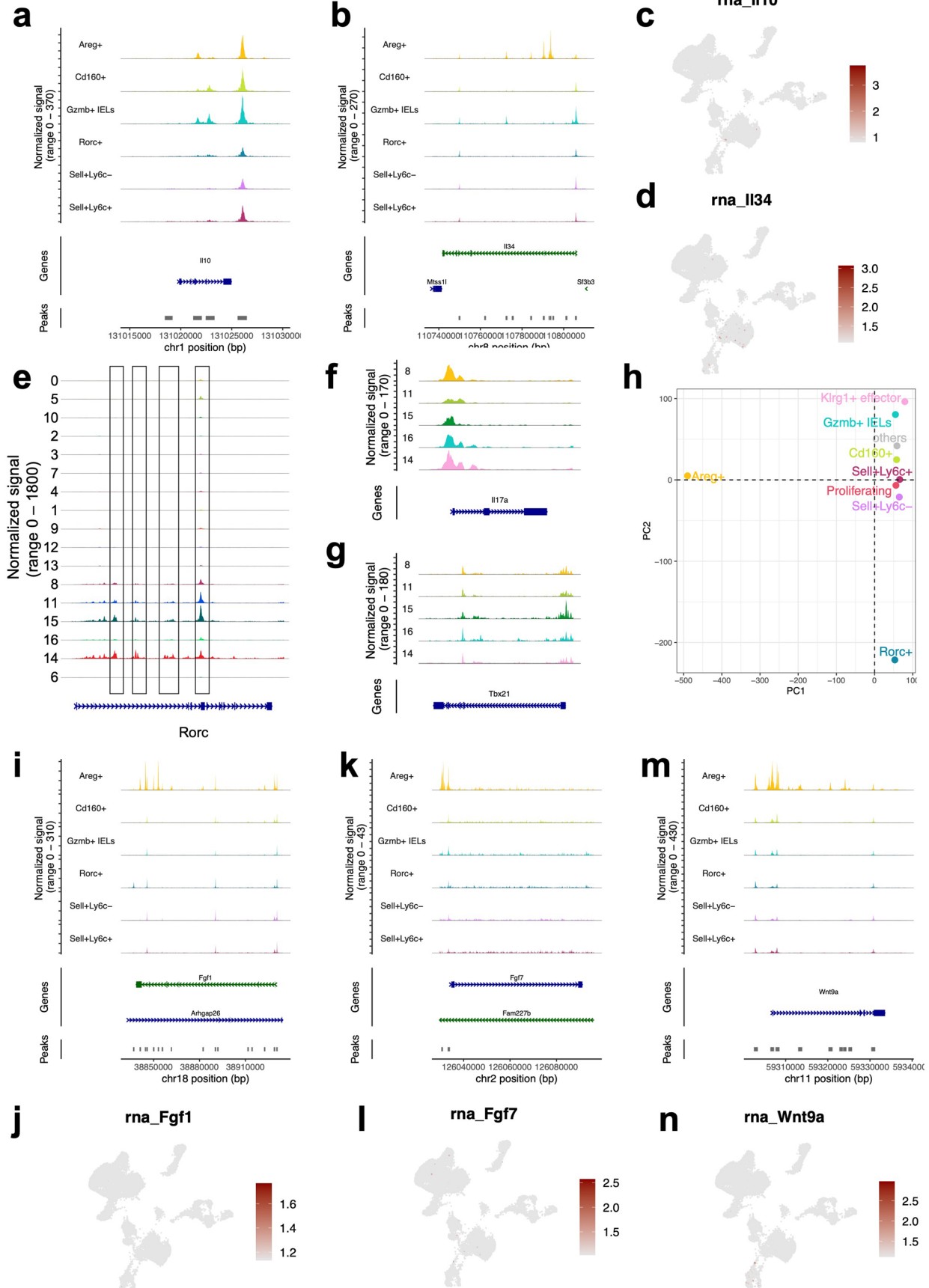

**Extended Data Fig. 6 | See next page for caption.**

**Extended Data Fig. 6 | Simultaneous gene expression and chromatin accessibility profiling reveals novel features of γδ subsets. a, b**, Chromatin accessibility tracks showing the frequency of Tn5 integration across regions of the genome encoding *Il10* (**a**) and *Il34* (**b**) for shortlisted γδ subsets. **c, d**, UMAP representation of scATAC-seq data showing the normalized transcript counts of *Il10* (**c**) *and Il34* (**d**). **e-g**, Chromatin accessibility tracks showing the frequency of Tn5 integration across regions of the genome encoding *Rorc* (**e**), *Il17a* (**f**) and *Tbx21* (**g**) for shortlisted γδ scATAC-seq clusters. **h**, PCA based on the average highly variable chromatin accessibility profiles of eight γδ T subsets revealed that *Areg*⁺ and *Rorc*⁺ subsets exhibit distinct transcriptional signatures compared to other subsets. **i**, Chromatin accessibility tracks showing the frequency of Tn5 integration across regions of the genome encoding *Fgf1* for shortlisted γδ subsets. **j**, UMAP representation of scATAC-seq data showing the normalized transcript counts of *Fgf1*. **k**, Chromatin accessibility tracks showing the frequency of Tn5 integration across regions of the genome encoding *Fgf7* for shortlisted γδ subsets. **l**, UMAP representation of scATAC-seq data showing the normalized transcript counts of *Fgf7*. **m**, Chromatin accessibility tracks showing the frequency of Tn5 integration across regions of the genome encoding *Wnt9a* for shortlisted γδ subsets. **n**, UMAP representation of scATAC-seq data showing the normalized transcript counts of *Wnt9a*.

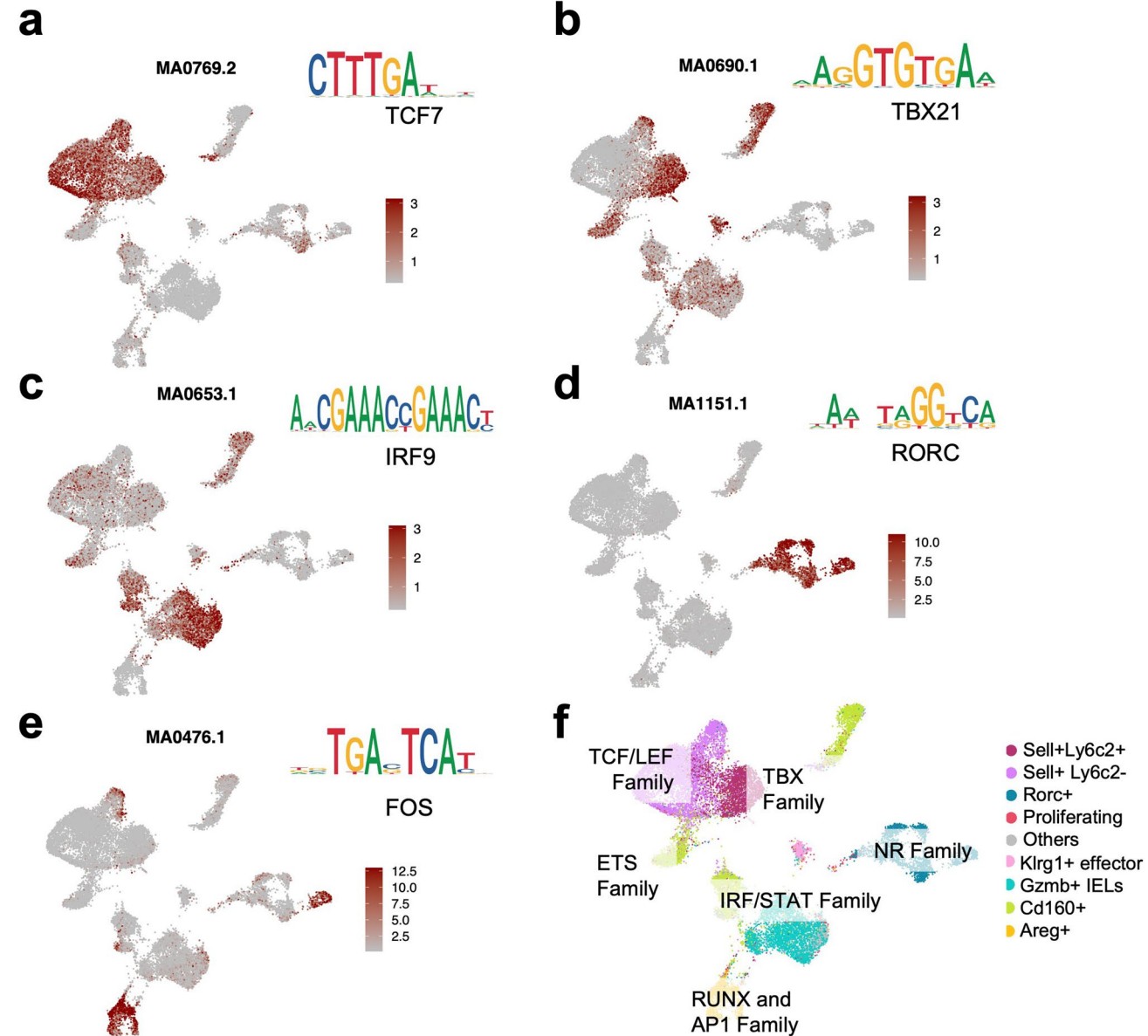

**Extended Data Fig. 7 | Transcription factor families specific to γδ T cell subsets. a-e**, DNA sequence motifs and UMAP representation showing the motif activity score per cell for key overrepresented TF motifs in the scATAC-seq data – *Tcf7* (**a**), *Tbx21* (**b**), *Irf9* (**c**), *Rorc* (**d**) and *Fos* (**e**). **f**, UMAP representation showing γδ subsets in different colors overlaid with TF family motifs enriched in each cell subset as identified by chromVAR analysis.

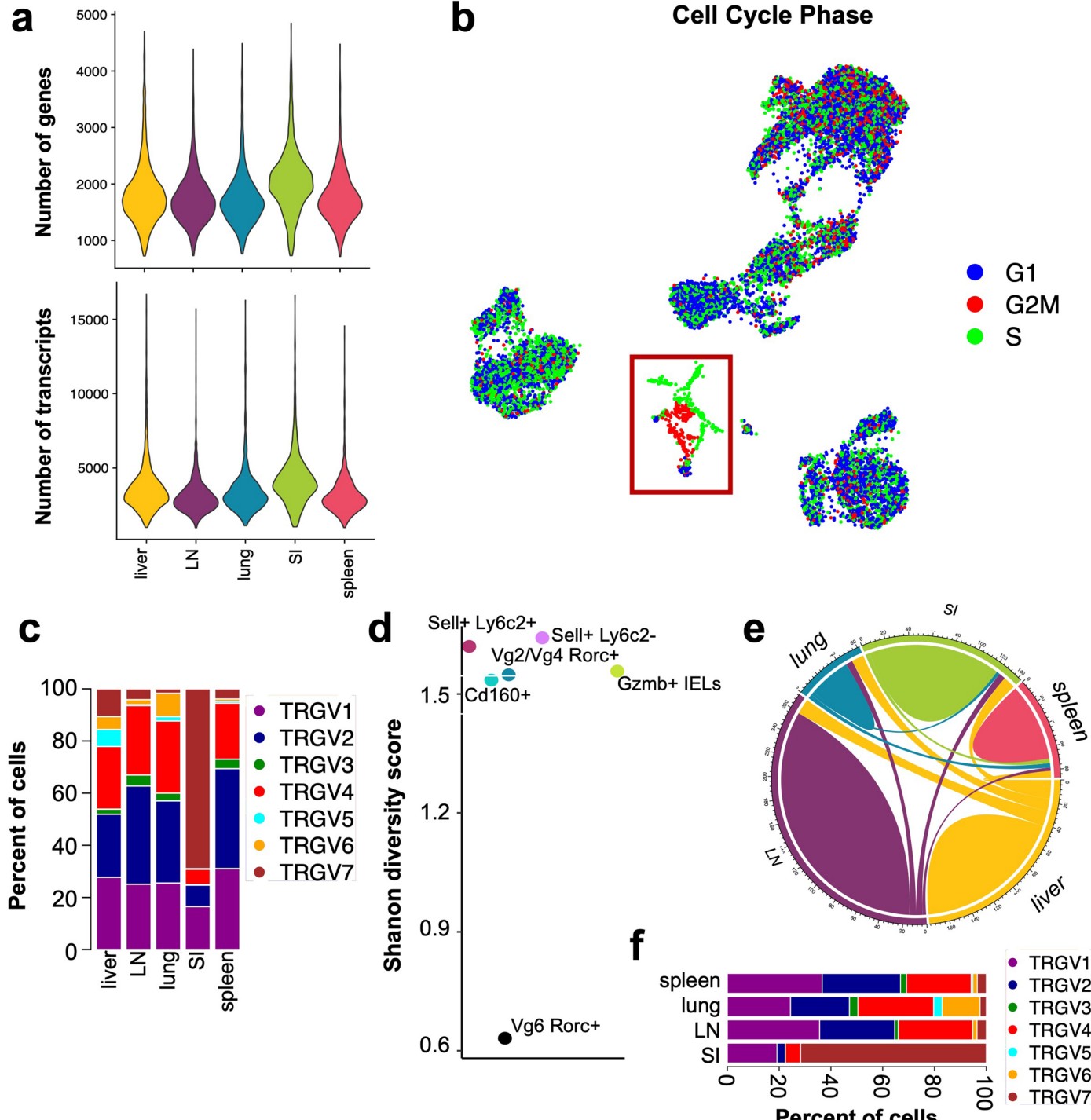

**Extended Data Fig. 8 | TCR repertoire analysis of γδ T cells across tissues.**
**a**, Violin plots showing the number of genes and transcripts quantified per cell
from each tissue in the data where cells were simultaneously profiled for gene
expression and TCR rearrangement configurations. **b**, UMAP representation
depicting the cell cycle phase of each cell. Cells in S, $G_2$ and M phases are
highlighted using a dark red rectangle. **c**, Bar plot quantifying the variable γ
chain usage of γδ T cells within each tissue. Seven variable γ chains are depicted

in different colors. **d**, Dot plot showing the Shannon diversity score calculated
based on the diversity of TCR γδ clonotypes in different cell types. Dots in
different colors represent different cell types. **e**, Chord diagram depicting
the clonal overlap between profiled organs. **f**, Bar plot quantifying the shared
variable γ chain usage of liver γδ T cells with other organs. Seven variable γ chains
are depicted in different colors. SI, small intestine; LN, lymph nodes.

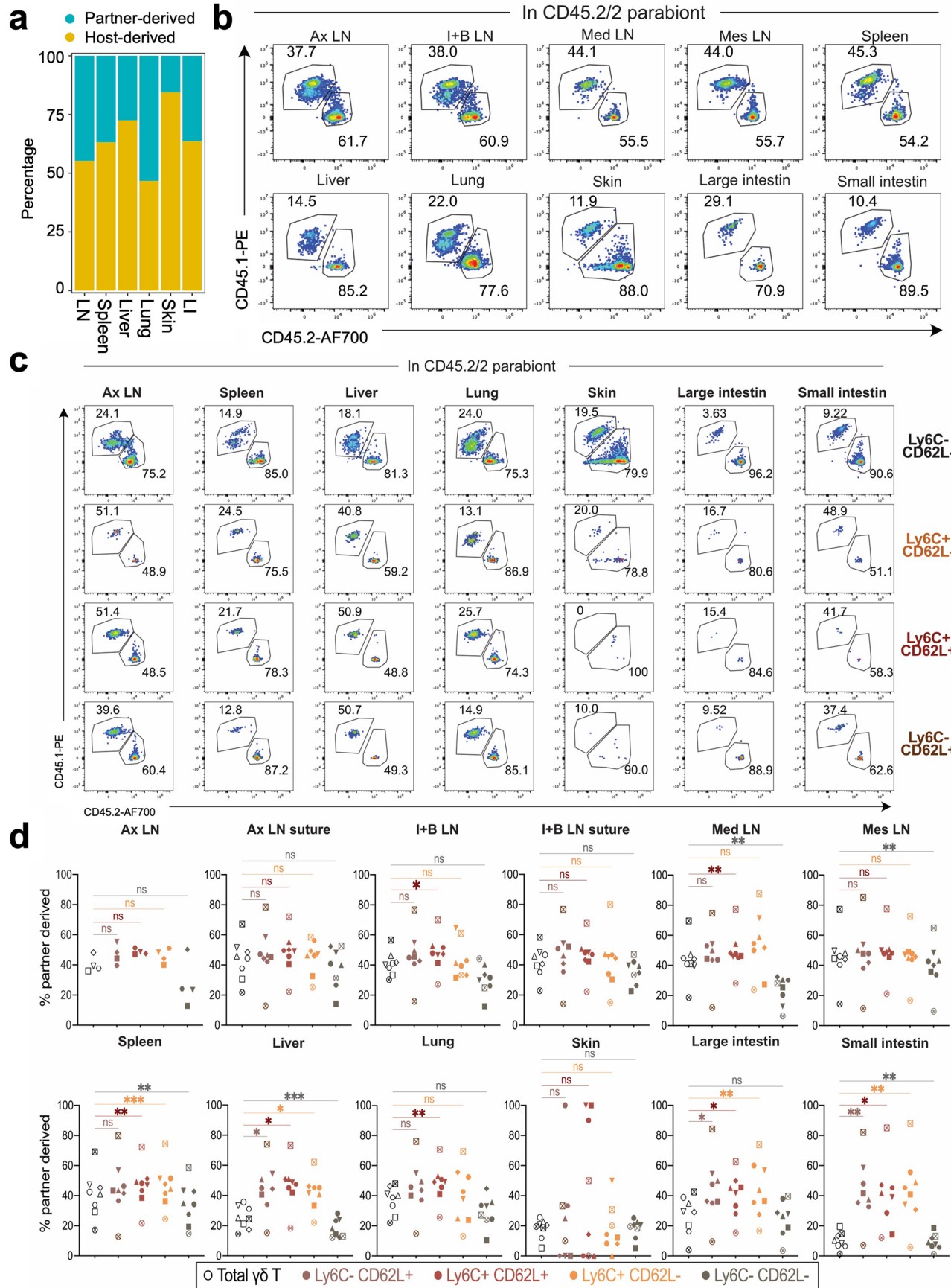

**Extended Data Fig. 9 | See next page for caption.**

**Extended Data Fig. 9 | Quantification of exchanging and resident γδ T cells across tissues. a**, Bar plot showing the quantification of partner- and host-derived γδ T cells across different tissues using scRNA-seq. **b**, Representative FACS plots showing the percentage of exchanging and resident γδ T cells across different tissues. Quantification of the data is shown in Fig. 6f. **c**, Representative FACS plots showing the percentages of exchanging and resident CD62L$^+$LY6C$^-$, CD62L$^+$LY6C$^+$, CD62L$^-$LY6C$^+$ and CD62L$^-$LY6C$^-$ γδ T cells across different tissues. **d**, Plots showing the quantification of partner-derived total, CD62L$^+$LY6C$^-$, CD62L$^+$LY6C$^+$, CD62L$^-$LY6C$^+$ and CD62L$^-$LY6C$^-$ γδ T cells across different

tissues (n = 8 mice; four parabiotic pairs). Statistical analysis using Dunnett's multiple comparisons test (One-way ANOVA), with $P$ values: *$P$ < 0.05; **$P$ < 0.01; ***$P$ < 0.001 and ****$P$ < 0.0001 and ns>0.05 (95% CI). With exact $P$ values for panel d. as follows: I + B LN (*red = 0.0195), MedLN (**red = 0.0019; **grey = 0.0024), MesLN (**grey = 0.0063), Spleen (**red = 0.0055; ***orange = 0.0006; **grey = 0.0027), Liver (*pink = 0.0308; *red = 0.0110; *orange = 0.0103; ***grey = 0.0004), Lung (**red = 0.0054), Large intestine (*pink = 0.0138; *red = 0.0162; **orange = 0.0081), Small intestine (**pink = 0.0061; *red = 0.0226; **orange = 0.0091; **grey = 0.0070).

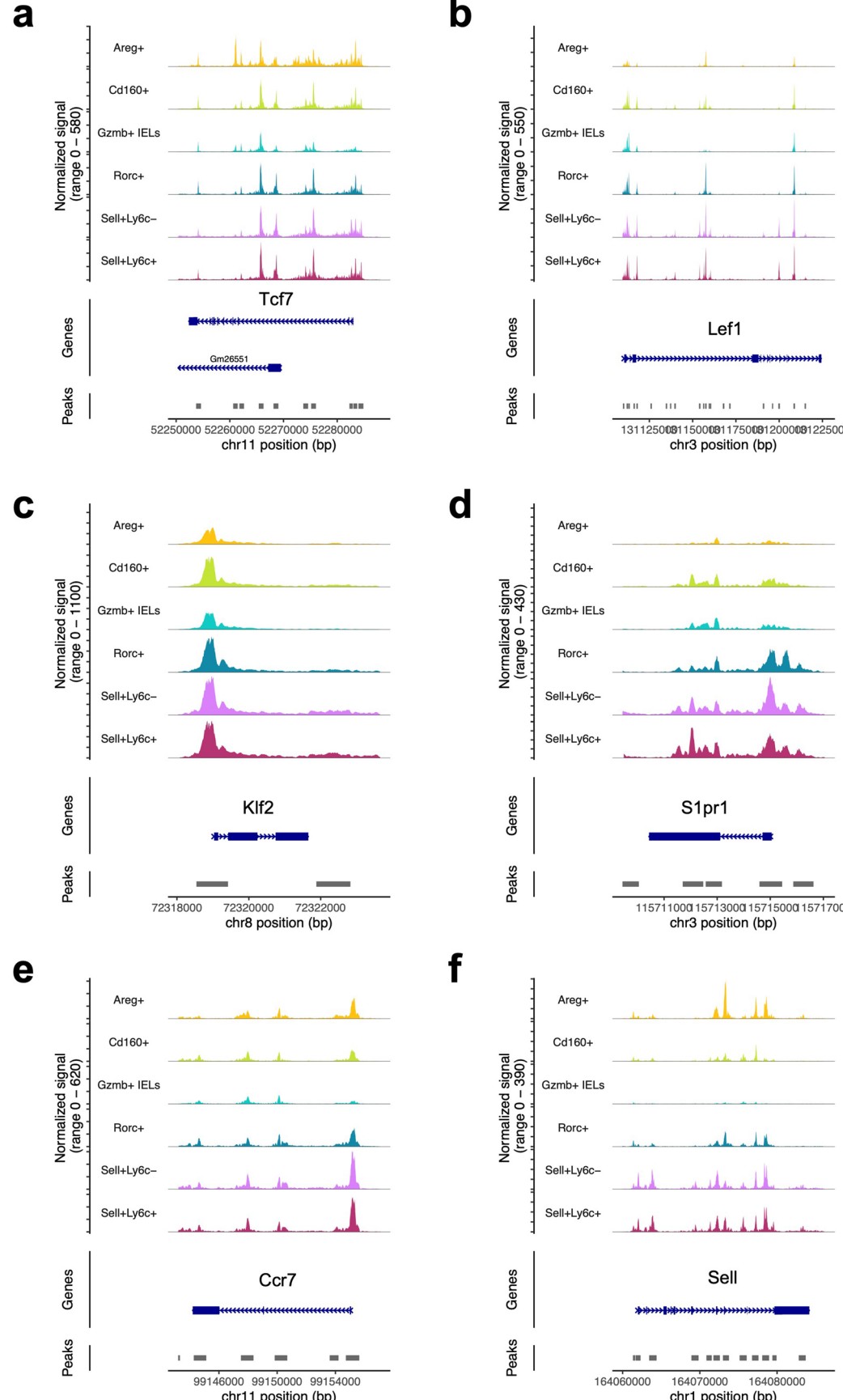

**Extended Data Fig. 10 | Chromatin landscape of genes associated with circulating memory T cells in γδ subsets. a-f,** Chromatin accessibility tracks showing the frequency of Tn5 integration across regions of the genome encoding *Tcf7* (**a**), *Lef1* (**b**), *Klf2* (**c**), *S1pr1* (**d**), *Ccr7* (**e**) and *Sell* (**f**) for six γδ subsets.

# Reporting Summary

## Statistics

For all statistical analyses, confirm that the following items are present in the figure legend, table legend, main text, or Methods section.

| n/a | Confirmed | |
|---|---|---|
| ☐ | ☒ | The exact sample size (*n*) for each experimental group/condition, given as a discrete number and unit of measurement |
| ☐ | ☒ | A statement on whether measurements were taken from distinct samples or whether the same sample was measured repeatedly |
| ☐ | ☒ | The statistical test(s) used AND whether they are one- or two-sided *Only common tests should be described solely by name; describe more complex techniques in the Methods section.* |
| ☒ | ☐ | A description of all covariates tested |
| ☐ | ☒ | A description of any assumptions or corrections, such as tests of normality and adjustment for multiple comparisons |
| ☐ | ☒ | A full description of the statistical parameters including central tendency (e.g. means) or other basic estimates (e.g. regression coefficient) AND variation (e.g. standard deviation) or associated estimates of uncertainty (e.g. confidence intervals) |
| ☐ | ☒ | For null hypothesis testing, the test statistic (e.g. *F*, *t*, *r*) with confidence intervals, effect sizes, degrees of freedom and *P* value noted *Give P values as exact values whenever suitable.* |
| ☒ | ☐ | For Bayesian analysis, information on the choice of priors and Markov chain Monte Carlo settings |
| ☒ | ☐ | For hierarchical and complex designs, identification of the appropriate level for tests and full reporting of outcomes |
| ☒ | ☐ | Estimates of effect sizes (e.g. Cohen's *d*, Pearson's *r*), indicating how they were calculated |

*Our web collection on statistics for biologists contains articles on many of the points above.*

## Software and code

Policy information about availability of computer code

| | |
|---|---|
| Data collection | BD FACSDiva software v6 (Paris) and v8.0.2 (Freiburg) for sorting of cells for single-cell experiments and Cytoflex software (CytExpert) v2.4 for flow cytometry to collect the data for parabiosis FACS validation experiments. |
| Data analysis | Single-cell data alignment: cellranger-4.0.0, cellranger-6.0.0, cellranger-arc-2.0.1 and cellranger-7.1.0. <br> Single-cell data analysis: R 4.1.3, R 4.2.2, Seurat 4.3.0, Signac 1.8.0, miloR 1.2.0 and scRepertoire 1.7.0 <br> FACS data analysis: FlowJo v10.8.0 <br> Data representation: Prism v8 and custom codes <br> Codes to reproduce the data analysis and figures are available at: https://github.com/sagar161286/multimodal_gdTcells |

For manuscripts utilizing custom algorithms or software that are central to the research but not yet described in published literature, software must be made available to editors and reviewers. We strongly encourage code deposition in a community repository (e.g. GitHub). See the Nature Portfolio guidelines for submitting code & software for further information.

## Data

Policy information about availability of data

All manuscripts must include a data availability statement. This statement should provide the following information, where applicable:
- Accession codes, unique identifiers, or web links for publicly available datasets
- A description of any restrictions on data availability
- For clinical datasets or third party data, please ensure that the statement adheres to our policy

The primary read files and the raw counts for all single-cell sequencing datasets reported in this paper are available to download from GEO (accession number: GSE222454, https://www.ncbi.nlm.nih.gov/geo/query/acc.cgi?acc=GSE222454). Processed data can be downloaded from https://github.com/sagar161286/multimodal_gdTcells

## Research involving human participants, their data, or biological material

Policy information about studies with human participants or human data. See also policy information about sex, gender (identity/presentation), and sexual orientation and race, ethnicity and racism.

| | |
|---|---|
| Reporting on sex and gender | n/a |
| Reporting on race, ethnicity, or other socially relevant groupings | n/a |
| Population characteristics | n/a |
| Recruitment | n/a |
| Ethics oversight | n/a |

Note that full information on the approval of the study protocol must also be provided in the manuscript.

# Field-specific reporting

Please select the one below that is the best fit for your research. If you are not sure, read the appropriate sections before making your selection.

☒ Life sciences     ☐ Behavioural & social sciences     ☐ Ecological, evolutionary & environmental sciences

For a reference copy of the document with all sections, see nature.com/documents/nr-reporting-summary-flat.pdf

# Life sciences study design

All studies must disclose on these points even when the disclosure is negative.

| | |
|---|---|
| Sample size | Sample size was mainly determined by accessing the reproducibility of the experiments. To profile gd T cells for single cell experiments, minimum two independent experiments each in Freiburg and Paris were performed. For each independent experiment, 3 mice were used. The data from different independent experiments resulted in the identification of the same clusters/cell types after batch correction reflecting reproducibility. Three independent experiments were performed for single-cell RNA-Sequencing measurements and two independent experiments were performed for simultaneous measurements of RNA and chromatin accessibility at single-cell resolution. Both experiments were performed at two different animal facilities in two different countries (Freiburg, Germany and Paris, Fance). Parabiosis experiments were performed in two independent batches and in each batch 4 pairs of mice (8 in total) were analyzed for tissue resident and exchanging gd T cells. All experiments were highly reproducible. |
| Data exclusions | While performing single cell analysis, we identified cells of B cell and myeloid cell lineages forming separate and distinct clusters in the dataset. Cells forming these clusters were excluded from the analysis. |
| Replication | In order to ensure reproducibility, each experiment was performed at least twice independently. After batch correction, we identified all cell types recovered from different single cell experiments. Moreover, we performed such experiments at two different animal facilities to ensure that same cell subsets are present in the mice house in both the facilities. Parabiosis experiments were always performed with 4 mice pairs (8 in total) in each experiment and in single cell experiments, 2-3 mice were always pooled. All replication efforts were successful and the data was highly reproducible. |
| Randomization | Given age-matched and female mice, mice were randomly assigned to the single-cell and parabiosis experiments. |
| Blinding | Age-matched and only female mice were used for the experiment. No blinding was applicable as there was no treatment invloved in the experiments. |

# Reporting for specific materials, systems and methods

We require information from authors about some types of materials, experimental systems and methods used in many studies. Here, indicate whether each material, system or method listed is relevant to your study. If you are not sure if a list item applies to your research, read the appropriate section before selecting a response.

## Materials & experimental systems

| n/a | Involved in the study |
|---|---|
| ☐ | ☒ Antibodies |
| ☒ | ☐ Eukaryotic cell lines |
| ☒ | ☐ Palaeontology and archaeology |
| ☐ | ☒ Animals and other organisms |
| ☒ | ☐ Clinical data |
| ☒ | ☐ Dual use research of concern |
| ☒ | ☐ Plants |

## Methods

| n/a | Involved in the study |
|---|---|
| ☒ | ☐ ChIP-seq |
| ☐ | ☒ Flow cytometry |
| ☒ | ☐ MRI-based neuroimaging |

## Antibodies

| | |
|---|---|
| Antibodies used | The following antibodies were used at a 1:100 dilution-<br>Brilliant Violet 421™ anti-mouse TCR β chain Antibody, 109229, BioLegend.<br>APC Anti-mouse TCR γ/δ Antibody, 118115, BioLegend.<br>Brilliant Violet 421™ anti-mouse CD62L Antibody, 104435, BioLegend.<br>PE/Cyanine7 anti-mouse CD160 Antibody, 143009, BioLegend.<br>Alexa Fluor® 700 anti-mouse CD45.2 Antibody, 109821, BioLegend.<br>Brilliant Violet 510™ anti-mouse Ly-6C Antibod, 128033, BioLegend.<br>PerCP-Cy™5.5 Mouse Anti-Mouse CD45.2, 552950, BD.<br>PE Mouse Anti-Mouse CD45.1, 561872, BD.<br><br>The following antibodies were used at a concentration of 1μg per million cells-<br>TotalSeq™-B 0001 anti-mouse CD4, 100573, BioLegend.<br>TotalSeq™-B 0002 anti-mouse CD8a, 100783, BioLegend.<br>TotalSeq™-B 0012 anti-mouse CD117 (c-kit), 105849, BioLegend.<br>TotalSeq™-B 0073 anti-mouse/human CD44, 103071, BioLegend.<br>TotalSeq™-B 0093 anti-mouse CD19, 115563, BioLegend.<br>TotalSeq™-B 0097 anti-mouse CD25, 102067, BioLegend.<br>TotalSeq™-B 0105 anti-mouse CD115 (CSF-1R), 135543, BioLegend.<br>TotalSeq™-B 0106 anti-mouse CD11c, 117359, BioLegend.<br>TotalSeq™-B 0111 anti-mouse CD5, 100645, BioLegend.<br>TotalSeq™-B 0112 anti-mouse CD62L, 104465, BioLegend.<br>TotalSeq™-B 0116 anti-mouse Ly-6G/Ly-6C (Gr-1), 108465, BioLegend.<br>TotalSeq™-B 0118 anti-mouse NK-1.1, 108763, BioLegend.<br>TotalSeq™-B 0120 anti-mouse TCR β chain, 109261, BioLegend.<br>TotalSeq™-B 0130 anti-mouse Ly-6A/E (Sca-1), 108149, BioLegend.<br>TotalSeq™-B 0171 anti-human/mouse/rat CD278 (ICOS), 313551, BioLegend.<br>TotalSeq™-B 0184 anti-mouse CD335 (NKp46), 137641, BioLegend.<br>TotalSeq™-B 0191 anti-mouse/rat/human CD27, 124247, BioLegend.<br>TotalSeq™-B 0197 anti-mouse CD69, 104555, BioLegend.<br>TotalSeq™-B 0198 anti-mouse CD127 (IL-7Rα), 135055, BioLegend.<br>TotalSeq™-B 0211 anti-mouse TCR Vγ2, 137715, BioLegend.<br>TotalSeq™-B 0212 anti-mouse CD24 M1/69, 101847, BioLegend.<br>TotalSeq™-B 0225 anti-mouse CD196 (CCR6), 129827, BioLegend.<br>TotalSeq™-B 0227 anti-mouse CD122 (IL-2Rβ), 105913, BioLegend.<br>TotalSeq™-B 0250 anti-mouse/human KLRG1 (MAFA), 138435, BioLegend.<br>TotalSeq™-B 0377 anti-mouse CD197 (CCR7), 120133, BioLegend.<br>TotalSeq™-B 0426 anti-mouse CD192 (CCR2), 150633, BioLegend.<br><br>anti–FcR 2.4G2 (Institut Curie, produced in house, 0.25 μg per million cells) |
| Validation | All validations are performed by the manufacturer from where the antibodies were bought. BioLegend mentions that each lot of the antibody is quality control tested by immunofluorescent staining with flow cytometric analysis. For flow cytometric staining, the suggested use of is ≤ 0.5 μg per million cells in 100 μl volume. BD states that all flow cytometry reagents are titrated on the relevant positive or negative cells. To save time and cell samples for researchers, test size reagents are bottled at an optimal concentration with the best signal-to-noise ratio on relevant models during the product development. To ensure consistent performance from lot-to-lot, each reagent is bottled to match the previous lot MFI. |

# Animals and other research organisms

Policy information about studies involving animals; ARRIVE guidelines recommended for reporting animal research, and Sex and Gender in Research

| | |
|---|---|
| Laboratory animals | Experiments were performed using mice from two different animal facilities - Max Planck Institute of Immunobiology and Epigenetics in Freiburg (Germany) and Curie Institute in Paris (France). Parabiosis experiments were performed in Paris. C57BL/6J mice in Freiburg were obtained from in-house breeding and were kept in the animal facility of the Max Planck Institute of Immunobiology and Epigenetics in specific-pathogen-free (SPF) conditions with a 12h light /12h dark cycle, a temperature ranging between 20-23°C and 60% humidity. For the experiments performed in Paris, CD45.1/1 and CD45.1/2 animals were generated in-house by crossing CD45.1/1 B6 animals to CD45.2/2 RORɣt-GFP B6-MAITCAST mice. At the beginning of the experiments, all mice were 6-10 weeks old and were housed in an SPF facility at the Curie Institute with a 12h light /12h dark cycle, a temperature ranging between 22-24°C and 70% humidity. |
| Wild animals | No wild animals were used. |
| Reporting on sex | In order to minimize technical variabilities and batch effects, only females were used in all the experiments. Furthermore, females were chosen to avoid the heightened risk of dominance and injury (as we were collecting the skin and this would be a very strong limitation for using males). And consequently, to avoid the sex effect all subsequent mice used in this study were females. |
| Field-collected samples | This study did not utilize field-collected samples. |
| Ethics oversight | All animal experiments in Freiburg were performed in accordance with the relevant guidelines and regulations, approved by the review committee of the Max Planck Institute of Immunobiology and Epigenetics and the Regierungspräsidium Freiburg, Germany. For the experiments performed in Paris, All experiments were conducted in an accredited animal facility by the French Veterinarian Department following ethical guidelines approved by the ethics committee of the Institut Curie CEEA-IC (Authorization APAF1S no. 24245–2020021921558370-v1 given by National Authority) in compliance with the international guidelines. |

Note that full information on the approval of the study protocol must also be provided in the manuscript.

# Flow Cytometry

## Plots

Confirm that:

☒ The axis labels state the marker and fluorochrome used (e.g. CD4-FITC).

☒ The axis scales are clearly visible. Include numbers along axes only for bottom left plot of group (a 'group' is an analysis of identical markers).

☒ All plots are contour plots with outliers or pseudocolor plots.

☒ A numerical value for number of cells or percentage (with statistics) is provided.

## Methodology

| | |
|---|---|
| Sample preparation | Cross-tissue single cell preparation<br>All animals were sacrificed using carbon dioxide or cervical dislocation. All organs were collected fresh (i.e., right after sacrifice) in CO2 independent medium (Gibco) and maintained on ice until processing.<br>Spleen and lymph nodes. To isolate cells from the spleen and lymph nodes, tissues were dissected and placed on a 40-μm cell strainer (Falcon, Corning) kept on a 50-ml tube (Falcon, Corning) and were mashed on the cell strainer using the back of the 1-ml syringe plunger. 10 ml phosphate-buffered saline (PBS) containing 0.5% BSA and 2mM EDTA was continuously added while mashing to collect the single-cell suspension in the 50-ml tube. Collected cells were centrifuged at 400 g for 5 min at 4°C. The pellet was resuspended in 10 ml PBS and passed through the 30-μm nylon filter (CellTrics, Sysmex) kept on a 15-ml tube (Falcon, Corning). Cells were again centrifuged at 400 g for 5 min at 4°C. Afterwards the pellet was resuspended in 100 μl of PBS containing 0.5% BSA and 2mM EDTA for subsequent fluorescence-activated cell sorting (FACS) staining. Red blood cell lysis was performed for splenic samples using red blood cell lysis buffer (RBC lysis buffer, 10x, BioLegend) according to manufacturer's protocol.<br>Skin. Skin single cell suspensions were obtained as previously described46. Briefly, Dorsal skin tissue was dissected (flattened, epidermis side up) and incubated at 37 °C for 45 min in 1 mL of 500 CU Dispase (Corning). The tissue was then chopped in RPMI 1640 GlutaMAX media supplemented with 1 mM sodium pyruvate, 1 mM non-essential amino acids, 50 μM β-mercaptoethanol, 20 mM HEPES, 100 U/mL penicillin, 100 mg/mL streptomycin, 0.5 mg/mL DNase I (all products from Sigma-Aldrich), and 0.25 mg/ml Liberase TL (Roche) and incubated for 1h45 min at 37 °C in a 5 % CO2 incubator. After filtering on a 40-μm filter kept on a 50 ml tube, cells were washed twice in PBS, BSA 0.5 %, 2 mM EDTA, the cell suspension was removed of skin debris using the cell debris removal solution (Miltenyi) following manufacturer's instructions.<br>Liver and lung. After perfusion and dissection of the liver and lung, the tissues were finely minced and digested using Collagenase D (0.7 mg/ml in PBS) for 30 min at 37ºC on a shaker in Freiburg while in Paris, the Gentlemacs operating system (Miltenyi) with the m_impTumor_01 program was used as described previously21. After washing the cell pellet twice at 400 x g for 5 min, the pellet was resuspended in 8 mL of 44% Percoll density gradient solution and underlaid with 5 ml 67% Percoll density gradient solution. Centrifugation (without breaks) was performed at 1600 x g for 20 min at room temperature. The cell layer containing mononuclear cells at the interface of the 44% and 67% density gradient centrifugation media was removed, transferred and washed. The resulting pellet was resuspended in 100 μl staining buffer (PBS containing 0.5% BSA |

and 2mM EDTA).
Intestinal intraepithelial cells. To isolate intraepithelial lymphocytes from the large and small intestine, tissues were dissected, cleaned to remove feces, cut open and chopped into 2 cm pieces. The pieces were treated with 1 mM Dithioerythritol (DTE) to release intra-epithelial lymphocytes (2X, 20 min each at 37ºC, constant shaking). The supernatant was filtered through 70-µm cell strainers (Falcon, Corning) kept on a 50-ml tube (Falcon, Corning) on ice. Cells were washed one with PBS containing 0.5% BSA and 2mM EDTA and a 44% and 67% density gradient centrifugation was performed as described above. After washing, the resulting pellet was resuspended in 100 µl staining buffer (PBS containing 0.5% BSA and 2mM EDTA).

Antibody staining, flow cytometry, and single-cell sorting. A 100 µl of antibody staining solution was prepared in PBS containing 0.5% BSA and 2mM EDTA and added to the isolated cells resuspended in 100 µl staining buffer as described above. Cells were incubated for 20 min on ice, washed thrice with 2 ml of 0.5% BSA in PBS and resuspended in 3 ml after the last wash for cell sorting. The following antibodies were used (all from BioLegend): TCRgd-APC, TCRb-BV421, CD45.1-PE, CD45.2-AF700, CD45.2-PerCP5.5, CD160-PECy7 and CD62L-BV421. Zombie Aqua and Zombie Green fixable viability kits were used to distinguish dead and living cells. Living TCRgd+ single gd T cells were sorted in BSA-coated tubes containing 50 µl of PBS. Using pulse geometry gates (FSC-W × FSC-H and SSC-W × SSC-H), doublets/multiplets were excluded. After the completion of sorting, the cells were processed through the different 10x Genomics workflows.

Instrument

BD FACSAria™ Fusion and BD CytoFLEX Flow Cytometer

Software

BD FACSDiva software v6 (Paris) and v8.0.2 (Freiburg) for sorting of cells for single-cell experiments and Cytoflex software (CytExpert) v2.4 for flow cytometry to collect the data for parabiosis FACS validation experiments.

Cell population abundance

Purity and the viability of gd T cells were validated by running a purity check on a few microliter of the sorted samples using the same flow cytometer used for sorting.

Gating strategy

Firstly, cells were gated using forward and side scatter to identify events corresponding to lymphocytes, doublets were excluded by gating on single cells using forward scatter height vs. area, living cells were selected by negativity for the viability dye (either Aqua L/D or Zombie Green™ Fixable Viability Kit), afterwards gd T cells were sorted using anti-mouse TCR γ/δ antibody to perform single-cell sequencing experiments. For parabiosis experiments, anti-Mouse CD45.1 and CD45.2 were used to distinguish and sort host-derived and partner-derived gd T cells for single cell sorting and validation experiments.

☒ Tick this box to confirm that a figure exemplifying the gating strategy is provided in the Supplementary Information.

