## [Peer Review File · Nature Immunology]

Peer Review Information

Journal: Nature Immunology

Manuscript Title: Multimodal profiling reveals site-specific adaptation and tissue residency hallmarks of $\gamma\delta$ T cells across organs in mice

Corresponding author name(s): Dr Sagar

Reviewer Comments & Decisions:

Decision Letter, initial version:
--

9th May 2023

Dear Dr Sagar,

Thank you for sending your proposed response to reviewers concerns on your Resource, "Multimodal profiling reveals site-specific adaptation and tissue residency hallmarks of $\gamma\delta$ T cells across epithelial and lymphoid tissues". We would be interested in considering a revised version that addresses reviewers concerns as you have outlined.

We hope you will find the referees' comments useful as you decide how to proceed. If you wish to submit a substantially revised manuscript, please bear in mind that we will be reluctant to approach the referees again in the absence of major revisions.

****Editor to outline here which reviewer concerns they are willing to overrule, and which must be addressed****

If you choose to revise your manuscript taking into account all reviewer and editor comments, please highlight all changes in the manuscript text file [OPTIONAL: in Microsoft Word format].

* If you have not done so already please begin to revise your manuscript so that it conforms to our Resource format instructions at <http://www.nature.com/ni/authors/index.html>. Refer also to any guidelines provided in this letter.

The Reporting Summary can be found here:

[REDACTED]

If you wish to submit a suitably revised manuscript we would hope to receive it within 6 months. If you cannot send it within this time, please let us know. We will be happy to consider your revision so long as nothing similar has been accepted for publication at Nature Immunology or published elsewhere.

Nature Immunology is committed to improving transparency in authorship. As part of our efforts in this direction, we are now requesting that all authors identified as 'corresponding author' on published papers create and link their Open Researcher and Contributor Identifier (ORCID) with their account on the Manuscript Tracking System (MTS), prior to acceptance. ORCID helps the scientific community achieve unambiguous attribution of all scholarly contributions. You can create and link your ORCID from the home page of the MTS by clicking on 'Modify my Springer Nature account'. For more information please visit www.springernature.com/orcid.

Thank you for the opportunity to review your work.

Sincerely,

Stephanie Houston
Editor
Nature Immunology

Reviewers' Comments:

Reviewer #1:

Remarks to the Author:

In this manuscript, submitted as a resource paper, the authors apply multiple single-cell technologies (RNA, ATAC, surface protein profiling) on gammadelta T cells derived from series of mouse epithelial and lymphoid tissues. Such an analysis is important as in the mouse gd T cells are highly tissue-restricted and different tissues can harbor different gd T cell types or subpopulations. Within these tissues, gd T cells can perform heterogeneous functions in tissue homeostasis and under disease conditions. In this manuscript, the single-cell analysis, combined with a parabiosis approach, provided insight into the tissue adaptation features of gd T cells in multiple organs and led to the identification of tissue-resident gd subsets and the underlying common or specialized regulatory programs governing their tissue residency. Furthermore, the provided resource data can serve the community to verify particular features, to compare with other immune cells etc.

The authors did well in organizing the wealth of data in a series of organized figures and they describe them in a logical flow in the text. However, the manuscript can be improved according to the points described below.

Major

1. Mention species in the title

Mouse and human gd T cells are regarded as highly different. For example, phosphoantigen-reactive gd T cells do not exist in mice and mouse skin dendritic epidermal T cells (DETC, also included in the current manuscript) have not been described in human. Therefore, it is important to mention the species in the title and in the abstract as well.

Line 71: I suggest to mention 'mouse' at the first line of this paragraph as well. As the authors refer here also only to mouse gd T cells regarding intrathymic development and the seeding to peripheral organs (and associated references). But this may not be clear for the reader. Thus please specify the species.

2. TCR usage

Line 205/Fig 3a: why was the Vg2 chain chosen to be part of oligonucleotide-conjugated antibodies? While it is known that in the mouse Vg usage can be associated with tissue type (for example the vast majority of DETC in the skin epidermis are Vg5+ (Tonegawa nomenclature), the paper does not introduce or discuss in any detail TCR associated with function or tissue localization. It is therefore surprising that an antibody against a Vg chain is included, and that this is the Vg2 chain (which gd TCR nomenclature?), a chain that is not particularly associated with a particular function/tissue as is the case for other Vg chains (such as Vg5 for the skin epidermis and Vg7 for the intestine).

Ideally, these resource data could also contain data on gd TCR/CDR3 at the single-cell level. However,

it is understandable that this was not the case since 10x genomics does not provide a kit for the gd TCR (only for the ab TCR). Furthermore, it is not compatible with the scATAC approach. Maybe the authors could mention that the TCR usage was not a main focus of their study and at least explain why the Vg2 antibody was chosen an oligonucleotide-conjugated antibody.

3.

Line 398

'To understand the organ-specific tissue residency features of gd T cell and the associated molecular programs, we performed parabiosis experiments, sorted partner- and host-derived gd T cells from six different organs and profiled them using scRNA-seq (Fig. 6a).'

Why is the small intestine not included like in the other single-cell approaches used in this paper? While in Fig 6f, in a flow cytometry approach to assess tissue-residency, the small intestine was included.

4.

Recently, two mouse single-cell gd papers were published that are not referenced in the current manuscript:

- Edwards et al 2022 JEM: mouse gd lung gene expression, with a focus on regulation by PD1 and TIM3. Thus for example the PD1 result (eg Fig 7i) of the current manuscript on RORC+ non circulating cells could be discussed in this context.
- Li et al 2022 Science Bulletin : mouse lymph node, spleen and thymus: single cell gene expression; thymus: also scATAC.

Please include these papers in the manuscript (like in the introduction and/or discussion)

5.

Tissue residency is an important part of the current manuscript. Therefore, while it is described in the methods section that lung and liver are perfused, it would be good to know how exactly these organs were perfused. These organs can contain a lot of blood, thus what is done to remove the blood from these organs is important in the context of assessing tissue-residency of lymphocytes.

6.

The proliferating cluster might be composed of different cell states and subsets that are cycling and are clustered in the same cluster because the majority of their transcriptome are related to cycling genes. This can explain the results mentioned in line 333-334 ('Interestingly, proliferating cells did not exhibit a specific open chromatin program and were scattered on UMAP representation (Fig. 4b)'). The authors could try to use the CellCycleScore function from the Seurat package to calculate the % of cycling genes and exclude this influence during the scale.data step prior to running Harmony package. Both Seurat and Harmony are used in the current paper.

Minor

1.

Line 187: 'Interestingly, skin Rorc+ gd T cells (mainly cluster 5) clustered separately from their counterparts in other organs (cluster 3), suggesting a distinct transcriptional signature of skin gd T17 cells (Fig. 1c, d).'

DGE analysis of skin gd17 vs other gd17 would be interesting to confirm this result and provide more insight.

2.

Line 304 (Fig 4c):

'Cluster 4, representing the intestinal Cd160+ gd subset, displayed open chromatin accessibility across several genes associated with intestinal IELs (Fig. 4c).'

Which genes? (this cannot be seen from Fig 4c). Why not providing a table with the data associated with Fig 4c?

3.

Line 286: error: should be Fig 4C (not Fig 5C)

4.

Why is there no green (small intestine, SI) labelling in Fig 1d? While from Fig 1e it appears that SI is largely enriched in cluster 1 (even more than the liver; and the liver is clearly present in Fig 1d).

5.

Line 133:

'In the second approach, simultaneous single-nucleus RNA sequencing (snRNA-seq) and assay for transposase-accessible chromatin using sequencing (scATAC-seq) were performed on pooled gd T cells from different organs without sample barcoding. Hence, these cells are labelled 'pooled' in Fig. 1b.'

There is no 'pooled' in Fig. 1B?

6.

References 52 and 53 are repetitions.

Reviewer #2:

Remarks to the Author:

Summary

du Halgouet et al provide a broad assessment of transcriptional profile and epigenetic state in $\gamma\delta$ T cells across numerous mouse tissues, an issue highly germane to their organismal biology.

The study appears to be technically sound and generally well written, although close scrutiny of the bioinformatics procedures would be warranted by more bioinformatically informed reviewers.

In terms of the biology it reveals, given the wealth of studies on mouse $\gamma\delta$ T cell populations, the judgement as to how significant the findings are, is somewhat subjective.

My overall feeling is that it does provide a useful pan-organ assessment of how $\gamma\delta$ biology is sculpted at anatomically distinct sites. The biological lessons are arguably somewhat scant, but three points emerge as the most significant. Firstly, the study provides evidence for phenotypic/transcriptional adaptation of $\gamma\delta$ T cells to different anatomical sites. Secondly, there is circumstantial evidence from epigenetic profiling (ATACseq) that some tissue subsets may be plastic in their influence on immunity, with potential for pro- and anti-inflammatory impacts, possibly influenced by distinct stimuli in different scenarios. Finally, the data indicate strong parallels between $\gamma\delta$ Trm biology with Trm cells from 'conventional' compartments.

Major criticisms

1. Marker choice

Around Figure 2a-d, the choice of markers to delineate populations was supervised. While these markers and populations will no doubt be of some interest to the field, it is unclear whether these optimally delineate prototypically distinct subsets, and a non-supervised comparison would be of interest. In some cases, eg Rorc+ cells, where the authors highlight considerable heterogeneity, it is unclear, especially in the absence of pseudo-time/trajectory analyses (see 2) or TCR analyses (see 3) whether this supervised approach best captures overall biology, and indeed whether conclusions about lineage relationships and tissue adaptation are wholly justified. Another obvious example is the 'proliferating cells' category, which highlights diverse transcriptional profiles, most probably indicating a finite proportion of many subsets are proliferating.

2. Lineage relationships

Building on point 1, the study's methodology cannot distinguish between cells of different origin that converge with respect to expression of the given marker in question, OR cells that share a similar origin and subsequently diverge, retaining expression of an original core gene.

From that perspective, the analysis of relationships between the different populations detected is arguably a weak point of the manuscript – there appears to be no formal approach employed capable of assigning either lineage relationships or alternatively direct mother-daughter relationships between cells. This is relevant to several aspects of the study, including distinct populations that share expression of CD160, but also Sell+Ly6c-/+ cells, and eg Rorc+ cells.

In this regard, one would assume some form of pseudo-time/trajectory analysis would not be beyond the capability of the authors given their current collective dataset and skillsets, and this could provide a reasonably objective test of some of the lineage relationships that the authors propose.

3. Co-association with $\gamma\delta$ TCR profile and clonality

Aside from V γ 2, and despite the commercial availability of other anti-TCR chain antibodies (including against V γ 1, and also against V γ 3/V γ 5 DETC chain), there was essentially no link made between any of the transcriptional/epigenetic profiles and the singular defining molecular feature of the $\gamma\delta$ T cell compartment, namely the $\gamma\delta$ TCR itself. This seems to be a major shortcoming of the study, particularly given emerging functional alignment between V γ chain usage and specificity for certain butyrophilin family members (eg V γ 7 IELs with BTNL1.6 in the mouse). For some subsets with almost homogenous TCRs and reasonably uniform transcriptional profiles (eg DETC V γ 5V δ 1 T cells selected by Skint-1) additional TCR data admittedly may not add much to understanding. However for many others it is likely to add an additional dimension to the study's findings. In particular, delineating how TCR chain (and ideally clonotype) identity segregates with transcriptional/epigenetic and ultimately functional status would have been particularly informative for intestinal T cells and might go a long way to determining the normality-associated sensing/homeostatic functions underpinned by TCR/BTN family interactions in the mouse gut. Finally, an additional benefit of analysing the TCR is that it represents a natural lineage tracking element that could help resolve lineage inter-relationships between different $\gamma\delta$ T cell subpopulations. This could be highly relevant to RORc+ cells, which are known to derive from two different lineages, the embryonically derived V γ 6V δ 1, and the neonatally V γ 4 population, which might be expected to display quite distinct transcriptional/epigenetic profiles. The authors should therefore strongly consider even a relatively shallow assessment of $\gamma\delta$ TCR repertoire and clonality to bolster the current findings.

4. Relationship to human $\gamma\delta$ biology

Although the study is overtly focussed on the mouse $\gamma\delta$ T cell compartment, there is no attempt to relate the findings to human work. Although analysis of human tissue $\gamma\delta$ T cells is at a relatively early stage, a number of previous human studies provide data and insights that overlap with those in the study and should be discussed. These include Hunter et al, 2018 (PMID: 29758330) who studied intrahepatic $\gamma\delta$ T cells and noted both circulating and $\gamma\delta$ Trm subpopulations. Secondly, Zakeri et al, 2022 (PMID 35296658), who provided strong evidence for long-lived hepatic Trm in different $\gamma\delta$ T cell subsets. Also the expression of TCF7 and LEF1 on naïve human V δ 1 T cells could be noted, for which McMurray et al, 2022 (PMID 35613583) would be relevant.

Author Rebuttal to Initial comments

Reviewer #1

(Remarks to the Author)

In this manuscript, submitted as a resource paper, the authors apply multiple single-cell technologies (RNA, ATAC, surface protein profiling) on gammadelta T cells derived from series of mouse epithelial and lymphoid tissues. Such an analysis is important as in the mouse gd T cells are highly tissue-restricted and different tissues can harbor different gd T cell types or subpopulations. Within these tissues, gd T cells can perform heterogeneous functions in tissue homeostasis and under disease conditions. In this manuscript, the single-cell analysis, combined with a parabiosis approach, provided insight into the tissue adaptation features of gd T cells in multiple organs and led to the identification of tissue-resident gd subsets and the underlying common or specialized regulatory programs governing their tissue residency. Furthermore, the provided resource data can serve the community to verify particular features, to compare with other immune cells etc. The authors did well in organizing the wealth of data in a series of organized figures and they describe them in a logical flow in the text. However, the manuscript can be improved according to the points described below.

*Response: We sincerely thank the reviewer for acknowledging the value of our manuscript and the data analysis approach. We share the reviewer's perspective that conducting a multimodal analysis of gd T cells in both mouse and human at a single-cell resolution can significantly contribute to our understanding of their biology and to address questions concerning the redundancy or uniqueness of gd T cells in their roles in homeostasis and disease compared to the other T cell lineages. Furthermore, we extend our gratitude to the reviewer for carefully reviewing our manuscript and offering valuable suggestions. We have worked on addressing each of the reviewer's concerns and comments, and below we provide a point-to-point response. **Changes in the revised manuscript are highlighted in yellow for the convenience of reviewer and editor.***

Major

1. Mention species in the title

Mouse and human gd T cells are regarded as highly different. For example, phosphoantigen-reactive gd T cells do not exist in mice and mouse skin dendritic epidermal T cells (DETC, also included in the current manuscript) have not been described in human. Therefore, it is important to mention the species in the title and in the abstract as well.

Line 71: I suggest to mention 'mouse' at the first line of this paragraph as well. As the authors refer here also only to mouse gd T cells regarding intrathymic development and the seeding to peripheral organs (and associated references). But this may not be clear for the reader. Thus please specify the species.

Response: Indeed, we agree that several features of $\gamma\delta$ T cells are species-specific. Therefore, we have now included "in mice" in the title and the abstract, as well as where we talk about intrathymic development-

*New Title: Multimodal profiling reveals site-specific adaptation and tissue residency hallmarks of gd T cells across epithelial and lymphoid tissues **in mice***

*Abstract: Here, by multimodal single-cell profiling of gd T cells across seven organs **in mice**, we reveal that various tissues harbor unique site-adapted gd subsets.....*

*Previous line 71: The intrathymic development and tissue homing of gd T cells take place in a wavelike fashion **in mice**.....*

*Furthermore, the end paragraph in the discussion of the manuscript says: Altogether, our study provides the first single-cell multimodal landscape of gd T cells across multiple tissues **in mice** and uncovers their heterogeneity and tissue-specific adaptation based on gene expression, chromatin accessibility and TCR features....*

2. TCR usage

Line 205/Fig 3a: why was the Vg2 chain chosen to be part of oligonucleotide-conjugated antibodies?

While it is known that in the mouse Vg usage can be associated with tissue type (for example the vast majority of DETC in the skin epidermis are Vg5+ (Tonegawa nomenclature), the paper does not introduce or discuss in any detail TCR associated with function or tissue localization. It is therefore surprising that an antibody against a Vg chain is included, and that this is the Vg2 chain (which gd TCR nomenclature?), a chain that is not

particularly associated with a particular function/tissue as is the case for other Vg chains (such as Vg5 for the skin epidermis and Vg7 for the intestine).

Ideally, these resource data could also contain data on gd TCR/CDR3 at the single-cell level. However, it is understandable that this was not the case since 10x genomics does not provide a kit for the gd TCR (only for the ab TCR). Furthermore, it is not compatible with the scATAC approach. Maybe the authors could mention that the TCR usage was not a main focus of their study and at least explain why the Vg2 antibody was chosen an oligonucleotide-conjugated antibody.

Response: We thank the reviewer to bring this important point forward, the Vg2 antibody was included in the CITE-Seq panel as it was the only oligo-conjugated antibody that was commercially available. Since both reviewers raised concerns about the utility of this resource article without the TCR repertoire profiling at single-cell resolution, we have performed the scTCR-seq in the revised version. Using the TCR rearrangement data at a single-cell resolution, we were further able to characterize the TCR characteristics of the identified gd subsets and their developmental relationships among each other in five different organs. The data is presented in main Figure 6 and Extended data figure 8.

Major conclusions are- Rorc+ and Gzmb+ cells are very distinct set of cells while other subsets show overlapping clones. Liver and small intestine contains the least diverse clones. Sell+ Ly6c2-, Sell+ Ly6c2+ and Cd160+ cells are developmentally related. Rorc+ cells with Vg6 and Vg4 chains are transcriptionally distinct set of cells. Liver and lymph node clones exhibit the highest and least overlap with other tissues, respectively.

3. Line 398

'To understand the organ-specific tissue residency features of gd T cell and the associated molecular programs, we performed parabiosis experiments, sorted partner- and host-derived gd T cells from six different organs and profiled them using scRNA-seq (Fig. 6a).'

Why is the small intestine not included like in the other single-cell approaches used in this paper? While in Fig 6f, in a flow cytometry approach to assess tissue-residency, the small intestine was included.

Response: The small intestinal sample preparation during the day of the experiment when we were profiling $\gamma\delta$ T cells for single-cell RNA-seq from the parabiotic mice did not work well. Fig. 6f (now Fig. 7f) experiments were performed independently of Fig. 6a (now Fig. 7a) experiments. While performing flow cytometry experiments for Fig. 6f (now Fig. 7f), we realized that the majority of cells in the small intestine are tissue-resident. Therefore, we decided not to profile these cells using single-cell RNA-seq anymore, as we assumed that this may not provide much extra information.

4. Recently, two mouse single-cell gd papers were published that are not referenced in the current manuscript:

- Edwards et al 2022 JEM: mouse gd lung gene expression, with a focus on regulation by PD1 and TIM3. Thus for example the PD1 result (eg Fig 7i) of the current manuscript on RORC+ non circulating cells could be discussed in this context.
- Li et al 2022 Science Bulletin : mouse lymph node, spleen and thymus: single cell gene expression; thymus: also scATAC. Please include these papers in the manuscript (like in the introduction and/or discussion)

Response: We thank the reviewer for pointing this out. We find both references valuable for our study and apologize for not citing them. Both references are included in the revised version of the manuscript.

Context of ref 1: Resident gdT17 cells exclusively expressed transmembrane heparan sulfate proteoglycan, Sdc4 and Pdccl1 (encoding PD-1) (Fig. 8I). PD-1 has been shown previously to be expressed by a subset of gdT17 cells which are Vg6+ and display a TRM phenotype.

Context of ref 2: Retrospectively, to determine key TFs associated with various gd subsets using our scRNA-seq data, we also performed SCENIC analysis, which identified few TFs consistent with chromVAR including Nr1d1 which has been recently shown to regulate gdT17 differentiation.

5.

Tissue residency is an important part of the current manuscript. Therefore, while it is described in the methods section that lung and liver are perfused, it would be good to know how exactly these organs were perfused. These organs can contain a lot of blood, thus what is done to remove the blood from these organs is important in the context of assessing tissue-residency of lymphocytes.

Response: We agree that the lung and liver contain a significant amount of blood, and it is essential to provide a detailed description of the perfusion protocols. During our experiments, we took extreme care to ensure complete perfusion of the organs. We used dual perfusion via both the heart and portal vein to minimize the amount of remaining blood in these tissues. As a result, these tissues displayed changes in color indicating an efficient perfusion (e.g., white lungs, "pale" liver). We have now included these details in the methods section by adding the following sentence:

To ensure complete lung and liver perfusion (evidenced by organ color change caused by the loss of red blood cells), a 20 mL syringe with a 22G needle was used to inject 1X PBS

starting with the right ventricle of the heart (10 mL) followed by the hepatic portal vein (10 mL).....

6.

The proliferating cluster might be composed of different cell states and subsets that are cycling and are clustered in the same cluster because the majority of their transcriptome are related to cycling genes. This can explain the results mentioned in line 333-334 ('Interestingly, proliferating cells did not exhibit a specific open chromatin program and were scattered on UMAP representation (Fig. 4b)'). The authors could try to use the CellCycleScore function from the Seurat package to calculate the % of cycling genes and exclude this influence during the scale.data step prior to running Harmony package. Both Seurat and Harmony are used in the current paper.

Response: Indeed, we agree that our single-cell RNA-seq data shows different subsets clustering together based on cell cycle-related genes (Quantified in Fig. 2e and 3f). However, our intention was not to regress out these genes, as it is essential to understand that proliferating cells belonging to many gd subsets exist in different organs. As the reviewer correctly pointed out, unlike gene expression, proliferating cells do not exhibit a unique chromatin associated with the cell cycle, and they cluster based on the cell state. We find these observations significant as they highlight the difference between RNA and chromatin accessibility. While both modalities are generally very correlated, in the case of proliferation, it seems that gene expression and chromatin accessibility landscapes differ. Therefore, we did not regress out the cell cycle effects. In the revised version, we have included a UMAP indicating the cell cycle phases of the gd T cells but still decided not to regress out the cell cycle effect (Extended data figure 1g) so that the readers acknowledge that a very minor fraction of many gd subsets proliferate within the tissues.

Furthermore, we have edited the following sentence-

We further identified two minor groups of gd T cells – an effector-like gd T cell population from the liver and lung (cluster 17) characterized by the expression of *Klrg1* and *Zeb2* and proliferating gd T cells (cluster 15, *Mki67+*)

to

We further identified two minor groups of gd T cells – an effector-like gd T cell population from the liver and lung (cluster 17) characterized by the expression of *Klrg1* and *Zeb2* and **proliferating gd T cells at S, G2 and M phases of cell cycle** (cluster 15).

Minor

1.

Line 187: ‘Interestingly, skin *Rorc*+ gd T cells (mainly cluster 5) clustered separately from their counterparts in other organs (cluster 3), suggesting a distinct transcriptional signature of skin gd T17 cells (Fig. 1c, d).’

DGE analysis of skin gd17 vs other gd17 would be interesting to confirm this result and provide more insight.

Response: This analysis is included in the revised version as Extended Data Fig. 1f and Supplementary Table 2. Following sentence is added in the manuscript-

Differential gene expression analysis between clusters 3 and 5 revealed that skin gdT17 cells upregulate genes regulating several metabolic processes and NF-κB pathway as well as factors associated with epigenetic modifications (Extended Data Fig. 1f, Supplementary Table 2).

2.

Line 304 (Fig 4c):

‘Cluster 4, representing the intestinal Cd160+ gd subset, displayed open chromatin accessibility across several genes associated with intestinal IELs (Fig. 4c).’

Which genes? (this cannot be seen from Fig 4c). Why not providing a table with the data associated with Fig 4c?

Response: Tables (supplementary tables 3 and 4) corresponding to the heatmap shown as Fig. 4c are included in the revised version.

3.

Line 286: error: should be Fig 4C (not Fig 5C)

Response: We thank the reviewer for reviewing the manuscript so carefully. This error has been rectified.

4.

Why is there no green (small intestine, SI) labelling in Fig 1d? While from Fig 1e it appears that SI is largely enriched in cluster 1 (even more than the liver; and the liver is clearly present in Fig 1d).

Response: There are small intestinal (SI) cells ($n = 3520$ cells), but they are difficult to notice. We tried to change the color of the SI cells, but this did not lead to better visualization as they are masked by the large intestinal (LI) and pooled cells. Therefore, we added an extra figure as Extended Data Fig. 1d to highlight the contribution of different organs.

5.

Line 133:

'In the second approach, simultaneous single-nucleus RNA sequencing (snRNA-seq) and assay for transposase-accessible chromatin using sequencing (scATAC-seq) were performed on pooled gd T cells from different organs without sample barcoding. Hence, these cells are labelled 'pooled' in Fig. 1b.'

There is no 'pooled' in Fig. 1B?

Response: There is a grey bar labelled as pooled in Fig 1b.

6.
References 52 and 53 are repetitions.

Response: We thank the reviewer for highlighting this issue. This has been rectified now.

Reviewer #2

(Remarks to the Author)

Summary

du Halgouet et al provide a broad assessment of transcriptional profile and epigenetic state

in $\gamma\delta$ T cells across numerous mouse tissues, an issue highly germane to their organismal biology.

The study appears to be technically sound and generally well written, although close scrutiny of the bioinformatics procedures would be warranted by more bioinformatically informed reviewers.

In terms of the biology it reveals, given the wealth of studies on mouse $\gamma\delta$ T cell populations, the judgement as to how significant the findings are, is somewhat subjective.

My overall feeling is that it does provide a useful pan-organ assessment of how $\gamma\delta$ biology is sculpted at anatomically distinct sites. The biological lessons are arguably somewhat scant, but three points emerge as the most significant. Firstly, the study provides evidence for phenotypic/transcriptional adaptation of $\gamma\delta$ T cells to different anatomical sites. Secondly, there is circumstantial evidence from epigenetic profiling (ATACseq) that some tissue subsets may be plastic in their influence on immunity, with potential for pro- and anti-inflammatory impacts, possibly influenced by distinct stimuli in different scenarios. Finally, the data indicate strong parallels between $\gamma\delta$ T_{rm} biology with T_{rm} cells from 'conventional' compartments.

*Response: We sincerely thank the reviewer for the appreciation of the analysis and for highlighting the three major conclusions of our study. Indeed, the study is aimed to take a cross-organ approach and look at the heterogeneity of gd T cells and their tissue adaptation features in a systematic way using a multimodal approach and compare these features with other T cell lineages. Parabiosis experiments were performed to elucidate the tissue residency features of these subsets. We appreciate the reviewer's valuable comments, which we have duly addressed and incorporated into the revised manuscript. In the following section, we provide a detailed response to each of the reviewer's concerns, addressing them point by point. **Changes in the revised version of manuscript are highlighted in yellow for the convenience of reviewer and editor.***

Major criticisms

1. Marker choice

Around Figure 2a-d, the choice of markers to delineate populations was supervised. While these markers and populations will no doubt be of some interest to the field, it is unclear whether these optimally delineate prototypically distinct subsets, and a non-supervised comparison would be of interest. In some cases, eg Rorc+ cells, where the authors highlight considerable heterogeneity, it is unclear, especially in the absence of pseudo-time/trajectory analyses (see 2) or TCR analyses (see 3) whether this supervised approach best captures overall biology, and indeed whether conclusions about lineage relationships and tissue adaptation are wholly justified. Another obvious example is the 'proliferating cells' category,

which highlights diverse transcriptional profiles, most probably indicating a finite proportion of many subsets are proliferating.

Response: We think that this could be a misunderstanding because of the way we wrote the following sentence-

“Although we identified 22 clusters in our scRNA-seq data, we noticed that differentially expressed genes in each cluster were not mutually exclusive. Instead, many clusters showed differential regulation of identical sets of genes. These results suggest that many of the identified clusters represent varying cellular states of the same cell type, requiring us to classify $\gamma\delta$ subsets based on a supervised analysis of distinct key marker genes. Using this supervised approach, we grouped 22 clusters into 8 major subsets with unique gene expression profiles (Fig. 2a-c).”

We edited the sentence to-

*“Although **unsupervised clustering identified** 22 clusters in our scRNA-seq data, we noticed that differentially expressed genes in each cluster were not mutually exclusive. Instead, many clusters showed differential regulation of identical sets of genes. These results suggest that many of the identified clusters represent varying cellular states of the same cell type, requiring us to classify $\gamma\delta$ subsets based on a supervised analysis of distinct key marker genes. Using this supervised approach, we grouped 22 clusters into 8 major subsets with unique gene expression profiles (Fig. 2a-c).”*

We would like to emphasize that the analysis for identifying clusters in the dataset was completely unsupervised. We have provided a list of differentially expressed genes in each cluster resulting from this unsupervised analysis. To define the distinct subsets, we looked at these unsupervised differentially expressed genes and then selected highly differentially expressed genes relevant to T cell biology to interpret the biological significance of these clusters. Furthermore, we focused on cell surface markers that were differentially expressed in these cell clusters, which allowed us to use them as a proxy to isolate and validate these transcriptionally distinct subsets. This approach provided a convenient way to identify these distinct unsupervised subsets of gd T cells for further FACS validation experiments. The epigenetic data supported our subset classification based on this unsupervised clustering.

The question regarding the different subsets clustering together based on cell cycle-related genes was also raised by reviewer 1. We did not regress out the cell cycle effects so that the readers acknowledge that a very minor fraction of many gd subsets proliferate within the tissues. In the revised version, we have included a UMAP indicating the cell cycle phases of the gd T cells (Extended data figure 1g).

Furthermore, we have edited the following sentence-

*We further identified two minor groups of gd T cells – an effector-like gd T cell population from the liver and lung (cluster 17) characterized by the expression of *Klrg1* and *Zeb2* and proliferating gd T cells (cluster 15, *Mki67*+)*

to

*We further identified two minor groups of gd T cells – an effector-like gd T cell population from the liver and lung (cluster 17) characterized by the expression of *Klrg1* and *Zeb2* and **proliferating gd T cells at S, G2 and M phases of cell cycle** (cluster 15).*

The reviewer correctly notes that the manuscript does not include lineage trees, pseudo-time analysis and single-cell TCR repertoire sequencing, which are important aspects, especially for some gd subsets that appear to be developmentally related. We have addressed these points below.

2. Lineage relationships

Building on point 1, the study's methodology cannot distinguish between cells of different origin that converge with respect to expression of the given marker in question, OR cells that share a similar origin and subsequently diverge, retaining expression of an original core gene.

From that perspective, the analysis of relationships between the different populations detected is arguably a weak point of the manuscript – there appears to be no formal approach employed capable of assigning either lineage relationships or alternatively direct mother-daughter relationships between cells. This is relevant to several aspects of the study, including distinct populations that share expression of CD160, but also Sell+Ly6c-/+ cells, and eg Rorc+ cells.

In this regard, one would assume some form of pseudo-time/trajectory analysis would not be beyond the capability of the authors given their current collective dataset and skillsets, and this could provide a reasonably objective test of some of the lineage relationships that

the authors propose.

Response: We thank the reviewer for bringing this issue here. This is not a trivial problem to solve.

As we stated in our review in 2020 (Sagar and Grün, PMID: 32780577):

“The reconstruction of differentiation trajectories from scRNA-seq data relies on the assumption that single-cell transcriptomes encompass all naïve, intermediate, and mature cell states with sufficient sampling coverage.”

Furthermore, we note that-

“It is a common assumption of lineage reconstruction methods that similarity in gene expression profiles reflects developmental proximity.”

Therefore, it is very tricky to apply these computational approaches to infer lineage trees in the present study as gd subsets are very heterogeneous across tissues and may not be developmentally related as they may develop at different timepoints in the thymus. As reviewer noticed correctly, we identify distinct population of gd T cells which share CD160 expression across tissues. We would like to draw the attention of the reviewer to the observation that these cells are highly tissue adapted in the organ they reside and therefore have very distinct gene expression signature in a particular organ. For example, CD160+ cells from the intestine are distinct from the liver. Therefore, applying a trajectory analysis algorithm may lead to some misleading results.

The problem is further complicated by the fact that most of the lineage tree algorithms (e.g., Monocle) do not predict the direction of differentiation and require the information about the root (i.e., progenitors) a priori. Given the distinct and heterogeneous site-adapted subsets in different organs, it will be erroneous to define one subset as the progenitor. One notable exception to this is RNA velocity which predicts differentiation trajectories based on the kinetics of the mRNA lifecycle and provide velocity vectors of each cell which translates into the directionality of differentiation.

The reviewer has already hinted the other way to deduce the lineage relationship - through TCR clonotype analysis which was missing in the first version of the manuscript. This analysis is now included in the manuscript and discussed in detail in the next concern of the reviewer.

Nevertheless, to respect the reviewer’s valid point about lineage relations, we applied RNA velocity and Monocle to the dataset of 13,753 cells where we also profiled the cells using single-cell TCR sequencing. We did not apply the trajectory analysis on the completely

integrated datasets as we noticed batch effects between different datasets and RNA velocity doesn't allow for batch correction.

The results were inconclusive and in some cases contradictory. For example, RNA velocity predicted directionality of differentiation from *Sell+ Ly6c2+* cells to *Sell+ Ly6c2-* cells which believe is biologically incorrect as *Sell+ Ly6c2-* seem to be the progenitors with highest expression of *CD24*, closed locus of *Tbx21* and highest expression of *Tcf7* and *Lef1*. Monocle analysis was also inconclusive as it connected all the cell types to each other.

We also performed trajectory analysis for each cell types and organ separately (data not shown), but this was also not conclusive. Therefore, we did not include this data in the manuscript and rather focused on shared clonotypes to deduce lineage relationships.

3. Co-association with $\gamma\delta$ TCR profile and clonality

Aside from V γ 2, and despite the commercial availability of other anti-TCR chain antibodies (including against V γ 1, and also against V γ 3/V γ 5 DETC chain), there was essentially no link made between any of the transcriptional/epigenetic profiles and the singular defining molecular feature of the $\gamma\delta$ T cell compartment, namely the $\gamma\delta$ TCR itself. This seems to be a major shortcoming of the study, particularly given emerging functional alignment between V γ chain usage and specificity for certain butyrophilin family members (eg V γ 7 IELs with BTNL1.6 in the mouse). For some subsets with almost homogenous TCRs and reasonably uniform transcriptional profiles (eg DETC V γ 5V δ 1 T cells selected by Skint-1) additional TCR data admittedly may not add much to understanding. However for many others it is likely to add an additional dimension to the study's findings. In particular, delineating how TCR chain (and ideally clonotype) identity segregates with transcriptional/epigenetic and ultimately functional status would have been particularly informative for intestinal T cells and might go a long way to determining the normality-associated sensing/homeostatic functions underpinned by TCR/BTN family interactions in the mouse gut. Finally, an additional benefit of analysing the TCR is that it represents a natural lineage tracking element that could help resolve lineage inter-relationships between different $\gamma\delta$ T cell subpopulations. This could be highly relevant to RORc+ cells, which are known to derive from two different lineages, the embryonically derived V γ 6V δ 1, and the neonatally V γ 4 population, which might be expected to display quite distinct transcriptional/epigenetic profiles. The authors should therefore strongly consider even a relatively shallow assessment of $\gamma\delta$ TCR repertoire and clonality to bolster the current findings.

Response: We acknowledge that the lack of TCR repertoire analysis was a major limitation of the study, and we appreciate both the reviewers bringing this up. We have now performed single-cell TCR repertoire analysis in the liver, intestine, lung, spleen and the lymph nodes and have included the data as Figure 6 and Extended Data Fig. 8. We did not profile skin as reviewers suggested.

Major conclusions are as the reviewer predicted- Rorc+ and Gzmb+ cells are very distinct set of cells in terms of their TCR while other subsets show overlapping clones. Liver and small intestine contains the least diverse clones. Sell+ Ly6c2-, Sell+ Ly6c2+ and Cd160+ cells are developmentally related. Rorc+ cells with Vg6 and Vg4 chains are transcriptionally distinct set of cells.

We thank the reviewer once again for suggesting this analysis. We believe that the manuscript has significantly improved because of this analysis.

4. Relationship to human $\gamma\delta$ biology

Although the study is overtly focussed on the mouse $\gamma\delta$ T cell compartment, there is no

attempt to relate the findings to human work. Although analysis of human tissue $\gamma\delta$ T cells is at a relatively early stage, a number of previous human studies provide data and insights that overlap with those in the study and should be discussed. These include Hunter et al, 2018 (PMID: 29758330) who studied intrahepatic $\gamma\delta$ T cells and noted both circulating and $\gamma\delta$ Trm subpopulations. Secondly, Zakeri et al, 2022 (PMID 35296658), who provided strong evidence for long-lived hepatic Trm in different $\gamma\delta$ T cell subsets. Also the expression of TCF7 and LEF1 on naïve human V δ 1 T cells could be noted, for which McMurray et al, 2022 (PMID 35613583) would be relevant.

Response: We appreciate the reviewer for bringing up the important point that our study does not focus on human gd biology. As pointed out by the previous reviewer, there are clear differences between human and mouse gd subsets. However, our analysis also highlights some common regulatory mechanisms. We thank the reviewer for providing relevant literature and guiding us towards discussing these findings. We have added the following paragraph in the discussion to address this point-

“Although our study primarily focuses on gd T cells in mice, it highlights several parallels between gd T cells in mice and humans. For instance, in both mice and humans, tissue-resident gd T cells in the liver are characterized by the expression of CD49a and CD69. Furthermore, in both species they exhibit the expression of Gzmb, and Blimp1, and demonstrate restricted TCR diversity. In the intestine, gd IELs in both mice and humans primarily express Itgae, Gzma and Gzmb. Moreover, in humans, the peripheral blood contains a subset of naïve gd T cells that express TCF7 and LEF1, and these cells exhibit a diverse TCR repertoire similar to Sell+ Ly6c2- naïve cells observed in mice. Conducting a comprehensive single-cell multimodal profiling of human gd T cells across various organs and performing an in-depth analysis in conjunction with the data presented in this study has the potential to provide further unique insights into gd T cell biology across species.”

Decision Letter, first revision:

6th Sep 2023

Dear Dr Sagar,

Thank you for providing a proposed rebuttal to reviewers concerns on your resource "Multimodal profiling reveals site-specific adaptation and tissue residency hallmarks of $\gamma\delta$ T cells across epithelial and lymphoid tissues in mice". We are interested in the possibility of publishing your study in Nature Immunology revised as you proposed.

We therefore invite you to revise your manuscript, please highlight all changes in the manuscript text file in Microsoft Word format.

We are committed to providing a fair and constructive peer-review process. Do not hesitate to contact

us if there are specific requests from the reviewers that you believe are technically impossible or unlikely to yield a meaningful outcome.

* If you have not done so already please begin to revise your manuscript so that it conforms to our Resource format instructions at <http://www.nature.com/ni/authors/index.html>. Refer also to any guidelines provided in this letter.

* Please include a revised version of any required reporting checklist. It will be available to referees to aid in their evaluation of the manuscript goes back for peer review. They are available here:

Reporting summary:

When submitting the revised version of your manuscript, please pay close attention to our [href="https://www.nature.com/nature-portfolio/editorial-policies/image-integrity">Digital Image Integrity Guidelines. and to the following points below:](https://www.nature.com/nature-portfolio/editorial-policies/image-integrity)

[REDACTED]

We hope to receive your revised manuscript within two weeks. If you cannot send it within this time, please let us know. We will be happy to consider your revision so long as nothing similar has been accepted for publication at Nature Immunology or published elsewhere.

Nature Immunology is committed to improving transparency in authorship. As part of our efforts in this direction, we are now requesting that all authors identified as 'corresponding author' on published

papers create and link their Open Researcher and Contributor Identifier (ORCID) with their account on the Manuscript Tracking System (MTS), prior to acceptance. ORCID helps the scientific community achieve unambiguous attribution of all scholarly contributions. You can create and link your ORCID from the home page of the MTS by clicking on 'Modify my Springer Nature account'. For more information please visit www.springernature.com/orcid.

Sincerely,

Stephanie Houston, PhD
Senior Editor
Nature Immunology

Reviewers' Comments:

Reviewer #1:

Remarks to the Author:

Overall, the authors addressed adequately my previous comments. However, because of the new single-cell gd TCR part, I have few novel remarks regarding this part. While including this approach certainly improved the manuscript, following points need to be addressed:

1. Analysis of clonotype sharing.

Fig. 6H:

What is on the y-axis? This is not indicated.

It is confusing to mix different definitions of 'clonotypes' in the same figure. There are clonotypes only defined by the gamma CDR3 sequence, while other clonotypes are defined by both the gamma and delta CDR3. Why is this done? Why not only focusing on gamma CDR3 or delta CDR3 or both gamma+delta? Why doing this 'mix'? These different 'types' of clonotypes need either be analysed separately or it needs to be well explained why this analysis is done in such a way. For example, expanded clone 4 differ only by one amino acid in the gamma CDR3 from expanded clone 1, but in expanded clone 1 the gamma chain is put together with the delta chain, while this is not done for expanded clone 4.

Is the Shannon index based on such a 'mix' of clonotypes? Again, Shannon index is usually calculated per TCR chain, not a mix.

Furthermore, the presence of several of the described clonotypes have been described before. An overview can be found in the review by Papadopoulou et al 2020 Immunol Rev (DOI: 10.1111/imr.12926) and thus can be included as a reference.

Fig. 8E:

It would be good if the authors expand a bit on the type of the interorgan shared clones that the liver

has in more abundance (like usage of Vg chain: Vg6?).

2. Detection of NKT-like gd TCR clonotype?

The authors describe in the paper (line 173):

'Importantly, cluster 12 expressed Zbtb16 (encoding PLZF) and Il4, indicating that these cells are NKT-like IFN-g/IL-4 double producers³⁵.'

It is known that this population express a particular clonotype (CALWELVGGIRATDKLVF TRAV15-1-DV6-1: see overview in Papadopoulou et al 2020 Immunol Rev). Did the authors did not detect this one as an 'expanded clonotype'? If detected, was it specific for the liver and/or spleen? Note the annotation 'TRAV15-1DV6-1': thus this V gene segment can also be used in an alpha chain: this may need a double check whether such CDR3 data are adequately analysed by the Cell Ranger software.

3. How many mice were used in the scTCR experiments? While the sequences of other genes are not unique to each mouse, TCR rearrangement are mostly unique (private) per subject/mouse. If different mice were used: did the authors attempt to analyse the scTCR data separately per mouse? Were the data consistent between the different mice? Or were the number of cells not sufficient to perform such analysis?

More generally, the number of mice used in the other types of experiments may also be useful to include. But it is especially relevant to include that information for the scTCR experiments.

Reviewer #3:

Remarks to the Author:

Sagar and colleagues provide a well prepared and well-presented overview of the diverse more or less tissue-resident gd T cell populations found in WT laboratory C57BL/6 mice.

The ms does not comprise a dense and completely new story, as gd T cells are considered prototype tissue-resident T cells. But it will certainly be an extremely timely and valuable „go-to“ resource for many current projects aiming at deciphering the composition and functional diversity of human and mouse gd T cell populations in tissues. (... and highly cited). To this end, data accessibility and re-usability will be crucial for the community.

Comments:

The current ms is already a revised version of a previous submission. The two reviewers did a great job and most issues were adequately addressed by the authors. In particular, both suggested to include an analysis of clonal relationships between the identified clusters of mouse gd T cells. The authors complied by including a state-of-the art scTCR analysis using the 10x 5' VDJ workflow that is commercially available for ab TCR.

1. Fig. 6 Although this method is still a cutting-edge experimental procedure, TCR repertoire analyses (scTCR-seq) could still be more comprehensive (more cells). However, as is, it is already an important and decisive selling point of the current revised ms.

In that context, it might be useful to reference and discuss a recent report, which also experimented with combining mouse sc gd TCR-seq and scRNA-seq, doi: 10.1016/j.celrep.2023.112253.

2. Fig. 5 is fantastic, congratulations!

3. I think it is absolutely required to clearly point out which nomenclature is used throughout this resource work.

4. The discussion is concise, very suitable for a resource paper. A very short discussion of the analogy and perhaps differences of tissue-resident gd T cells and ab Trm cells could still be interesting.

Wish list:

- Multimodal analysis of gd T cells also from other inbred strains, or even outbred wildlings would be super exciting, but probably beyond the scope of this study.
- Analysis or integration of thymic gd T cells (e.g., their own data from <https://doi.org/10.15252/embj.2019104159>) would be a great complementary resource to map $\gamma\delta$ T-cell differentiation and thereby observe some freshly generated gd T cells acquiring their tissue-specific effector functions. At the same time, there is accumulating evidence for the presence of long-lived thymus-resident effector gd T cells in mice.

Minor

a) Line 73: the review article to be cited here should probably be reference #19?
Carding, S.R. & Egan, P.J. Gammadelta T cells: functional plasticity and heterogeneity. *Nat Rev Immunol* 2, 336-345 (2002).

b) Line 87: The context of reference #25 is misleading. The authors claim in the following that "... these studies were performed using flow cytometry with very few cell surface markers, providing limited insights into their tissue residency features. Hence, deciphering the tissue adaptation features of gd T cells in multiple organs using single-cell transcriptomics holds the key to identifying specific tissue-resident gd subsets and the underlying common or specialized regulatory programs governing their tissue residency."

This statement misleading because even the title of the work cited as reference #25 starts with "single-cell transcriptomics" and was thus not "performed using flow cytometry with very few cell surface markers" as suggested by the authors.

Ref. #25 Tan, L. et al. Single-Cell Transcriptomics Identifies the Adaptation of Scart1(+) Vgamma6(+) T Cells to Skin Residency as Activated Effector Cells. *Cell Rep* 27, 3657- 3671 e3654 (2019).

c) The main datasets are combining pooled and hashtagged scRNA libraries, which were all integrated using Harmony to correct for batch effects. A statement evaluating whether this may have led to over-correction would be helpful.

Author Rebuttal, first revision:

Reviewer #1

(Remarks to the Author)

Overall, the authors addressed adequately my previous comments.

We would like to express our sincere gratitude to the reviewer for acknowledging our efforts in addressing their previous comments. We are especially appreciative of the insightful feedback provided on the TCR analysis. These constructive comments are invaluable and will undoubtedly contribute to the improvement of our manuscript.

However, because of the new single-cell gd TCR part, I have few novel remarks regarding this part. While including this approach certainly improved the manuscript, following points need to be addressed:

1. Analysis of clonotype sharing.

Fig. 6H:

What is on the y-axis? This is not indicated.

We sincerely apologize for the oversight in not including the scale on the y-axis. The y-axis signifies the proportions of cells expressing the mentioned clonotypes within the organ, and we acknowledge that this information is crucial for a comprehensive understanding of the data. We have rectified this mistake in the revised version of the manuscript.

It is confusing to mix different definitions of 'clonotypes' in the same figure. There are clonotypes only defined by the gamma CDR3 sequence, while other clonotypes are defined by both the gamma and delta CDR3. Why is this done? Why not only focusing on gamma CDR3 or delta CDR3 or both gamma+delta? Why doing this 'mix'? These different 'types' of clonotypes need either be analysed separately or it needs to be well explained why this analysis is done in such a way. For example, expanded clone 4 differ only by one amino acid in the gamma CDR3 from expanded clone 1, but in expanded clone 1 the gamma chain is put together with the delta chain, while this is not done for expanded clone 4.

We appreciate the reviewer's observations regarding the clonotype quantification process using the Cellranger V(D)J pipeline. Indeed, the pipeline generated numerous clonotypes with only a single productive chain, which most likely occurred due to various factors, including low mRNA expression, especially of the TRD chain. This phenomenon is also observed in the TRB and TRA contig assembly procedures through Cellranger, where low expression of the TRA chain can lead to the unfaithful or no detection of productive TRA contigs at the single-cell resolution. Furthermore, since the primary goal of our TCR analysis was to identify lineage relationships, we reasoned that clonotypes with only one productive chain would still allow us to establish lineage relationships between the identified subsets as 'half-siblings.'

Out of the 2,309 unique clonotypes identified in our data, 1,191 comprised both TRG and TRD chains, while 742 and 376 of the clonotypes were annotated based only on TRG and TRD chains, respectively. We noticed that the resolution of our analysis improved significantly when using this 'mix' of clonotypes, and we were able to gain unique biological insights when considering all annotated clonotypes from Cellranger for understanding the lineage relationships among the identified subsets. Since almost half of the clonotypes were

derived from either the TRG or TRD chain, we observed that focusing solely on those clonotypes with successful contigs of both chains limited our analysis of lineage relationships.

We believe that this point is quite significant, and we have included a complete new supplementary note and a supplementary data figure 1 to illustrate why we chose to conduct the analysis using the clonotype annotation provided by Cell Ranger, even when it included a consortium of 'mix' clonotypes.

For the reviewer's convenience, the following is the note along with Supplementary Data Figure 1:

Defining clonotypes for $\gamma\delta$ TCR repertoire analysis

*We used Cellranger (pipeline Version 7.1.0) to identify distinct clonotypes in the single-cell data. Custom primers were employed to amplify TRG and TRD from single cells. It's important to note that $\gamma\delta$ TCR analysis is not a supported workflow in the Cellranger analysis. Therefore, this note provides details on how clonotypes were defined and used in our analysis. Approximately 25,000 cells were loaded into one well of the 10x Genomics chip K for single-cell immune profiling. Experimentally, cells from three female mice (age 8 weeks) were pooled together before sorting. Cells from five different organs (spleen, liver, lung, lymph node, and small intestine) were stained using hashtag antibodies, pooled, and $\gamma\delta$ T cells were FACS sorted for simultaneous single-cell gene expression and TCR profiling. After doublet removal and quality control, 13,753 cells were included in the analysis. Cellranger assigned $\gamma\delta$ TCR clonotypes to 4,220 cells, which were then used for TCR repertoire analysis (**Supplementary Table 5**). Among these 4,220 cells, we detected 2,309 unique clonotypes. However, it's important to note that we did not detect fully rearranged productive TRG and TRD chains simultaneously in each cell. This phenomenon is also observed in profiling $\alpha\beta$ TCR, primarily because TRA is generally expressed at lower levels than TRB. Therefore, sometimes the TRA chain is not detected. This also appears to be true for the TRD chain, as the median numbers of TRG and TRD UMIs per cell were 11 and 3, respectively. Out of the 2,309 clonotypes, 1,191 comprised both TRG and TRD chains, while 742 and 376 of the clonotypes were annotated based only on TRG and TRD chains, respectively. Although the clonotype annotations provided by Cellranger included a mix, such as those containing both or either of the TRG and TRD chains, they offered better resolution in single-cell TCR data and were therefore used in the analysis. For example, the TRG chain of the most expanded clonotype 1 (CSYGLYSSGFHKVF_CASGYIGGIRATDKLVF) was identical to that of clonotype 49 (CSYGLYSSGFHKVF_NA; NA corresponds to not detected) (**Supplementary Data Fig. 1a, Supplementary Table 5**). However, while clonotype 1 was mainly restricted to *Rorc*⁺ cells, clonotype 49 was mainly found in *Sell*⁺ *Ly6c*²⁻ and *Sell*⁺ *Ly6c*²⁺ cells (**Fig. 6c, Supplementary Data Fig. 1a**). The difference between these clonotypes stems from clonotype 1 being paired with a specific TRD chain, while no productive contig of TRD gene could be assigned to clonotype 49. Therefore, clonotype annotation provided by Cellranger in such cases allowed us to have a highly resolved view of cell type-specific clonotypes. Another similar example is clonotype 3 (CASWAGYSSGFHKVF_NA) and clonotype 77 (CASWAGYSSGFHKVF_CALMERESISEGYPTDKLVF), both exhibiting the rearrangement of an identical TRG chain (**Supplementary Data Fig. 1b, Supplementary Table 5**). While both clonotypes are present in the *Gzmb*⁺ compartment, clonotype 77 was more specific to cluster 2, further indicating the power of using both clonotypes wherever possible (**Fig. 6a, c, Supplementary Data Fig. 1b**). Interestingly, the TRD chain of clonotype 2 (CACWDSSGFHKVF_CGSDIGGSSWDTRQMFF) was identical to that of clonotype 28 (CACWDSSGFHKVF_CGSDIGGSSWDTRQMFF), but it was paired with TRGV6 in clonotype 2 and TRGV5 in clonotype 28 (**Supplementary Data Fig. 1c, Supplementary Table 5**). Importantly, clonotype 2*

was mainly restricted to *Rorc*⁺ cells across organs, while clonotype 28 was mainly present in liver tissue (*Cd160*⁺ and proliferating cells) (Fig. 6c, Supplementary Data Fig. 1c). This further underscores the importance of including either one or both detected clonotypes in the analysis; otherwise, such subtle differences would not be resolved if we only considered either TRG, TRD, or both in the analysis.

Supplementary Data Figure 1

Supplementary Data Figure 1. Defining $\gamma\delta$ TCR clonotypes. a-c, UMAP representation highlighting the cells with clonotypes 1 and 49 (a), 3 and 77 (b) and 2 and 28 (f) having identical TRG rearrangement. Each expanded clonotype is represented in a different color. Amino acid sequence of the complementarity-determining region 3 (CDR3) region is further listed. CT, clonotype.

In addition, clonotype detection was based on nucleotide sequences. Notably, TRG clonotypes 1 and 4 differ by only one amino acid, they exhibit a three-base difference in their nucleotide sequences.

Nucleotide sequence of TRGV4 clonotype 1:

TGTTCCCTACGGCCTATATAGCTCAGGTTTTTACAAGGTATTT

Nucleotide sequence of TRGV4 clonotype 4:

TGTTCCCTACGGCTATAGCTCAGGTTTTTACAAGGTATTT

We are hopeful that our detailed explanation will convince the reviewer that clonotype analysis based on the strategy described in the supplementary note represents the optimal way to extract the most from our dataset.

Is the Shannon index based on such a 'mix' of clonotypes? Again, Shannon index is usually calculated per TCR chain, not a mix.

This is indeed a valid point. Since our clonotype definition is based on the presence of either the gamma CDR3, delta CDR3, or both, the calculation of Shannon entropy was carried out accordingly. However, it is true that the Shannon index is generally calculated per TCR chain. Given that murine $\gamma\delta$ T cells are characterized based on their variable gamma chain

usage, we have revised Figure 6d and extended Figure 8d, calculating the Shannon diversity score based on the TRG chain nucleotide sequences.

Furthermore, the presence of several of the described clonotypes have been described before. An overview can be found in the review by Papadopoulou et al 2020 Immunol Rev (DOI: 10.1111/imr.12926) and thus can be included as a reference.

We sincerely regret the omission of this important reference in our TCR analysis and are grateful to the reviewer for bringing it to our attention. Reassuringly, we have detected all the clonotypes mentioned in the review except for TRAV15-1DV6-1 (please refer to point 2 for a detailed explanation). We genuinely appreciate the reviewer for highlighting this oversight. In our revised version, we have ensured the inclusion of this critical reference in the TCR section of the manuscript, with the following sentence:

Clonotypes were annotated using the cellranger vdj pipeline based on either one or both TRD and TRG chains (Supplementary Data Fig. 1a, b, c, Supplementary Note). Apart from the new clonotypes, we identified all the clonotypes of $\gamma\delta$ T cells previously summarized, except for those exhibiting the TRAV15-1-DV6-1 usage found in NKT-like IFN- γ /IL-4 double producers.

Fig. 8E:

It would be good if the authors expand a bit on the type of the interorgan shared clones that the liver has in more abundance (like usage of Vg chain: Vg6?).

We wholeheartedly acknowledge the validity of this request and have addressed it in the revised version of our manuscript. We've incorporated a bar plot illustrating the usage of TRG chains in the shared clonotypes between the liver and other organs. The results reveal that shared clonotypes between the liver and small intestine predominantly feature TRGV7 chains, while in other tissues, they consist of TRGV1, TRGV2, and TRGV4 chains. Additionally, shared clonotypes between the liver and lung exhibit TRGV6 rearrangements. Due to word limits, we were unable to include these specific details in the main text, but we have included the figure as Extended Data Figure 8f.

2. Detection of NKT-like $\gamma\delta$ TCR clonotype?

The authors describe in the paper (line 173):

‘Importantly, cluster 12 expressed Zbtb16 (encoding PLZF) and IL4, indicating that these cells are NKT-like IFN- γ /IL-4 double producers³⁵.’

It is known that this population express a particular clonotype (CALWELVGGIRATDKLVF

TRAV15-1-DV6-1: see overview in Papadopoulou et al 2020 Immunol Rev). Did the authors did not detect this one as an 'expanded clonotype'? If detected, was it specific for the liver and/or spleen? Note the annotation 'TRAV15-1DV6-1': thus this V gene segment can also be used in an alpha chain: this may need a double check whether such CDR3 data are adequately analysed by the Cell Ranger software.

We appreciate the reviewer for bringing up this point. Upon a more detailed examination of our data, we confirmed that we detected all the clonotypes summarized by Papadopoulou et al. in their 2020 Immunol Rev paper, with one exception - TRAV15-1DV6-1. Upon scrutinizing the unfiltered data, which includes all clonotypes, even those not marked as productive, we noticed that Cellranger did detect this particular clonotype. However, it was not marked as productive, leading to its exclusion from our analysis.

Interestingly, we plotted the cells displaying the unproductive rearrangement of this clonotype on the UMAP. Instead of being concentrated in cluster 12, marked by the expression of *Zbtb16* and *Il4*, this clonotype was scattered across all clusters (kindly see the figures below).

Given our inability to identify a successful rearrangement of this clonotype, as well as a specific rearrangement in cluster 12, we have chosen not to include this analysis in our study. We think that, owing to the lower expression of this chain, we may have missed out on this clonotype in single cells.

Nevertheless, we think it's an important point, and we have mentioned this in a separate paragraph in our supplementary note and Supplementary Data Figures 1d and 1e. The reviewer can read it here:

Clonotype associated with NKT-like IFN- γ /IL-4 double producers

Previous studies have described a semi-invariant TCR with the TRD nomenclature – TRAV15-1-DV6-1, expressed by NKT-like $\gamma\delta$ T cells which produce both IL-4 and IFN- γ , mainly present in the liver and spleen.

Although we clearly identified cluster 12 as NKT-like $\gamma\delta$ T cell cluster in our dataset, also mainly from the liver and spleen, these cells did not exhibit a productive contig of TRAV15-1-DV6-1 (Fig. 6a, b, Supplementary Data Fig. 1d, e, Supplementary Table 5). Cluster 12 comprised 380 cells, out of which we recovered the clonotypes of 174 cells (100 unique clonotypes). In 156 of them, we only detected the TRG chain, indicating that this TRD chain might be lowly expressed in these cells. The cells where both chains were detected, TRDV1 and TRDV2-2, were the predominantly rearranged chains (Supplementary Table 5).

Supplementary Data Figure 1

Supplementary Data Figure 1. Defining $\gamma\delta$ TCR clonotypes. a-c, UMAP representation highlighting the cells with clonotypes 1 and 49 (a), 3 and 77 (b) and 2 and 28 (c) having identical TRG rearrangement. Each expanded clonotype is represented in a different color. Amino acid sequence of the complementarity-

determining region 3 (CDR3) region is further listed. CT, clonotype. **d, e**, UMAP representation showing the normalized transcript counts of *Ii4* (**d**) and *Zbtb16* (**e**). Note that the coexpression of both genes is specific to Cluster 12, as shown in **Fig. 6a**.

3. How many mice were used in the scTCR experiments? While the sequences of other genes are not unique to each mouse, TCR rearrangement are mostly unique (private) per subject/mouse. If different mice were used: did the authors attempt to analyse the scTCR data separately per mouse? Were the data consistent between the different mice? Or were the number of cells not sufficient to perform such analysis?

More generally, the number of mice used in the other types of experiments may also be useful to include. But it is especially relevant to include that information for the scTCR experiments.

In our scTCR-Seq experiment, we opted to pool samples from three mice. This decision was primarily driven by the necessity to obtain a sufficient number of cells, especially in organs where gamma delta T cells are a rare cell type for downstream steps (e.g., centrifugation steps which lead to significant cell loss with low amount of cells). Additionally, we anticipated that this pooling strategy would enhance resolution by allowing us to detect a greater number of clonotypes, which is particularly relevant for predicting lineage relations, as originally pointed out by reviewer 2. We have now incorporated these important details into the revised manuscript in the supplementary note as well as the methods section (Single-cell simultaneous gene expression and TCR profiling). Moreover, we have provided information on the number of mice and independent experiments conducted for all experiments in the corresponding figure legends.

Reviewer #3

(Remarks to the Author)

Sagar and colleagues provide a well prepared and well-presented overview of the diverse more or less tissue-resident gd T cell populations found in WT laboratory C57BL/6 mice. The ms does not comprise a dense and completely new story, as gd T cells are considered prototype tissue-resident T cells. But it will certainly be an extremely timely and valuable „go-to“ resource for many current projects aiming at deciphering the composition and functional diversity of human and mouse gd T cell populations in tissues. (... and highly cited). To this end, data accessibility and re-usability will be crucial for the community.

We extend our sincere gratitude to Reviewer 3 for undertaking the review of our manuscript. We are delighted to learn that the reviewer views our resource as both timely and valuable. We assure the reviewer that the data will be publicly available for the community to reanalyze. We plan not just to release the raw counts through the GEO but also all the scripts to generate the figures, as well as the analyzed Seurat and scRepertoire objects. These will also be available on the corresponding author's GitHub repository for any interested readers in the community to access easily. We thank the reviewer once again for mentioning this point.

Comments:

The current ms is already a revised version of a previous submission. The two reviewers did a great job and most issues were adequately addressed by the authors. In particular, both suggested to include an analysis of clonal relationships between the identified clusters of mouse gd T cells. The authors complied by including a state-of-the art scTCR analysis using the 10x 5' VDJ workflow that is commercially available for ab TCR.

We would like to express our appreciation to the reviewer for recognizing that we have adequately addressed the comments made by the two previous reviewers of our manuscript. Furthermore, we have made efforts to address all the new comments in the current submission, which we believe has significantly improved the manuscript.

1. Fig. 6 Although this method is still a cutting-edge experimental procedure, TCR repertoire analyses (scTCR-seq) could still be more comprehensive (more cells). However, as is, it is already an important and decisive selling point of the current revised ms. In that context, it might be useful to reference and discuss a recent report, which also experimented with combining mouse sc gd TCR-seq and scRNA-seq, doi: 10.1016/j.celrep.2023.112253.

We sincerely thank the reviewer for bringing this significant manuscript to our attention. Regrettably, it was published after the submission of our manuscript, causing us to overlook its inclusion in the references. We have included this report as a reference (no. 84) in the revised version of our manuscript with one extra sentence in the discussion:

While several studies have explored the heterogeneity of tissue-resident $\gamma\delta$ T cells at single-cell resolution^{25, 47, 74, 81, 84, 85}, a comprehensive cross-organ multimodal study detailing their site-specific adaptation and tissue residency features is still lacking.

2. Fig. 5 is fantastic, congratulations!

We are truly pleased to learn that the reviewer found Figure 5 to be favorable. Thank you for your appreciation.

3. I think it is absolutely required to clearly point out which nomenclature is used throughout this resource work.

While we discussed this in our manuscript, we inadvertently omitted the corresponding reference. We have now included the appropriate reference as well (Reference 41).

4. The discussion is concise, very suitable for a resource paper. A very short discussion of the analogy and perhaps differences of tissue-resident gd T cells and ab Trm cells could still be interesting.

We would like to express our gratitude to the reviewer for appreciating the discussion and raising this valid point, which we believe is indeed relevant. We have detected several markers in $\gamma\delta$ T cells that are also relevant for $\alpha\beta$ T cells. Regrettably, due to the manuscript's word limit, we were unable to delve into them in detail. Nevertheless, in response to the reviewer's suggestion, we have now edited the paragraph where we discuss the similarities and differences between tissue-resident $\gamma\delta$ and $\alpha\beta$ T cells. We have incorporated a few additional lines to emphasize this further, in the hope that it adequately addresses the reviewer's concern. Relevant references have also been cited.

*Using the parabiosis mouse model, we lay out the first highly resolved single-cell map of tissue residency features of $\gamma\delta$ T cells across organs. Our approach allowed us to identify numerous tissue-resident and exchanging $\gamma\delta$ subsets. All barrier sites harbor unique tissue-resident subsets, and we did not find a common universal transcriptional program associated with tissue residency. However, a core tissue residency program has been described for $CD8^+$ T_{RM} , NKT and MAIT cells. $CD69$, $CD103$, $CD49a$ and TFs (*Hobit* and *Blimp1*) associated with T_{RM} tissue residency display tissue- and subset-specific expression patterns in resident $\gamma\delta$ T cells. **While *Itgae*, encoding $CD103$, was restricted to skin- and gut-resident $\gamma\delta$ T cells, *Itga1*, encoding $CD49a$, and *Hobit* were predominantly expressed in liver-resident $\gamma\delta$ T cells. Liver T_{RM} subsets have also been shown to be $CD103^+$ and $CD49a^{+22, 87}$. Notably, *Runx3* was uniformly expressed in all tissue-resident $\gamma\delta$ T cells. Although *Runx3* is required for the development of DETCs and regulates $CD103$ expression⁸⁸, its role in establishing $\gamma\delta$ tissue residency in other organs has not been explored. Despite single genes and TFs associated with T_{RM} formation displayed tissue-specific regulation in distinct $\gamma\delta$ subsets, the core signatures associated with T_{RM} , NKT and MAIT cell tissue residency are remarkably analogous to those of all tissue-resident $\gamma\delta$ T cells, suggesting the existence of a core genome-wide transcriptional program associated with tissue residency across all lymphocyte lineages. Furthermore, the circulatory programs associated with effector and central memory $\alpha\beta$ T cells (e.g., *Klf2* and *S1pr1*) were strikingly similar to circulating $\gamma\delta$ T cells, which were mainly present in secondary lymphoid organs, indicating their adaptive-like phenotype. **Systematic and simultaneous analysis of both lineages will shed more light on the parallels of their tissue residency programs.*****

Wish list:

- Multimodal analysis of $\gamma\delta$ T cells also from other inbred strains, or even outbred wildlings would be super exciting, but probably beyond the scope of this study.

We appreciate the excellent point raised by the reviewer. While we do have access to wildling mice in Freiburg, we believe that delving into this aspect, as the reviewer has suggested, would extend beyond the current focus of our study. However, we fully acknowledge the importance of this direction and assure the reviewer that it will be a valuable component of our future research endeavors.

- Analysis or integration of thymic $\gamma\delta$ T cells (e.g., their own data from <https://doi.org/10.15252/emboj.2019104159>) would be a great complementary resource to map $\gamma\delta$ T-cell differentiation and thereby observe some freshly generated $\gamma\delta$ T cells acquiring their tissue-specific effector functions. At the same time, there is accumulating evidence for the presence of long-lived thymus-resident effector $\gamma\delta$ T cells in mice.

We appreciate the reviewer's valid comment, and we want to confirm that we do have data pertaining to this aspect. Interestingly, we have observed different subsets of $\gamma\delta$ T cells identified in the peripheral tissues in this manuscript developing at different timepoints in the thymus. However, our intention was not to include it in this manuscript, as doing so would shift the focus away from the primary theme of tissue adaptation in the periphery, which is the central topic of this study. Furthermore, we have already exceeded the limit of main figures and extended data figures. Adding more data, in our opinion, would make the manuscript less streamlined. Nevertheless, we would like to assure the reviewer that the relationship between the peripheral adaptation of $\gamma\delta$ T cells and their development in the thymus will be explored in detail and published as a separate, independent manuscript after the release of this study.

Minor

a) Line 73: the review article to be cited here should probably be reference #19? Carding, S.R. & Egan, P.J. Gammadelta T cells: functional plasticity and heterogeneity. *Nat Rev Immunol* 2, 336-345 (2002).

We appreciate the reviewer for bringing this to our attention. In this scenario, we believe both references are useful, as both have dedicated figures (Fig. 1 in ref. 9 and Fig. 2 in ref. 19) depicting the wave-like development of $\gamma\delta$ T cells. Therefore, we have decided to cite both of them.

b) Line 87: The context of reference #25 is misleading. The authors claim in the following that "... these studies were performed using flow cytometry with very few cell surface markers, providing limited insights into their tissue residency features. Hence, deciphering the tissue adaptation features of gd T cells in multiple organs using single-cell transcriptomics holds the key to identifying specific tissue-resident gd subsets and the underlying common or specialized regulatory programs governing their tissue residency." This statement misleading because even the title of the work cited as reference #25 starts with "single-cell transcriptomics" and was thus not "performed using flow cytometry with very few cell surface markers" as suggested by the authors. Ref. #25 Tan, L. et al. Single-Cell Transcriptomics Identifies the Adaptation of Scart1(+) Vgamma6(+) T Cells to Skin Residency as Activated Effector Cells. *Cell Rep* 27, 3657- 3671 e3654 (2019).

Indeed, although this was a single-cell study, but we would like to clarify that the parabiosis experiments conducted in this study were primarily based on flow cytometry and were specifically centered on the Vg4 and Vg6 subsets. We acknowledge that this distinction needs to be made more explicit in the manuscript, and we have ensured that in the revised version. The modified sentences read as follows:

*Using parabiotic mice, in which circulating cells can be distinguished from resident cells, it is well established that the skin, intestine, liver and adipose tissue host mainly tissue-resident $\gamma\delta$ T cell populations^{8, 23, 24, 25}. However, **parabiosis experiments in these studies** were performed using flow cytometry with very few cell surface markers, providing limited insights into their tissue residency features.*

Hence, deciphering the tissue adaptation features of $\gamma\delta$ T cells in multiple organs **combining parabiosis with single-cell transcriptomics** holds the key to identifying specific tissue-resident $\gamma\delta$ subsets and the underlying common or specialized regulatory programs governing their tissue residency.

c) The main datasets are combining pooled and hashtagged scRNA libraries, which were all integrated using Harmony to correct for batch effects. A statement evaluating whether this may have led to over-correction would be helpful.

We appreciate the reviewer for bringing this up. We have observed that, in most cases, using Harmony for batch correction does not lead to overcorrection. However, it's important to note that integrating the data across organs with the default Seurat resolution can hinder the identification of the full spectrum of $\gamma\delta$ T cell heterogeneity within each organ. To thoroughly investigate this heterogeneity, separate analyses of each tissue are necessary to fully utilize our resource dataset. As mentioned previously, all the datasets, analyzed objects, and scripts will be made publicly available to facilitate the community's ability to independently analyze their tissue of interest. Due to limited space and our main focus on cross-tissue features, we were unable to delve into the intraorgan heterogeneity. We have also added a supplementary note in the manuscript to provide additional details on this point:

Implications of cross-organ integration on tissue-specific $\gamma\delta$ heterogeneity

*In our study, our primary focus was on identifying the common features of tissue adaptation in $\gamma\delta$ T cells across various organs. To achieve this, we integrated the datasets by analyzing profiled $\gamma\delta$ T cells from different organs using Harmony¹. Furthermore, for our analysis, we utilized the default resolution provided by the Seurat algorithm². As mentioned in the main text, despite identifying 22 clusters in our scRNA-seq data through unsupervised clustering, we observed that differentially expressed genes in each cluster were not mutually exclusive. Instead, many clusters exhibited differential regulation of identical sets of genes. Consequently, we decided not to increase the clustering resolution. However, it is highly likely that the integration of datasets across organs may have obscured certain specific features of $\gamma\delta$ T cells within particular tissues. To investigate this, an analysis of each tissue dataset separately is required to uncover the full spectrum of heterogeneity in $\gamma\delta$ T cells within each tissue. We have already observed several examples that reinforce this conclusion. For instance, cluster 15, consisting of proliferating cells, comprises two transcriptionally distinct subsets of cells in the S, G₂ and M phases of the cell cycle (**Fig. 1c, Extended Data Fig. 1g**). Furthermore, we identified a subset of skin Rorc⁺ cells expressing Areg concentrated at a specific coordinate in the UMAP representation within cluster 5 (**Fig. 1c, d, Extended Data Fig. 3h**), and the same holds true for Rorc⁺ cells expressing V γ 4 TotalSeqTM antibody, which exclusively occupy the right corner of clusters 3 and 5 (**Fig. 1c, Extended Data Fig. 2q**). Although conducting such an analysis to unveil the full spectrum of $\gamma\delta$ T cell heterogeneity goes beyond the scope of this study due to space limitations, our resource data is freely available for readers to download and explore these questions.*

Decision Letter, second revision:

17th Oct 2023

Dear Dr. Sagar,

Thank you for submitting your revised manuscript "Multimodal profiling reveals site-specific adaptation and tissue residency hallmarks of $\gamma\delta$ T cells across organs in mice" (NI-RS35178C). It has now been seen by the original referees and their comments are below. The reviewers find that the paper has improved in revision, and therefore we'll be happy in principle to publish it in Nature Immunology, pending minor revisions to satisfy our editorial and formatting guidelines.

We will now perform detailed checks on your paper and will send you a checklist detailing our editorial and formatting requirements in about one or two weeks. Please do not upload the final materials and make any revisions until you receive this additional information from us.

If you had not uploaded a Word file for the current version of the manuscript, we will need one before beginning the editing process; please email that to immunology@us.nature.com at your earliest convenience.

Thank you again for your interest in Nature Immunology Please do not hesitate to contact me if you have any questions.

Sincerely,

Jamie D K Wilson, D.Phil
Chief Editor
For:

Stephanie Houston, PhD
Senior Editor
Nature Immunology

Reviewer #1 (Remarks to the Author):

The authors addressed adequately my comments.

Final Decision Letter:

Dear Dr. Sagar,

I am delighted to accept your manuscript entitled "Multimodal profiling reveals site-specific adaptation and tissue residency hallmarks of $\gamma\delta$ T cells across organs in mice" for publication in an upcoming issue of Nature Immunology.

Over the next few weeks, your paper will be copyedited to ensure that it conforms to Nature Immunology style. Once your paper is typeset, you will receive an email with a link to choose the appropriate publishing options for your paper and our Author Services team will be in touch regarding any additional information that may be required.

Please note that *Nature Immunology* is a Transformative Journal (TJ). Authors may publish their research with us through the traditional subscription access route or make their paper immediately open access through payment of an article-processing charge (APC). Authors will not be required to make a final decision about access to their article until it has been accepted. [Find out more about Transformative Journals](https://www.springernature.com/gp/open-research/transformative-journals).

Your paper will be published online soon after we receive your corrections and will appear in print in the next available issue. Content is published online weekly on Mondays and Thursdays, and the embargo is set at 16:00 London time (GMT)/11:00 am US Eastern time (EST) on the day of publication. Now is the time to inform your Public Relations or Press Office about your paper, as they might be interested in promoting its publication. This will allow them time to prepare an accurate and satisfactory press release. Include your manuscript tracking number (NI-RS35178D) and the name of the journal, which they will need when they contact our office.

About one week before your paper is published online, we shall be distributing a press release to news organizations worldwide, which may very well include details of your work. We are happy for your institution or funding agency to prepare its own press release, but it must mention the embargo date

and Nature Immunology. Our Press Office will contact you closer to the time of publication, but if you or your Press Office have any enquiries in the meantime, please contact press@nature.com.

Also, if you have any spectacular or outstanding figures or graphics associated with your manuscript - though not necessarily included with your submission - we'd be delighted to consider them as candidates for our cover. Simply send an electronic version (accompanied by a hard copy) to us with a possible cover caption enclosed.

Please note that we encourage the authors to self-archive their manuscript (the accepted version before copy editing) in their institutional repository, and in their funders' archives, six months after publication. Nature Portfolio recognizes the efforts of funding bodies to increase access of the research they fund, and strongly encourages authors to participate in such efforts. For information about our editorial policy, including license agreement and author copyright, please visit www.nature.com/ni/about/ed_policies/index.html

Sincerely,

Stephanie Houston, PhD

Senior Editor
Nature Immunology